# Non-Stationary Lipschitz Bandits

**Nicolas Nguyen**
University of Tübingen
nicolas.nguyen@uni-tuebingen.de

**Solenne Gaucher**
École Polytechnique, CMAP
solenne.gaucher@polytechnique.edu

**Claire Vernade**
University of Tübingen
claire.vernade@uni-tuebingen.de

## Abstract

We study the problem of non-stationary Lipschitz bandits, where the number of actions is infinite and the reward function, satisfying a Lipschitz assumption, can change arbitrarily over time. We design an algorithm that adaptively tracks the recently introduced notion of significant shifts, defined by large deviations of the cumulative reward function. To detect such reward changes, our algorithm leverages a hierarchical discretization of the action space. Without requiring any prior knowledge of the non-stationarity, our algorithm achieves a minimax-optimal dynamic regret bound of $\widetilde{\mathcal{O}}(\tilde{L}^{1/3}T^{2/3})$, where $\tilde{L}$ is the number of significant shifts and $T$ the horizon. This result provides the first optimal guarantee in this setting.

## 1 Introduction

Many practical applications involve decision-making over a continuous space of actions (or *arms*) where the environment evolves over time. For example, in dynamic pricing, the space of possible prices is inherently continuous, and market dynamics may shift abruptly. In such a setting, it is both natural and crucial to leverage smoothness in the reward function to generalize across similar actions, while also remaining adaptive to temporal changes. However, most existing works on non-stationary bandits focus on discrete action spaces and overlook the structure present in continuous domains.

The non-stationary bandit problem extends the classical stochastic multi-armed bandit setting [Slivkins et al., 2019, Lattimore and Szepesvári, 2020] by allowing the mean reward functions to change over time. Over the past two decades, several frameworks have been proposed to quantify non-stationarity. Common approaches quantify the number of changes $L_T$ of any arm's mean reward over $T$ rounds [Auer et al., 2002b, Garivier and Moulines, 2011], or their total variation $V_T$ in the reward functions [Besbes et al., 2014, Cheung et al., 2019]. Algorithms that are built upon these approaches achieve dynamic regret bounds of order $\sqrt{L_T T}$ or $V_T^{1/3}T^{2/3}$ respectively. Importantly, they all assume prior knowledge of the degree of non-stationarity $L_T$ or $V_T$. It remained an open question whether such rates could be achieved without knowing $L_T$ or $V_T$, until a line of work answered this affirmatively [Auer et al., 2019a,b, Chen et al., 2019, Wei and Luo, 2021].

Recent works have introduced more refined measures of non-stationarity. In fact, the standard metrics $L_T$ and $V_T$ can be overly pessimistic: both of them may be arbitrarily large even when the best arm remains unchanged. To address this, Abbasi-Yadkori et al. [2023] designed an algorithm that tracks the number of times the best arm changes over $T$ rounds, $S_T$, and showed that it achieves $\sqrt{S_T T}$ for the dynamic regret, *adaptively*[1]. The important work of Suk and Kpotufe [2022] proposed a finer notion of non-stationarity called *significant shifts*, which identifies only changes that meaningfully

---

[1]In all our work, "adaptively" should be understood the same as "without knowledge of the non-stationarity".

39th Conference on Neural Information Processing Systems (NeurIPS 2025).

affect regret through large aggregate differences in reward. In the $K$-armed bandit setting, their algorithm achieves the optimal rate of $\sqrt{\tilde{L}_T T}$ where $\tilde{L}_T$ is the number of significant shifts over $T$ rounds, with $\tilde{L}_T \ll L_T$.

In this work, we extend the ideas of Suk and Kpotufe [2022] to the setting of Lipschitz bandits, where the arm space is continuous (typically the interval $[0, 1]$) and the reward functions satisfy a Lipschitz condition at each round. Despite the maturity of research in both non-stationary and Lipschitz bandits individually, their intersection remains unexplored. Our work is the first one to tackle the problem of non-stationary Lipschitz bandits.

## 1.1 Outline and contributions

In this work, we answer the following question positively: *Can we design an algorithm that adaptively achieves the minimax-optimal rate in non-stationary Lipschitz bandits?* We argue that non-stationary Lipschitz bandits pose unique and fundamental challenges that go beyond a simple combination of non-stationarity and Lipschitz continuity. We highlight two central challenges addressed in this paper.

**Discretization level.** In the stationary Lipschitz bandit setting, a common approach is to discretize the continuous arm space into a finite set of *bins*, thereby allowing the application of standard multi-armed bandit algorithms [Kleinberg, 2004]. When the time horizon $T$ is known, the discretization level can be carefully calibrated, typically using $\mathcal{O}(T^{1/3})$ bins. This choice reflects the optimal balance between discretization error and the regret incurred by running a bandit algorithm over the discretized arms, leading to the minimax-optimal regret of $\mathcal{O}(T^{2/3})$. In the non-stationary setting, however, the durations of the time intervals during which the mean reward remains stable are unknown and not rigorously defined. Thus, direct horizon-dependent tuning becomes infeasible. This raises a key question: how to adaptively adjust the discretization level to match the unknown degree of non-stationarity in the environment?

**Forced exploration via restarts and replays.** In non-stationary bandits, detecting changes in the reward distribution requires frequent exploration of suboptimal arms. In the $K$-armed bandit setting, this is often handled through periodic restarts, where all $K$ arms are re-explored to detect shifts of the reward function. In the continuous setting, this strategy is significantly harder: not only must we decide when to restart, but also at what resolution to discretize the arm space during such restarts. Finer discretization enables more precise detection of changes but significantly increases the cost of exploration. Balancing this trade-off is a central technical challenge tackled in our work.

This paper is organized as follows. In Section 2, we formally introduce the problem setting, and present our algorithm, MDBE, in Section 3. Our main results are stated in Section 4, followed by a high-level proof sketch of its regret analysis in Section 5. For completeness, the full pseudocode of MDBE is provided in Appendix B, and all detailed regret proofs can be found in Appendix D.

## 1.2 Related Work

**Non-stationary bandits beyond $K$-armed.** A growing body of work has extended non-stationary bandit models beyond the classical $K$-armed setting. These include linear and generalized linear bandits [Russac et al., 2019, Faury et al., 2021], contextual bandits [Suk and Kpotufe, 2021], convex bandits [Zhao et al., 2021, Wang, 2023, Liu et al., 2025], kernelized bandits [Hong et al., 2023, Iwazaki and Takeno, 2025, Cai and Scarlett, 2025], and more structured problems [Seznec et al., 2020, Vernade et al., 2020, Azizi et al., 2022]. Some works further explore temporal regularity assumptions, such as smooth or slowly varying reward functions [Slivkins and Upfal, 2008, Krishnamurthy and Gopalan, 2021, Jia et al., 2023]. Across these settings, non-stationarity is typically quantified using either the number of global changes $L_T$ or the total variation $V_T$.

**Bandits with infinite arms.** In Lipschitz bandits [Agrawal, 1995], the infinite arm space is tackled via discretization strategies, either fixed or adaptive, that exploit the Lipschitz continuity of the reward function [Kleinberg, 2004, Kleinberg et al., 2008, Bubeck et al., 2011b,a, Magureanu et al., 2014]. More recently, adversarial formulations have been studied, where the goal is to minimize the *regret in hindsight* [Podimata and Slivkins, 2021, Kang et al., 2023b]. These differ fundamentally from our focus on dynamic regret in a stochastic setting. A separate line of work investigates infinite-armed bandits in the so-called *reservoir model*, where arm means are drawn independently from a fixed distribution [Wang et al., 2008, Carpentier and Valko, 2015, De Heide et al., 2021, Kim et al., 2022,

2024]. Unlike Lipschitz bandits, these models assume no structural relationship between arms and therefore do not exploit similarity across them. Hence, direct comparisons to our setting do not apply. The only work we are aware of that addresses the problem of non-stationary Lipschitz bandits is Kang et al. [2023a], who propose a zooming-like algorithm to tune a generalized bandit framework. However, their formulation of non-stationary Lipschitz bandits is tailored to their specific setting and does not achieve optimal theoretical guarantees.

**Tracking significant shifts.** The idea of tracking *significant shifts* was first introduced in the $K$-armed setting by Suk and Kpotufe [2022], and has since been extended to contextual bandits [Suk and Kpotufe, 2023], preference-based settings [Suk and Agarwal, 2023, Buening and Saha, 2023, Suk and Agarwal, 2024], and smooth non-stationary models with temporal structure [Suk, 2024].

The works closest to ours are Suk and Kpotufe [2023] and Suk and Kim [2025], both extending the framework of tracking significant shifts beyond the classical $K$-armed setting. The former considers contextual bandits where rewards are Lipschitz in the context space, but with a finite number of arms. After discretizing the context space, their algorithm effectively reduces to a finite-arm bandit problem at each context, allowing the use of techniques from the $K$-armed setting [Suk and Kpotufe, 2022] without requiring discretization of the arm space. The latter studies infinite-armed bandits under a reservoir model, where arms are independently drawn from a fixed distribution. While this shares the challenge of choosing from infinitely many arms, their setting assumes no regularity across arms. Therefore, their techniques and guarantees are not applicable to our setting.

## 1.3 Notations

For any integer $n$, we write $[\![n]\!] = \{1, \ldots, n\}$. For a filtration $(\mathcal{F}_t)_t$, we denote the conditional expectation by $\mathbb{E}_t\left[\cdot\right] = \mathbb{E}\left[\cdot \mid \mathcal{F}_t\right]$ (the underlying filtration will be clear from context). For real numbers $a$ and $b$, we write $a \vee b = \max\{a, b\}$ and $a \wedge b = \min\{a, b\}$.

## 2 Problem setting

Without loss of generality, we assume that the arm space is the unit interval $[0, 1]$.[2] An *oblivious adversary* selects a sequence of $T$ mean reward functions $(\mu_t)_{t=1}^T$, where each $\mu_t : [0, 1] \to [0, 1]$ satisfies the following Lipschitz condition:

**Assumption 1 (Lipschitz mean rewards $\mu_t$).** $\forall x, x' \in [0, 1], |\mu_t(x) - \mu_t(x')| \leq |x - x'|$.

At each round $t \in [T]$, an algorithm $\pi$ selects an arm $x_t \in [0, 1]$ and observes a stochastic reward $Y_t(x_t) \in [0, 1]$ independent from the past (arms and observations) and satisfying $\mathbb{E}\left[Y_t(x_t) \mid x_t = x\right] = \mu_t(x)$. The objective is to minimize the *dynamic regret*

$$R(\pi, T) = \sum_{t=1}^T \sup_{x \in \mathcal{X}} \mu_t(x) - \sum_{t=1}^T \mu_t(x_t),$$

defined as the difference between the cumulative gain of the algorithm and that of the dynamic benchmark oracle, which, at each round, knows the reward function $\mu_t$. In this paper, we design an algorithm and provide a bound on its *expected dynamic regret* $\mathbb{E}[R(\pi, T)]$, where the expectation is taken with respect to the randomness of the interactions between the algorithm and the environment. We denote the instantaneous gap between two arms $x'$ and $x$ as $\delta_t(x', x) = \mu_t(x') - \mu_t(x)$, and the instantaneous regret of arm $x$ as $\delta_t(x) = \max_{x' \in [0,1]} \mu_t(x') - \mu_t(x)$, so that $R(\pi, T) = \sum_{t=1}^T \delta_t(x_t)$.

### 2.1 Significant regret and significant shifts

We now introduce the notion of *significant regret* for an arm $x \in [0, 1]$.

**Definition 1 (Significant regret of an arm $x$).** *An arm $x \in \mathcal{X}$ incurs significant regret on interval $[s_1, s_2]$ if its cumulative regret on this interval is lower bounded as*

$$\sum_{t=s_1}^{s_2} \delta_t(x) \geq \log(T)(s_2 - s_1)^{2/3}.$$

*We call such arm an **unsafe** arm on interval $[s_1, s_2]$, and otherwise we call it **safe** on this interval.*

---

[2]Extensions to arbitrary metric spaces and other Lipschitz constants are discussed in Appendix G.

The right hand side of Definition 1 corresponds to the minimax-optimal regret for a stationary Lipschitz bandit over horizon $s_2 - s_1$, and thus captures the level of regret that is statistically detectable. Conversely, if no such interval exists where arm $x$ incurs significant regret, then its cumulative regret over the entire horizon must remain small. A *significant shift* occurs when *all* arms become unsafe, *i.e.* when each arm has incurred significant regret over some interval.

**Definition 2** (**Significant shift, significant phase**). *Let $\tau_0 = 1$. For $i \geq 1$, we define the $i^{th}$ significant shift as the smallest round $\tau_i \in ]\tau_{i-1}, T]$ such that for all arm $x \in \mathcal{X}$, there exists an interval $[s_1, s_2] \subseteq [\tau_{i-1}, \tau_i]$ on which arm $x$ incurs significant regret (Definition 1). For all $i$, we call $[\tau_i, \tau_{i+1}[$ a significant phase, and we denote by $\tilde{L}_T$ the (unknown) number of significant phases.*

This definition implies that tracking significant shifts amounts to detecting intervals where the regret exceeds what would be expected in a stationary Lipschitz environment.

## 2.2 Comparison with other non-stationary metrics.

The notion of significant shifts provides a refined perspective compared to classical metrics such as the *number of global changes* $L_T = \sum_{t=1}^{T-1} \mathbb{I}\{\exists x \in [0,1] : \mu_{t+1}(x) \neq \mu_t(x)\}$, the *total variation* $V_T = \sum_{t=1}^{T-1} \max_{x \in [0,1]} |\mu_{t+1}(x) - \mu_t(x)|$, and the *number of best-arm changes* over $T$ rounds, $S_T = \sum_{t=1}^{T-1} \mathbb{I}\left\{\arg\max_{x \in [0,1]} \mu_{t+1}(x) \neq \arg\max_{x \in [0,1]} \mu_t(x)\right\}$. Unlike these metrics, $\tilde{L}_T$ is robust to benign transformations. For instance, if each $\mu_t$ is shifted by the same constant across all $x$, then $L_T$ and $V_T$ may scale linearly with $T$, while $\tilde{L}_T = 0$. Similarly, $S_T$ may count frequent changes in the optimal arm due to small fluctuations, even if the overall impact is negligible. In contrast, $\tilde{L}_T$ captures only statistically meaningful shifts and satisfies $\tilde{L}_T \leq S_T \leq L_T$.

For example, consider a recommendation system with a continuous pool of content (*e.g.* movies), indexed by $x \in [0,1]$, where nearby values of $x$ correspond to similar content types. Suppose there exist two regions of high and comparable user preference, centred around $x_1 = 0.3$ and $x_2 = 0.7$. Imagine a scenario where preferences near $x_1$ remain stable over time (*e.g.* a timeless classic), while preferences near $x_2$ undergo very frequent changes (*e.g.* a trending topic that evolves daily). In this case, an algorithm that consistently recommends content near $x_1$ would incur little to no regret, even though the underlying reward function changes frequently in other regions. From a global perspective, the number of changes or the total variation in mean reward could be as large as $L_T = V_T = \mathcal{O}(T)$. An algorithm that relies solely on such metrics would unnecessarily restart its estimates too frequently, as it would overestimate the effective difficulty of the problem. This leads to both theoretical sub-optimality and practical inefficiency. However, such changes do *not* constitute a significant shift, as the latter captures only changes that are *statistically consequential* to learning, rather than indiscriminately counting all shifts in the environment.

## 3 Algorithm MDBE: Multi-Depth Bin Elimination

We now present the key ideas leading to the design of our algorithm, before detailing each one of them[3]. At a high level, our goal is to detect significant changes in the aggregate gaps $\sum_{t=s_1}^{s_2} \delta_t(x', x)$ between pairs of arms over time intervals. This quantity lower bounds the dynamic regret $\sum_{t=s_1}^{s_2} \delta_t(x)$ over this interval, meaning that large changes in these aggregate gaps are indicative of large dynamic regret. However, directly monitoring such changes for all possible pairs of arms is infeasible in a continuous action space. To address this, we adopt a **discretization** strategy and instead monitor aggregate gaps between pairs of *bins* $B, B'$ (*i.e.*, subintervals of $[0,1]$) in the form $\sum_{t=s_1}^{s_2} \delta_t(B', B)$.

Our algorithm operates in stable **episodes**, each split into **blocks** of doubling length with varying discretization levels. Within each block, a variant of the *Successive Elimination* algorithm removes empirically suboptimal regions. Since discarded bins may become optimal later, we use the replay mechanism of Suk and Kpotufe [2022] to revisit them and avoid missing promising regions. A key challenge is selecting the right discretization level for each replay. Our **sampling procedure** guarantees sufficient exploration at different scales, while a carefully designed **bin eviction mechanism** progressively eliminates regions that appear suboptimal based on empirical evidence.

---

[3]The full pseudo-code of Multi-Depth Bin Elimination (MDBE) can be found in Appendix B.

**Discretization.** To estimate the reward function at multiple discretization levels, we use a recursive dyadic partitioning of the space, described below.

**Definition 3** (**Depth and Dyadic tree**). *$\mathcal{T}_d$ denotes the partition of $[0,1]$ into $2^d$ bins of size $1/2^d$ each. We say that $d$ is the corresponding **depth** of this discretization. We define the **dyadic tree** $\mathcal{T} = \{\mathcal{T}_d\}_{d\in\mathbb{N}}$ as the hierarchy of nested partitions of $[0,1]$ at all possible depth $d \in \mathbb{N}$. For any bin $B \in \mathcal{T}_d$, we denote by $\mathrm{Children}(B,d')$ the **set of children** of bin $B$ at depth $d' > d$, and $\mathrm{Parent}(B,d'')$ the unique **parent bin** of $B$ at depth $d'' < d$. In particular we have $d'' < d < d' \implies \forall B' \in \mathrm{Children}(B,d'),\ B' \subset B \subset \mathrm{Parent}(B,d'')$.*

For any bin $B \in \mathcal{T}_d$, we define the mean reward of bin $B$ at round $t \in [\![T]\!]$ as

$$\mu_t(B) = \frac{1}{|B|} \int_{x \in B} \mu_t(x)\mathrm{d}x\,,$$

where $|B| = 1/2^d$ denotes the width of bin $B \in \mathcal{T}_d$. We define the instantaneous relative regret between two bins $B, B' \in \mathcal{T}_d$ as $\delta_t(B',B) = \mu_t(B') - \mu_t(B)$, and introduce its empirical estimate, to be formalized later, as $\hat{\delta}_t(B',B) = \hat{\mu}_t(B') - \hat{\mu}_t(B)$. We also define the instantaneous regret of a bin $B \in \mathcal{T}_d$ as $\delta_t(B) = \max_{B' \in \mathcal{T}_d} \mu_t(B') - \mu_t(B)$. For simplicity, we use the same notation $\mu_t(B), \delta_t(B)$ and $\mu_t(x), \delta_t(x)$ to denote the mean and regret for bins and arms, respectively. The distinction should be clear from context.

**Episodes and blocks.** The algorithm proceeds in *episodes*, indexed by $l \in \mathbb{N}$, with episode $l$ starting at round $t_l$ (with $t_0 = 1$). Each episode is divided into *blocks*, where the $m^{\text{th}}$ block lasts for $8^m$ rounds. Within the $m^{\text{th}}$ block, the action space $[0,1]$ is discretized into $2^m = (8^m)^{1/3}$ bins, corresponding to the $m^{\text{th}}$ depth $\mathcal{T}_m$. We denote the block interval by $[\tau_{l,m}, \tau_{l,m+1}[$, and maintain a set $\mathcal{B}_{\text{MASTER}}(m) \subseteq \mathcal{T}_m$ of *safe bins* at depth $m$ (referred to as the MASTER set). A *significant shift* is detected when $\mathcal{B}_{\text{MASTER}}(m) = \emptyset$, meaning all bins at depth $m$ have been evicted. In such cases, the block (and thus the episode) ends prematurely. Otherwise, a block ends naturally after $8^m$ rounds.

---

**Algorithm 1** Routine procedure for one block

---

**Input:** Starting round of the block $\tau_{l,m}$.
$\mathcal{B}_{\text{MASTER}}(m) \leftarrow \mathcal{T}_m$;         `// Initialize MASTER set with all bins at depth` $m$
Schedule replays for $t = \tau_{l,m} + 1, \ldots, \tau_{l,m} + 8^m - 1$ and $d \in [\![m-1]\!]$ according to (1);
  **for** $t = \tau_{l,m}, \ldots, \tau_{l,m} + 8^m - 1$ **do**
    **if** *Enter replay at depth $d$* **then**
      $\mathcal{D}_t \leftarrow \mathcal{D}_t \cup \{d\}$;                           `// Activate depth` $d$ `for replay`
    **if** *Exit replay at depth $d$* **then**
      $\mathcal{D}_t \leftarrow \mathcal{D}_t \setminus \{d\}$;                           `// Deactivate depth` $d$ `after replay ends`
    Choose $x_t$ using **Sampling scheme** (Algorithm 2);
    Update mean estimates and evict bins using ($\star$);
    **if** $\mathcal{B}_{\text{MASTER}}(m) = \emptyset$ **then**
      **Break**; terminate episode $l \leftarrow l + 1$, restart from block $m \leftarrow 1$;     `// Shift detected`
Change block: $m \leftarrow m + 1, \tau_{l,m+1} \leftarrow \tau_{l,m} + 8^m$;                     `// If` $\mathcal{B}_{\text{MASTER}}(m) \neq \emptyset$

---

At each round $t \in [\![T]\!]$, `MDBE` maintains a set of *active depths* $\mathcal{D}_t \subset \mathbb{N}$. For each active depth $d \in \mathcal{D}_t$, the algorithm maintains a set of *active bins* $\mathcal{B}_t(d) \subseteq \mathcal{T}_d$, which are those not yet evicted based on their observed performance. If $d$ is not active, *i.e.* $d \notin \mathcal{D}_t$, then $\mathcal{B}_t(d) = \emptyset$.

**Replays.** To detect non-stationarity at different scales, the algorithm performs *replays* at various depths. Each block $[\tau_{l,m}, \tau_{l,m+1}[$ always initiates a replay at depth $m$ by setting $\mathcal{B}_{\text{MASTER}}(m) \leftarrow \mathcal{T}_m$. Additionally, replays may be triggered for all rounds $s = \tau_{l,m} + 1, \ldots, \tau_{l,m} + 8^m - 1$ at coarser depths $d < m$ with probability

$$p_{s,d} = \sqrt{\frac{8^d}{s - \tau_{l,m}}}\mathbb{1}\left\{s - \tau_{l,m} \equiv 0[8^d]\right\}\,. \tag{1}$$

Each replay at depth $d$ runs for $8^d$ rounds, unless it is interrupted by the end of the current block. At the beginning of such a replay, the set of *active bins* at each active depth $d \in \mathcal{D}_t$ is reset to contain

*all* bins at that depth: $\mathcal{B}_t(d) \leftarrow \mathcal{T}_d$ for all $d \in \mathcal{D}_t$. When the replay at depth $d$ concludes, depth $d$ is simply removed from the set of active depths $\mathcal{D}_t$, but any other active depths may continue their replays independently. Thanks to this design, multiple replays at different depths can overlap in time, allowing the algorithm to monitor for potential significant shifts at multiple scales simultaneously (Algorithm 1).

**Sampling scheme.** At each round, the algorithm selects an action using a hierarchical, top-down sampling procedure from the *minimum active depth* $d_0(t) = \min \mathcal{D}_t$ down to the deepest active depth that contains an active bin (Algorithm 2). Remark that if $\mathcal{D}_t = \{m\}$, then $\mathcal{B}_t(m) = \mathcal{B}_{\mathrm{MASTER}}(m)$. In this case, the sampling procedure reduces to selecting a bin uniformly from $\mathcal{B}_{\mathrm{MASTER}}(m)$, and subsequently sampling an action from the chosen bin (see Figure 1).

---

**Algorithm 2** Sampling scheme

---

**Input:** Round $t$, active depths $\mathcal{D}_t$, active bins $(\mathcal{B}_t(d))_{d \in \mathcal{D}_t}$.
Compute $d_0(t) = \min \mathcal{D}_t$ ;              // Identify the minimum active depth
$B_{\mathrm{parent}} \sim \mathcal{U}(\mathcal{B}_t(d_0(t)))$;         // Sample a parent bin at depth $d_0(t)$
**for** $d \in \mathrm{Sort}(\mathcal{D}_t) \setminus \{d_0(t)\}$ **do**
    **if** $\mathrm{Children}(B_{\mathrm{parent}}, d) \cap \mathcal{B}_t(d) = \emptyset$;       // No active child bin found
    **then**
        $x_t \sim \mathcal{U}(B_{\mathrm{parent}})$;            // Sample uniformly from current bin
        **Return** $x_t$;
    **else**
        $B_{\mathrm{child}} \sim \mathcal{U}(\mathrm{Children}(B_{\mathrm{parent}}, d) \cap \mathcal{B}_t(d))$;   // Sample an active child bin
        $B_{\mathrm{parent}} \leftarrow B_{\mathrm{child}}$;            // Continue to the selected node

---

**Bin eviction criterion.** Our algorithm aims to identify and eliminate arms that incur significant regret, as defined in Definition 1. However, since the algorithm only relies on bin estimates $\delta_t(B)$, we introduce a notion of *significant regret for bins* to make localized decisions.

**Definition 4 (Significant regret for a bin).** *We say a bin $B \in \mathcal{T}_d$ incurs significant regret on interval $[s_1, s_2]$ if*

$$\sum_{t=s_1}^{s_2} \delta_t(B) \geq 3 \log(T) \sqrt{(s_2 - s_1) 2^d} \,.$$

*We call such bin an **unsafe bin** on this interval, and otherwise we call it **safe** on this interval.*

The right-hand side of the inequality corresponds to the rate of the minimax regret of a $2^d$-armed bandit over interval $[s_1, s_2]$. Intuitively, this threshold captures whether the cumulative sub-optimality of the bin is statistically significant over $[s_1, s_2]$. Importantly, we will later show that if a bin meets this threshold, then all arms within the bin also suffer significant regret in the sense of Definition 1 (see Proposition 1). We emphasize that the notion of significant shifts (Definition 2) is defined *independently of any depth* (and thus independently of the depth used by the algorithm).
To estimate the mean reward of a bin $B$, we use the following importance-weighted estimator:

$$\forall B \in \mathcal{T}_d, \quad \hat{\mu}_t(B) = \frac{Y_t(x_t)}{\mathbb{P}(x_t \in B \,|\, \mathcal{F}_{t-1})} \mathbb{1}\{x_t \in B\} \,, \tag{2}$$

where $\mathcal{F}_t$ is the natural filtration generated by the algorithm. Under uniform sampling within bin $B$, $\hat{\mu}_t(B)$ is an unbiased estimate of $\mu_t(B)$. In particular, during a replay at depth $d$, the algorithm leverages these estimates to potentially evict bins across active depths $d' \geq d$ that appear suboptimal: over some $[s_1, s_2]$ of length at most $8^d$ ($d$ is active), bin $B$ is evicted if

$$\max_{B' \in \mathcal{B}_{[s_1, s_2]}(d)} \sum_{t=s_1}^{s_2} \hat{\delta}_t(B', B) > c_0 \log(T) \sqrt{(s_2 - s_1) 2^d \vee 4^d} + \frac{4(s_2 - s_1)}{2^d}, \tag{$\star$}$$

where $c_0$ is a positive universal constant[4], and $\mathcal{B}_{[s_1, s_2]}(d)$ denotes the set of bins that remain active throughout interval $[s_1, s_2]$. Importantly, when a bin $B \in \mathcal{T}_d$ is evicted based on this criterion, it is

---

[4]An exact value of $c_0$ can be derived from the analysis.

immediately removed from the current set of active bins $\mathcal{B}_t$, and also permanently **removed from the global MASTER set** $\mathcal{B}_{\mathrm{MASTER}}(m)$. In addition, all children of $B$ in the dyadic tree are evicted as well. As a result, no future arms will be sampled from the entire region corresponding to $B$ and its descendants in $\mathcal{T}$ until a new replay is scheduled. The next proposition establishes that the eviction rule $(\star)$ is sound: evicted bins indeed correspond to areas where all arms suffer significant regret.

**Proposition 1** (**Significant regret of a bin implies significant regret of an action**). *If a bin $B \in \mathcal{T}_d$ incurs significant regret on an interval $[s_1, s_2]$ with $s_2 - s_1 \le 8^d$, then every point $x \in B$ also incurs significant regret over $[s_1, s_2]$.*

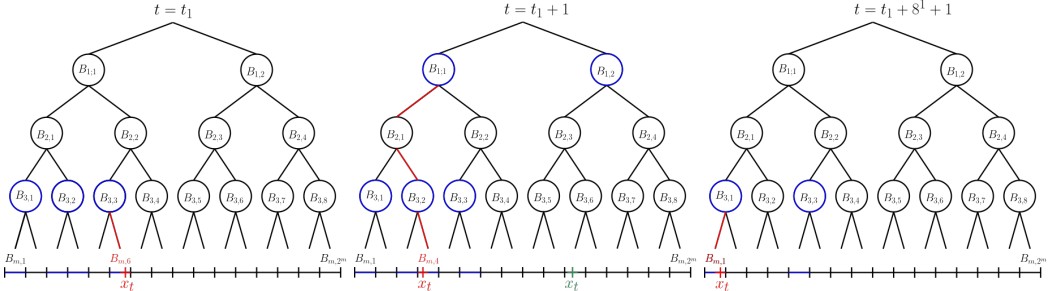

Figure 1: Example of sampling with $m = 4$; active bins are in blue. *Left*: At time $t_1$, depths 3 and $m$ are active. A sample path may select bin $B_{3,3}$ uniformly at random (*u.a.r.*) at depth 3, then $B_{m,6}$ *u.a.r.* among its active children, then arm $x_t$ *u.a.r.* in $B_{m,6}$ (red path). *Center*: At $t_1 + 1$, a replay starts at depth 1. A path may go through $B_{1,1} \to B_{3,2} \to B_{m,4}$, selecting $x_t$ in $B_{m,4}$ (red path). Alternatively, $B_{1,2}$ could be chosen; with no active children, $x_t$ is sampled directly from it (green choice). *Right*: At $t_1 + 9$, depth 1 exits replay. Bin has been $B_{3,2}$ eliminated during the replay, and a path may select $B_{3,1} \to B_{m,1}$, then $x_t$ in $B_{m,1}$ (red path).

## 4 Main results

We establish the fundamental limits for the non-stationary Lipschitz bandit problem, and prove that our algorithm MDBE achieves minimax optimal dynamic regret. We first derive a lower bound on the dynamic regret, expressed in terms of both the number of significant shifts $\tilde{L}_T$ and the total variation $V_T$. To the best of our knowledge, this is the first lower bound for non-stationary Lipschitz bandits.

**Theorem 1** (**Lower bound on the dynamic regret**). *Let $\mathcal{E}_{\mathrm{Lip}}(T, \tilde{L})$ denote the class of non-stationary Lipschitz bandit environments over the arm space $[0, 1]$ with at most $\tilde{L}_T \le \tilde{L}$ significant shifts. There exists a numerical constant $\underline{c}_1 > 0$ such that for any algorithm $\pi$,*

$$\sup_{\boldsymbol{\mu} \in \mathcal{E}_{\mathrm{Lip}}(T, \tilde{L})} \mathbb{E}_{\boldsymbol{\mu}}\left[R(\pi, T)\right] \ge \underline{c}_1 \tilde{L}^{1/3} T^{2/3}.$$

*Similarly, for the class $\mathcal{E}_{\mathrm{Lip}}(T, V)$ with at most $V_T \le V$ total variation, there exists a numerical constant $\underline{c}_2 > 0$ such that for any algorithm $\pi$,*

$$\sup_{\boldsymbol{\mu} \in \mathcal{E}_{\mathrm{Lip}}(T, V)} \mathbb{E}_{\boldsymbol{\mu}}\left[R(\pi, T)\right] \ge \underline{c}_2 \left(T^{2/3} + V^{1/4} T^{3/4}\right).$$

To give intuition whether the optimal rate $\mathcal{O}(\tilde{L}^{1/3} T^{2/3})$ is achievable, consider an *oracle algorithm* $\pi_{\mathrm{oracle}}$ with access to the exact times of the significant shifts $\{\tau_i\}_{i=0}^{\tilde{L}_T}$. The optimal algorithm partitions the horizon into $\tilde{L}_T$ phases and, within each phase, runs a minimax-optimal bandit algorithm (*e.g.* UCB [Auer et al., 2002a]) using a discretization of $(\tau_{i+1} - \tau_i)^{1/3}$ arms. This oracle algorithm suffers regret at most $c_{\mathrm{oracle}}(\tau_{i+1} - \tau_i)^{2/3}$ per phase, where $c_{\mathrm{oracle}} > 0$ is a numerical constant, yielding

$$\mathbb{E}\left[R(\pi_{\mathrm{oracle}}, T)\right] \le c_{\mathrm{oracle}} \sum_{i=0}^{\tilde{L}_T} (\tau_{i+1} - \tau_i)^{2/3} \le c_{\mathrm{oracle}} \tilde{L}_T^{1/3} T^{2/3}.$$

However, this oracle relies on complete knowledge of $\tau_i$'s. In contrast, our algorithm MDBE is *fully adaptive*. Theorem 2 proves that it achieves minimax optimal regret, up to poly-logarithmic factors.

**Theorem 2** (**Adaptive upper bound on dynamic regret**). *Let $\boldsymbol{\mu} \in \mathcal{E}_{\mathrm{Lip}}(T, \tilde{L}_T)$ be a Lipschitz bandit environment with $\tilde{L}_T$ unknown significant shifts $\{\tau_i(\boldsymbol{\mu})\}_{i=0}^{\tilde{L}_T}$. There exists a numerical constant $\bar{c}_1 > 0$ such that the expected dynamic regret of `MDBE` satisfies*

$$\mathbb{E}_{\boldsymbol{\mu}}\left[R(\pi_{\mathtt{MDBE}}, T)\right] \le \bar{c}_1 \log^2(T) \sum_{i=1}^{\tilde{L}_T} \left(\tau_i(\boldsymbol{\mu}) - \tau_{i-1}(\boldsymbol{\mu})\right)^{2/3}.$$

*In particular, this yields the worst-case upper bound*

$$\mathbb{E}_{\boldsymbol{\mu}}\left[R(\pi_{\mathtt{MDBE}}, T)\right] \le \tilde{O}(\tilde{L}_T^{1/3} T^{2/3}).$$

*Therefore, `MDBE` is minimax optimal for the number significant shifts $\tilde{L}_T$ up to poly-log factors.*

Since $\tilde{L}_T \le S_T \le L_T$, Theorem 2 achieves minimax optimal rates of $\tilde{\mathcal{O}}(S_T^{1/3} T^{2/3})$ and $\tilde{\mathcal{O}}(L_T^{1/3} T^{2/3})$, without requiring prior knowledge of $S_T$ or $L_T$, respectively. The bound provided by Theorem 2 can be significantly sharper when $\tilde{L}_T \ll S_T$. The following result further shows that Theorem 2 also recovers the optimal rate in terms of the total variation $V_T$. The proof is deferred to Appendix D.

**Remark 1** (Beyond 1-Lipschitz bandits). *Minimax optimality of Theorem 2 extends beyond Lipschitz bandits with Lipschitz constant 1. In particular, for $(\kappa, \beta)$-Hölder bandits we can show that the minimax optimal rate (up to polylogarithmic factors) can be adapted as $\mathbb{E}\left[R(\pi_{\mathtt{MBDE}}, T)\right] \le \widetilde{\mathcal{O}}\left(T^{\frac{1+\beta}{1+2\beta}} \tilde{L}_T^{\frac{\beta}{1+2\beta}} \kappa^{\frac{1}{1+2\beta}}\right)$, where the minimax optimality is with respect to $T$, $\tilde{L}_T$ and Hölder constants $\kappa$ and $\beta$) jointly. The necessary modifications to our setting and algorithm are detailed in Appendix G.*

**Corollary 1** (**Regret bound in terms of total variation $V_T$**). *Let $\boldsymbol{\mu} \in \mathcal{E}_{\mathrm{Lip}}(T, V_T)$ be any Lipschitz bandit environment with total variation $V_T$. There exists a numerical constant $\bar{c}_2 > 0$ such that the expected dynamic regret of `MDBE` satisfies:*

$$\mathbb{E}_{\boldsymbol{\mu}}\left[R(\pi_{\mathtt{MDBE}}, T)\right] \le \bar{c}_2 \log^2(T) \left(T^{2/3} + V_T^{1/4} T^{3/4}\right).$$

*Therefore, `MDBE` is minimax optimal for the total variation $V_T$ up to poly-log factors.*

## 5 Proof intuition of Theorem 2

A natural question is whether one could directly apply the `MASTER` procedure of Wei and Luo [2021] using a discretization of the arm space as the base algorithm. However, such a direct application would not be possible in our setting, for the following reasons. The `MASTER` procedure is specifically designed to detect and adapt to changes in the *mean reward functions* $\mu_t$, whereas our goal is to handle *significant shifts*, defined in terms of *cumulative reward gaps* $\sum_t \delta_t$. This distinction is crucial: the quantity $\sum_t \delta_t$ is inherently more stable than the instantaneous mean rewards, as it captures *suboptimality* over time, even when the underlying reward functions change smoothly or gradually. As a result, the detection problem in our setting is more subtle and requires finer control over the accumulation of regret. Therefore, even when discretizing the arm space with replays at different depths, the analysis of Wei and Luo [2021] would not achieve the desired dependence on $\tilde{L}_T$.

We further emphasize that our analysis goes beyond a straightforward adaptation of the techniques developed for the $K$-armed setting by Suk and Kpotufe [2022]. In their setting, the algorithm maintains a set of *safe arms* and applies a variant of *Successive Elimination* [Even-Dar et al., 2006] by uniformly sampling from the current set and progressively eliminating arms identified as suboptimal. Roughly speaking, to detect changes of magnitude $\Delta$, they schedule replays of duration $d = K/\Delta^2$, during which all $K$ arms are re-sampled and their estimates updated. However, such a replay scheme is infeasible in our infinite-arm setting. If we were to discretize the arm space into $2^m$ bins at each replay, reliably detecting a shift of magnitude $\Delta$ would require roughly $2^m/\Delta^2$ rounds, resulting in a prohibitive regret cost. To circumvent this issue, our algorithm schedules replays at different discretization scales. By leveraging the hierarchical structure of the dyadic tree $\mathcal{T}$, we update estimates of the gaps at multiple depths *simultaneously*. We exploit the key observation that each sample $x_t$ provides information not only for its bin at depth $d$, but also for all its child bins at depths $d' \ge d$. Through systematic aggregation of information across these nested bins, our multi-scale estimation strategy allows the algorithm to detect significant shifts efficiently at different resolutions, while avoiding the prohibitive sample complexity associated with naive discretization.

We begin by highlighting the technical challenges that arise when estimating these cumulative gaps, in particular how they differ from estimating changes in the mean reward itself (Section 5.1). We then present the high-level ideas for bounding the dynamic regret in two scenarios: (*i*) when the process is in a *stable block*, and no significant shift is detected, we show that the replays do not increases the regret excessively (Section 5.2); and (*ii*) when a *significant shift* does occur, we show that the algorithm responds appropriately (Section 5.3). The proofs are provided in Appendix D.

## 5.1 Estimating the reward gaps

The continuous arm space setting presents key technical challenges. These difficulties are specific to our framework and are not addressed in existing work on non-stationary bandits, and we believe these challenges are of independent interest and may be relevant beyond our specific setting.

The main challenge in this setting lies in estimating a continuous, non-stationary mean reward function without prior knowledge of the magnitude of distributional shifts, or, equivalently, the discretization level and time scale at which such shifts become detectable. Coarse discretization is essential to rapidly detect *large* shifts, while finer discretization is required to capture *smaller yet statistically significant changes*. This motivates the multi-scale replay framework of MDBE in which each scale contributes adaptively based on the magnitude of the underlying shift.

**Bias in mean reward estimation.** Our algorithm actively prunes bins at different depths, meaning that for a given bin $B \in \mathcal{T}_d$, its sub-bins $B' \subset B$ at deeper depths $d' > d$ may be evicted. As a result, the empirical mean reward $\hat{\mu}_t(B)$ can be biased, since arms $x_t$ are only drawn from non-evicted sub-bins. Leveraging Assumption 1, we control this bias, which decays as

$$\forall t \in [\![T]\!], \ \forall d \in \mathcal{D}_t, \ \forall B, B' \in \mathcal{B}_t(d), \quad \left| \mathbb{E}_{t-1}\left[ \hat{\delta}_t(B', B) \right] - \delta_t(B', B) \right| \leq \frac{4}{2^d}. \tag{3}$$

Thus, the bias in estimating the gap $\sum_{t=s_1}^{s_2} \hat{\delta}_t(B', B)$ of bins at depth $d$ scales as $(s_2 - s_1)/2^d$.

**Concentration of mean reward estimates.** Due to the martingale property of $(\hat{\delta}_t(B', B))_t$, we can tightly control its deviation from its conditional expectation. Recall that our *hierarchical sampling scheme* (Algorithm 2) ensures that the algorithm selects an active bin at the minimum active depth $d_0(t)$ and recursively samples from child bins at finer active depths. This choice guarantees that

$$\forall t \in [\![T]\!], \ \forall d \in \mathcal{D}_t, \ \forall B \in \mathcal{B}_t(d), \quad \mathbb{P}(x_t \in B' \,|\, \mathcal{F}_{t-1}) \geq \frac{1}{2^d}.$$

This ensures that all active bins at any depth $d$ are sampled with at least uniform probability, allowing us to control the variance of the inverse weighted estimates at all scales and to derive the following high-probability bound.

**Proposition 2** (**Concentration event**). *Let $\mathcal{E}_1$ be the following event: for all intervals $[s_1, s_2]$, depths $d \in \bigcap_{t=s_1}^{s_2} \mathcal{D}_t$, and bins $B, B' \in \mathcal{B}_{[s_1, s_2]}(d)$, we have*

$$\left| \sum_{t=s_1}^{s_2} \hat{\delta}_t(B', B) - \sum_{t=s_1}^{s_2} \mathbb{E}_{t-1}\left[ \hat{\delta}_t(B', B) \right] \right| \leq c_1 \log(T) \sqrt{(s_2 - s_1)2^d \vee 4^d}, \tag{4}$$

*where $c_1$ is a positive numerical constant. Then $\mathcal{E}_1$ holds with probability at least $1 - 1/T^3$.*

Unlike in classical finite-armed bandits [Suk and Kpotufe, 2022, 2023, Suk and Agarwal, 2023], uniform sampling only holds strictly at depth $d_0(t)$. This subtle distinction plays a critical role in the analysis. Thanks to both the bias control (3) and the concentration control (4), we can ensure that bins are only evicted when truly unsafe, despite the complexity introduced by the hierarchical sampling.

## 5.2 Regret within a block without significant shift

Assume, for the sake of clarity, that no significant shift occurs during the block $[\tau_{l,m}, \tau_{l,m+1}[$. When only the replay at depth $m$ is active, *i.e.* $\mathcal{D}_t = \{m\}$, the algorithm effectively runs a *Successive Elimination* strategy at depth $m$: at each round, it selects a bin uniformly at random from the set of safe bins $\mathcal{B}_{\mathrm{MASTER}}(m)$, samples an arm $x_t$ from this bin, and eliminates bins that are deemed *unsafe* at this depth. In this regime, since the discretization level at depth $m$ is tuned to the block length $\tau_{l,m+1} - \tau_{l,m}$, classical results indicate that the regret incurred within the block is of order $(8^m)^{2/3} = 4^m$ (up to logarithmic factors).

It remains to control the regret from replays across different scales. A key difficulty is that such replays may overlap, making it delicate to control the cumulative regret contribution at each depth simultaneously. A crucial property leveraged in the analysis is that our importance-weighted estimator (2) allows each sampled arm $x_t$ to update the mean estimates $\hat{\mu}_t(B)$ for *all* bins $B$ containing $x_t$. This multi-scale update property ensures that replays at any given depth $d$ contribute information *not only at that level* but also propagate to finer depths $d' > d$. In particular, we leverage Proposition 1, which guarantees that if a bin is identified as unsafe at *any* depth $d$, then *all* arms it contains are suboptimal.

Our proof strategy is to quantify the contribution of each replay at depth $d$ *independently*. Since each such replay lasts at most $8^d$ rounds, and arms are always selected from the set of safe bins $\mathcal{B}_t(d)$ at that depth (otherwise they would have been evicted under the concentration event $\mathcal{E}_1$), the regret contribution of each replay can be upper bounded by $(8^d)^{2/3} = 4^d$. Thanks to the well-calibrated replay scheduling probability (1), there are roughly $\sqrt{8^{m-d}}$ replays at depth $d$ that are scheduled during a block, so overall we can show that the aggregate regret from all replays across depths $d < m$ remains bounded by $4^m$, thus matching the regret incurred by the main replay at depth $m$.

### 5.3 Detecting significant shifts

When a significant shift $\tau_i$ occurs within a block $[\tau_{l,m}, \tau_{l,m+1}[$, the primary challenge is that the shift is initially *undetected*, potentially causing the algorithm to suffer large regret if it continues exploiting outdated information. In particular, such regret can accumulate if a bin that was previously evicted (due to being deemed unsafe) becomes optimal after the shift, while the algorithm keeps ignoring it. More precisely, detecting a shift of magnitude $\Delta$ at depth $d$ requires a replay of length approximately $2^d/\Delta^2$. Undetected shifts of this magnitude are tolerable at this resolution over a period of duration $D$ as long as $\Delta D \leq 4^d$. By carefully calibrating the probability of scheduling replays at each depth (see (1)), we guarantee that *replays at depth $d$ occur frequently enough* to ensure, with high probability, the timely detection of shifts of size $\Delta$ before their contribution to regret becomes significant.

## 6 Conclusion

We studied the previously unexplored problem of non-stationary Lipschitz bandits. We introduced MDBE, a novel algorithm that leverages a hierarchical discretization of the action space to exploit the Lipschitz structure of the evolving reward function. We established a lower bound of $\mathcal{O}(\tilde{L}_T^{1/3} T^{2/3})$ on the expected dynamic regret and showed that MDBE achieves this rate *adaptively*.

While our work is mainly theoretical, we also analyze the computational worst-case complexity of MDBE in Appendix H and show its empirical performance on a synthetic example in Appendix I. Developing computationally efficient algorithms with strong empirical performance that do not require prior knowledge of the non-stationarity [Gerogiannis et al., 2025a,b] remains an open problem. Future work could extend the significant shift detection framework to other structured bandit models such as convex or linear bandits.

## Broader impacts

This work is mainly theoretical and contributes to the analysis of algorithms for sequential-decision making under uncertainty. Our generic setting and algorithms have broad potential use; practitioners will therefore need to specifically address possible social impacts with respect to the relevant application.

## Acknowledgements

The authors gratefully acknowledge Joe Suk for his insightful discussions and for sharing his code, which served as inspiration for our numerical experiments.
N.Nguyen and C.Vernade are funded by the Deutsche Forschungsgemeinschaft (DFG) under both the project 468806714 of the Emmy Noether Programme and under Germany's Excellence Strategy–EXC number 2064/1–Project number 390727645. Both also thank the international Max Planck Research School for Intelligent Systems (IMPRS-IS).

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

# Appendix

## A   Notations

We present in Table 1 a compilation of used notations in our paper.

| Notation | Description |
|---|---|
| $T$ | Time horizon. |
| $L_T$ | Number of arm changes in $T$ rounds: $L_T = \sum_{t=1}^{T-1} \mathbb{1}\{\exists x \in [0,1], \ \mu_{t+1}(x) \neq \mu_t(x)\}$. |
| $S_T$ | Number of best-arm changes in $T$ rounds: $S_T = \sum_{t=1}^{T-1} \mathbb{1}\{\arg\max_x \mu_{t+1}(x) \neq \arg\max_x \mu_t(x)\}$. |
| $V_T$ | Total variation in $T$ rounds: $V_T = \sum_{t=1}^{T-1} \max_{x \in [0,1]} \lvert \mu_{t+1}(x) - \mu_t(x) \rvert$. |
| $\tilde{L}_T$ | Number of significant shifts in $T$ rounds (see Definition 2). |
| $\mathcal{T}$ | Dyadic tree, *i.e.* hierarchical partition of $[0,1]$ (see Definition 3). |
| $\mathcal{T}_d$ | Partition of $[0,1]$ into $2^d$ bins of size $1/2^d$ each. |
| $\mathrm{Parent}(B,d)$ | Unique parent of bin $B$ in $\mathcal{T}_d$. |
| $\mathrm{Children}(B,d)$ | Children (set of bins) of bin $B$ at depth $d$. |
| $[t_l, t_{l+1}[$ | Episode $l$. An episode terminates when a significant shift is detected. |
| $[\tau_{l,m}, \tau_{l,m+1}[$ | Block $m$ within episode $l$; if uninterrupted, its duration is $\tau_{l,m+1} - \tau_{l,m} = 8^m$. |
| $M_l$ | Maximum number of blocks during one episode: $M_l = \lceil \log_8(7(t_{l+1} - t_l) + 1) - 1 \rceil$. |
| $\mathcal{D}_t$ | Set of active depths at round $t$. |
| $d_0(t)$ | Minimum active depth at round $t$: $d_0(t) = \min \mathcal{D}_t$. |
| $\mathcal{B}_t(d)$ | Set of active bins at depth $d$ at time $t$. |
| $\mathcal{B}_{[s_1,s_2]}(d)$ | Bins at depth $d$ active at all time in $[s_1, s_2]$. |
| $\mathcal{B}_{\mathrm{MASTER}}(m)$ | MASTER set of block $m$: $\mathcal{B}_{\mathrm{MASTER}}(m) \subseteq \mathcal{T}_m$. |
| $R_{t,d}$ | Replay trigger variable: $R_{t,d} = 1$ indicates a replay at depth $d$ starts at time $t$. |
| $\textsc{Replay}(t,d)$ | Replay initiated at round $t$ at depth $d$ (see Definition 6). |
| $\{B_d^{(1)}, \ldots, B_d^{(2^d)}\}$ | Ordered bins in $\mathcal{T}_d$ (see Definition 5). |
| $\mathcal{E}_1$ | Concentration event (see Proposition 2). |
| $\mathcal{E}_2(\tau_{l,m})$ | Event defined in Proposition 10. |
| $x_t$ | Arm selected at round $t$ by the algorithm. |
| $x_i^\sharp$ | Last safe arm in phase $i$ (see Definition 7). |
| $x_t^\sharp$ | $x_t^\sharp = x_i^\sharp$ for $t \in [\tau_i, \tau_{i+1}[$. |
| $\delta_t(x)$ | Instantaneous regret at round $t$ of arm $x$: $\delta_t(x) = \max_{x \in [0,1]} \mu_t(x) - \mu_t(x)$. |
| $B_{t,d}$ | Bin selected at depth $d$: $x_t \in B_{t,d}$. |
| $B_{i,d}^\sharp$ | Bin at depth $d$ containing $x_i^\sharp$ (see Definition 8). |
| $B_{t,d}^\sharp$ | $B_{t,d}^\sharp = B_{i,d}^\sharp$ for $t \in [\tau_i, \tau_{i+1}[$. |
| $B_{l,m}^{\mathrm{last}}$ | Last bin of block $[\tau_{l,m}, \tau_{l,m+1}[$ in $\mathcal{B}_{\mathrm{MASTER}}(m)$ (see Definition 10). |
| $\delta_t(B)$ | Instantaneous regret at round $t$ of bin $B$: if $B \in \mathcal{T}_d$, $\delta_t(B) = \max_{B' \in \mathcal{T}_d} \mu_t(B') - \mu_t(B)$. |
| $[s_{i,j}^m(B), s_{i,j+1}^m(B)]$ | $j^{\mathrm{th}}$ bad segment of bin $B$ (see Definition 12). |
| $\tilde{s}_{i,j}^m(B)$ | Approximate midpoint of bad segment $[s_{i,j}^m(B), s_{i,j+1}^m(B)]$ (see Definition 12). |
| $s_{l,m}(B)$ | Bad round of bin $B$ (see Definition 13). |
| $M(s,d,B)$ | Last round where bin $B$ is active during $\textsc{Replay}(s,d)$ (see Definition 11). |
| $\mathcal{K}(\mathbb{P}, \mathbb{Q})$ | Kullback-Leibler divergence between two distributions $\mathbb{P}$ and $\mathbb{Q}$. |
| $kl(p,q)$ | Kullback-Leibler divergence between two Bernoulli with parameters $p$ and $q$. |

Table 1: Summary of notations.

# B  Full pseudo-code of `MDBE`

**Algorithm: `MDBE`: Multi-Depth Bin Elimination**

---

**Input.** horizon $T$, Lipschitz constant.

**Init.** $l \leftarrow 1, t \leftarrow 1$;                          // Global initialization

**while** $t \leq T$ **do**

   (★) $m \leftarrow 1, \tau_{l,m} \leftarrow t$;                          // Init first block of episode $l$

   **if** $t = \tau_{l,m} + 8^m$ **then**                          // Check if current block has ended

      $m \leftarrow m + 1$ ;                          // Change block

      $\mathcal{B}_{\mathrm{MASTER}}(m) \leftarrow \mathcal{T}_m$;          // Reset MASTER set to full bin set at depth $m$

      $\mathcal{D}_t \leftarrow \{m\}$;                          // Activate only the current depth $m$

      $\tau_{l,m} \leftarrow t$;                          // Update block start time

      $\mathrm{StoreActive}[m][t_{\mathrm{start}}] \leftarrow \tau_{l,m}$;          // Record start time of this replay

      **for** $s = \tau_{l,m} + 1, \ldots, \tau_{l,m} + 8^m$ **do**          // Schedule replays in this block

         **for** $d = 0, \ldots, m-1$ **do**

            **if** $s - \tau_{l,m} \equiv 0[8^d]$ **then**

               Sample $R_{s,d} \sim \mathrm{Ber}\big(\sqrt{8^d/(s - \tau_{l,m})}\big)$;

            **else**

               $R_{s,d} = 0$;                          // No replay triggered at depth $d$ in round $s$

      **for** $d = 0, \ldots, m-1$ **do**

         $\mathrm{StoreActive}[d][t_{\mathrm{start}}] \leftarrow \emptyset, \mathrm{StoreActive}[d][t_{\mathrm{end}}] \leftarrow \emptyset$;   // Initialize variable

         $\mathcal{B}_t(d) \leftarrow \emptyset$ ;                          // Initialize set of active bins

   **for** $d = 0, \ldots, m-1$ **do**     // Check which replays are triggered in this round

      **if** $R_{s,d} = 1$ **then**                          // If a replay at depth $d$ is triggered

         $\mathcal{D}_t \leftarrow \mathcal{D}_t \cup \{d\}$;          // Add this depth to current active depths

         $\mathrm{StoreActive}[d][t_{\mathrm{start}}] \leftarrow t$;                          // Store replay start time

         $\mathrm{StoreActive}[d][t_{\mathrm{end}}] \leftarrow t + 8^d$;                          // Store replay end time

         $\mathcal{B}_t(d) \leftarrow \mathcal{T}_d$;          // Activate all bins at this new active depth

   **for** $d \in \mathcal{D}_t$ **do**                          // Check if any active replays are ending

      **if** $\mathrm{StoreActive}[d][t_{\mathrm{end}}] = t$ **then**

         $\mathcal{D}_t \leftarrow \mathcal{D}_t \setminus \{d\}$;                          // Deactivate this depth

         $\mathcal{B}_t(d) \leftarrow \emptyset$;                          // Deactivate its bins

   $d_0(t) \leftarrow \min \mathcal{D}_t$;                          // Identify the minimum active depth

   $B_{\mathrm{parent}} \sim \mathcal{U}(\mathcal{B}_t(d_0(t)))$;          // Sample a bin from active bins at depth $d_0(t)$

   **for** $d \in \mathrm{Sort}(\mathcal{D}_t \setminus \{d_0(t)\})$ **do**   // Go trough the active depths, from shallowest
   to deepest

      **if** $\mathrm{Children}(B_{\mathrm{parent}}, d) \cap \mathcal{B}_t(d) = \emptyset$ **then**   // No active children at this depth

         $x_t \sim \mathcal{U}(B_{\mathrm{parent}})$;                          // Sample arm uniformly from current bin

      **else**

         $B_{\mathrm{child}} \sim \mathcal{U}(\mathrm{Children}(B_{\mathrm{parent}}, d) \cap \mathcal{B}_t(d))$ ;          // Sample active child bin

         $B_{\mathrm{parent}} \leftarrow B_{\mathrm{child}}$;                          // Move to child bin

   **for** $d \in \mathcal{D}_t$ **do**

      **for** $B \in \mathcal{B}_t(d)$ **do**

         **if** $\exists [s_1, s_2] \subseteq [\mathrm{StoreActive[d][t_{\mathrm{start}}]}, t]$ such that (⋆) holds for some active bin $B' \in$
         $\mathcal{B}_t(d)$ **then**

            **for** $d' \in [\![d, m]\!] \cap \mathcal{D}_t$ **do**

               $\mathcal{B}_t(d') \leftarrow \mathcal{B}_t(d') \setminus \bigcup_{B' \in \mathcal{T}_{d'} : B' \subseteq B} \{B'\}$ ;// Evict bins at active depths

   $\mathcal{B}_{\mathrm{MASTER}}(m) \leftarrow \mathcal{B}_{\mathrm{MASTER}}(m) \cap \mathcal{B}_t(m)$;                          // Update MASTER set

   **if** $\mathcal{B}_{\mathrm{MASTER}}(m) = \emptyset$ **then**                          // A significant shift is detected

      $\tau_{l,m+1} \leftarrow t + 1$;

      $l \leftarrow l + 1$ and Restart from (★);                          // Change episode

   $t \leftarrow t + 1$;

---

## C Proof of Proposition 1

Let $d$ be any fixed depth, and let $B \in \mathcal{T}_d$ be a bin that incurs significant regret over an interval $[s_1, s_2]$. Let $x \in B$ be any action within this bin. For each round $t$, define $B_{t,d}^* = \text{argmax}_{B' \in \mathcal{T}_d} \mu_t(B')$ as the best bin at depth $d$ at round $t$, and let $x_{t,d}^* \in B_{t,d}^*$ be any action contained in it.

To relate the regret of bin $B$ with respect to $B_{t,d}^*$ to the regret of any action $x \in B$ with respect to $x_{t,d}^*$, we discretize the bin-level regret at depth $d$ as follows:

$$\delta_t(B_{t,d}^*, B) = \frac{1}{|B_{t,d}^*|} \int_{u \in B_{t,d}^*} \mu_t(u) \mathrm{d}u - \frac{1}{|B|} \int_{u \in B} \mu_t(u) \mathrm{d}u$$

$$= \frac{1}{|B_{t,d}^*|} \int_{u \in B_{t,d}^*} (\mu_t(u) - \mu_t(x_{t,d}^*)) \mathrm{d}u + \mu_t(x_{t,d}^*)$$

$$- \frac{1}{|B|} \int_{u \in B} (\mu_t(u) - \mu_t(x)) \mathrm{d}u - \mu_t(x)$$

$$\leq \frac{1}{|B_{t,d}^*|} \int_{u \in B_{t,d}^*} |\mu_t(u) - \mu_t(x_{t,d}^*)| \mathrm{d}u + \mu_t(x_{t,d}^*)$$

$$+ \frac{1}{|B|} \int_{u \in B} |\mu_t(u) - \mu_t(x)| \mathrm{d}u - \mu_t(x)$$

$$\leq \frac{2}{2^d} + \delta_t(x_{t,d}^*, x) \quad \text{by Assumption 1}.$$

Summing over the interval $[s_1, s_2]$, we obtain:

$$\sum_{t=s_1}^{s_2} \delta_t(B) = \sum_{t=s_1}^{s_2} \delta_t(B_{t,d}^*, B) \leq \sum_{t=s_1}^{s_2} \delta_t(x_{t,d}^*, x) + \frac{2}{2^d}(s_2 - s_1). \tag{5}$$

By assumption, bin $B$ incurs significant regret over $[s_1, s_2]$; that is, from Definition 4,

$$\sum_{t=s_1}^{s_2} \delta_t(B) \geq 3 \log(T) \sqrt{(s_2 - s_1)2^d}. \tag{6}$$

Substituting (6) into (5), we get a lower bound on the cumulative regret of action $x$ with respect to $x_{d,t}^*$:

$$\sum_{t=s_1}^{s_2} \delta_t(x_{t,d}^*, x) \geq 3 \log(T) \sqrt{(s_2 - s_1)2^d} - \frac{2}{2^d}(s_2 - s_1)$$

$$= (s_2 - s_1)^{2/3} \left( 3 \log(T) \sqrt{\frac{2^d}{(s_2 - s_1)^{1/3}}} - 2 \frac{(s_2 - s_1)^{1/3}}{2^d} \right)$$

$$\geq (s_2 - s_1)^{2/3} \left( 3 \log(T) \sqrt{\frac{2^d}{(s_2 - s_1)^{1/3}}} - 2 \log(T) \frac{(s_2 - s_1)^{1/3}}{2^d} \right),$$

for horizon $T$ satisfying $\log(T) \geq 1$.

Now, since $s_2 - s_1 \leq 8^d$, it follows that

$$\frac{2^d}{(s_2 - s_1)^{1/3}} > 1,$$

which implies the above expression is at least:

$$\log(T)(s_2 - s_1)^{2/3}.$$

Therefore

$$\sum_{t=s_1}^{s_2} \delta_t(B) \geq 3 \log(T) \sqrt{(s_2 - s_1)2^d} \implies \log(T)(s_2 - s_1)^{2/3} \leq \sum_{t=s_1}^{s_2} \delta_t(x_{t,d}^*, x) \leq \sum_{t=s_1}^{s_2} \delta_t(x),$$

*i.e.* action $x$ incurs significant regret on interval $[s_1, s_2]$, as per Definition 1.

# D   Proof of the upper bound of `MDBE` (Theorem 2)

Before giving the proof, we introduce a few definitions and notations that will be useful. A full summary of notations used in the proof can be found in Appendix A.

**Definition 5** (**Bin ordering**). *For each depth $d \in \mathbb{N}$, we partition the bins of $\mathcal{T}_d$ by ordering them as $\mathcal{T}_d = \{B_d^{(1)}, \ldots, B_d^{(2^d)}\}$.*

**Definition 6** (`Replay`$(s, d)$). *`Replay`$(s, d)$ denotes a replay at depth $d$ starting at round $s$.*

**Remark 2** (**Eviction time**). *A bin $B \in \mathcal{T}_d$ is said to be evicted at round $t$ if it is evicted at the* end *of round $t$, in the sense that $B \in \mathcal{B}_t(d)$ and $B \notin \mathcal{B}_{t+1}(d)$.*

## D.1   Bias of the mean estimates

One challenge arising from our discretization scheme is that the importance-weighted estimate defined in (2) is inherently biased. This bias comes from the fact that our algorithm selectively samples arms $x_t$ only from active bins, while evicting bins at different depths over time.

**Proposition 3** (**Sampling bias**). *For all round $t$, any active depth $d \in \mathcal{D}_t$, and any active bin $B \in \mathcal{B}_t(d)$ at this depth, the bias in the mean estimate for bin $B$ is bounded as*

$$\left| \mathbb{E}_{t-1}\left[ \hat{\mu}_t(B) \right] - \mu_t(B) \right| \leq \frac{2}{2^d} \,.$$

*Moreover, the cumulative bias in the gap estimates between any pair of **active** bins at depth $d$ over an interval is bounded as:*

$$\forall [s_1, s_2], \forall d \in \bigcap_{t=s_1}^{s_2} \mathcal{D}_t, \forall B, B' \in \mathcal{B}_{[s_1, s_2]}, \quad \left| \sum_{t=s_1}^{s_2} \mathbb{E}_{t-1}\left[ \hat{\delta}_t(B', B) \right] - \sum_{t=s_1}^{s_2} \delta_t(B', B) \right| \leq \frac{4(s_2 - s_1)}{2^d} \,.$$

*Proof of Proposition 3.* We begin with the first claim. Let $d \in \mathcal{D}_t$ be an active depth, and let $B \in \mathcal{B}_t(d)$ be an active bin at this depth. For all round $t$, by the definition of the estimator, we have for all $B \in \mathcal{T}_d$,

$$\hat{\mu}_t(B) = \frac{Y_t(x_t)}{\mathbb{P}(x_t \in B \mid \mathcal{F}_{t-1})} \mathbb{1}\{x_t \in B\} \,.$$

Taking the conditional expectation with respect to $\mathcal{F}_{t-1}$ yields:

$$\mathbb{E}_{t-1}\left[ \hat{\mu}_t(B) \right] = \int_{u \in B} \mu_t(u) f_{t-1}(u) \, \mathrm{d}u \,,$$

where $f_{t-1}$ denotes the density of $x_t$ given $\mathcal{F}_{t-1}$ and $x_t \in B$.

From Assumption 1, for all bin $B \in \mathcal{T}_d$ and any $x \in B$, we have:

$$\left| \mu_t(B) - \mu_t(x) \right| = \left| \frac{1}{|B|} \int_{u \in B} \mu_t(u) - \mu_t(x) \, \mathrm{d}u \right| \leq \frac{1}{|B|} \int_{u \in B} |\mu_t(u) - \mu_t(x)| \, \mathrm{d}u \leq |B| = \frac{1}{2^d} \,.$$

Using this, we can bound the estimation bias:

$$\begin{aligned}
\left| \mathbb{E}_{t-1}\left[ \hat{\mu}_t(B) \right] - \mu_t(B) \right| &= \left| \int_{u \in B} \mu_t(u) f_{t-1}(u) \, \mathrm{d}u - \mu_t(B) \right| \\
&\leq \int_{u \in B} |\mu_t(u) - \mu_t(B)| f_{t-1}(u) \, \mathrm{d}u \\
&\leq \frac{1}{2^d} \int_{u \in B} f_{t-1}(u) \, \mathrm{d}u = \frac{1}{2^d} \,.
\end{aligned}$$

For the second statement, let $B'$ also belong to $\mathcal{B}_t(d)$. Applying the triangle inequality and the previous bound, we can control the bias in the gap estimate:

$$\begin{aligned}
\left| \mathbb{E}_{t-1}\left[ \hat{\delta}_t(B', B) \right] - \delta_t(B', B) \right| &= \left| \mathbb{E}_{t-1}\left[ \mu_t(B') \right] - \mu_t(B') - \mathbb{E}_{t-1}\left[ \mu_t(B) \right] + \mu_t(B) \right| \\
&\leq \left| \mathbb{E}_{t-1}\left[ \mu_t(B') \right] - \mu_t(B') \right| + \left| \mathbb{E}_{t-1}\left[ \mu_t(B) \right] - \mu_t(B) \right| \\
&\leq 2\frac{1}{2^d} + 2\frac{1}{2^d} = \frac{4}{2^d} \,.
\end{aligned}$$

Summing this inequality over the interval $[s_1, s_2]$ establishes the second claim. $\qquad \square$

## D.2 Concentration of the mean estimates

Our sampling scheme (Algorithm 2) is carefully constructed to ensure that the estimated gap between any two active bins at any active depth concentrates around its conditional expectation, especially by *controlling its cumulative variance*. Our result relies on this following inequality.

**Lemma 1** (Beygelzimer et al. [2011]). *Let $(X_t)_t$ be a real-valued martingale difference sequence adapted to the natural filtration $\mathcal{F}_t = \sigma(X_1, \ldots, X_t)$, such that $\mathbb{E}[X_t \mid \mathcal{F}_{t-1}] = 0$. Suppose that $X_t \leq R$ almost surely and that $\sum_{i=1}^{t} \mathbb{E}_{t-1}[X_i^2] \leq V_t$. Then for all $\delta \in (0, 1)$, with probability at least $1 - \delta$,*

$$\sum_{i=1}^{t} X_i \leq (e - 1)\left(\sqrt{V_t \log\left(\frac{1}{\delta}\right)} + R \log\left(\frac{1}{\delta}\right)\right).$$

**Proposition 2** (**Concentration Event**). *Let $\mathcal{E}_1$ be the following event: for all intervals $[s_1, s_2]$, depths $d \in \bigcap_{t=s_1}^{s_2} \mathcal{D}_t$, and bins $B, B' \in \mathcal{B}_{[s_1, s_2]}(d)$, we have*

$$\left|\sum_{t=s_1}^{s_2} \hat{\delta}_t(B', B) - \sum_{t=s_1}^{s_2} \mathbb{E}_{t-1}\left[\hat{\delta}_t(B', B)\right]\right| \leq c_1 \log(T)\sqrt{(s_2 - s_1)2^d \vee 4^d}, \tag{7}$$

*where $c_1$ is a positive numerical constant. Then $\mathcal{E}_1$ holds with probability at least $1 - 1/T^3$.*

*Proof of Proposition 2.* For all depth $d$ and pair of bins $B, B' \in \mathcal{T}_d$, define the martingale difference sequence $(M_t)_t = \left(\hat{\delta}_t(B', B) - \mathbb{E}_{t-1}\left[\hat{\delta}_t(B', B)\right]\right)_t$, adapted to the natural filtration $(\mathcal{F}_t)_{t \geq 1} = (\sigma(x_t, Y_t(x_t)))_{t \geq 1}$.

**Lower Bound on Sampling Probability.** Let $B_d \in \mathcal{B}_t(d)$ be any active bin at depth $d$ at round $t$. We distinguish two cases:

**Case 1: $d = d_0(t)$.** By construction, arms are sampled uniformly at depth $d_0(t)$, implying

$$\mathbb{P}(x_t \in B_d \mid \mathcal{F}_{t-1}) = \frac{1}{|\mathcal{B}_t(d_0(t))|} \geq \frac{1}{2^d}.$$

**Case 2: $d \neq d_0(t)$.** The replay scheduling ensures that at round $t$, a replay at depth $d$ can only be triggered if $t - \tau_{l,m} \equiv 0[8^d]$. This implies that no replay at shallower depth scheduled before $t$ can still be active at round $t$. Thus, when a replay is scheduled at round $t$ at depth $d$, either $d = d_0(t)$, or all replays at depth $d' < d$ have also been scheduled at time $t$. No bin has been evicted at these depths before round $t$, so all bins at depth $d$ are children of active bins at any depth $d' < d$ such that $d' \in \mathcal{D}_t$. This ensures there exists a path of active bins from any bin $B$ up to depth $d_0(t)$.

Using induction, we now show that

$$\forall d' \in \mathcal{D}_t, \ \mathbb{P}(x_t \in B_{d'} \mid \mathcal{F}_{t-1}) \geq 1/2^{d'}. \tag{8}$$

The base case at $d_0(t)$ is already established. Assuming the claim holds at depth $d$. For a deeper depth $d' > d$ and any $B' \in \mathcal{B}_t(d')$ with $B' \subset B_d$, we have

$$\begin{aligned}
\mathbb{P}(x_t \in B' \mid \mathcal{F}_{t-1}) &= \mathbb{P}(x_t \in B' \mid x_t \in B_d, \mathcal{F}_{t-1})\mathbb{P}(x_t \in B_d \mid \mathcal{F}_{t-1}) \\
&\geq \frac{1}{|\mathcal{B}_t(d') \cap \text{Children}(B_d, d')|}\frac{1}{2^d} \\
&\geq \frac{2^d}{2^{d'}}\frac{1}{2^d} \\
&= \frac{1}{2^{d'}}.
\end{aligned}$$

**Bound on each $M_t$.** Recall the definition of $\hat{\delta}_t(B', B)$,

$$\hat{\delta}_t(B', B) = \frac{Y_t(x_t)}{\mathbb{P}(x_t \in B' \mid \mathcal{F}_{t-1})}\mathbb{1}\{x_t \in B'\} - \frac{Y_t(x_t)}{\mathbb{P}(x_t \in B \mid \mathcal{F}_{t-1})}\mathbb{1}\{x_t \in B\}.$$

Using the lower bound (8) and the fact that rewards lie in $[0, 1]$, we have:

$$\left| \hat{\delta}_t(B', B) - \mathbb{E}_{t-1} \left[ \hat{\delta}_t(B', B) \right] \right| \leq 2^{d+1}.$$

**Bound on the variance.** First, remark that

$$\mathbb{E}_{t-1} \left[ \left( \hat{\delta}_t(B', B) - \mathbb{E}_{t-1} \left[ \hat{\delta}_t(B', B) \right] \right)^2 \right] \leq \mathbb{E}_{t-1} \left[ \hat{\delta}_t^2(B', B) \right].$$

Applying the definition and the lower bound (8), we have

$$\mathbb{E}_{t-1} \left[ \hat{\delta}_t^2(B', B) \right] \leq \frac{2}{(1/2^d)^2} \cdot \frac{1}{2^d} = 2^{d+1},$$

where we use the fact that the probability that $x_t$ belong to $B_d$ or $B_d'$ is upper bounded by $1/2^d$.

**Final concentration bound.** Applying Lemma 1 over any interval $[s_1, s_2]$ for a fixed depth $d$ and for any pair of bins $B, B' \in \mathcal{T}_d$, we get that with probability at least $1 - \delta$,

$$\left| \sum_{t=s_1}^{s_2} \hat{\delta}_t(B', B) - \sum_{t=s_1}^{s_2} \mathbb{E}_{t-1} \left[ \hat{\delta}_t(B', B) \right] \right| \leq (e - 1) \left( \sqrt{(s_2 - s_1) 2^{d+1} \log \left( \frac{1}{\delta} \right)} + 2^{d+1} \log \left( \frac{1}{\delta} \right) \right).$$

Taking an union bound over all possible choices of rounds $s_1, s_2$ (there are $T^2$ choices), all possible choices of depths (there are at most $m$ depths, where $m \leq \log(T)$), and all possible choices of pair of bins $\mathcal{B}$ (there are at most $\sum_{m=1}^{\log_8(T)} (2^m)^2 \leq T$ choices of pairing), we have with probability at least $1 - \delta T^3 \log(T)$,

$$\forall [s_1, s_2], \ \forall d \in \bigcap_{t=s_1}^{s_2} \mathcal{D}_t, \ \forall B', B \in \mathcal{B}_t(d),$$

$$\left| \sum_{t=s_1}^{s_2} \hat{\delta}_t(B', B) - \sum_{t=s_1}^{s_2} \mathbb{E}_{t-1} \left[ \hat{\delta}_t(B', B) \right] \right| \leq (e - 1) \left( \sqrt{(s_2 - s_1) 2^{d+1} \log \left( \frac{1}{\delta} \right)} + 2^{d+1} \log \left( \frac{1}{\delta} \right) \right).$$

Choosing $\delta = 1/T^7$ and $c_1 = 7(e - 1)\sqrt{2}$ concludes the proof. $\qquad \square$

We summarize the bias and concentration control of mean estimates in the following statement.

**Corollary 2 (Bias and Concentration Control).** *On event $\mathcal{E}_1$, for all intervals $[s_1, s_2]$, all depths $d \in \bigcap_{t=s_1}^{s_2} \mathcal{D}_t$, and all $B, B' \in \mathcal{B}_{[s_1,s_2]}(d)$,*

$$\left| \sum_{t=s_1}^{s_2} \hat{\delta}_t(B', B) - \sum_{t=s_1}^{s_2} \delta_t(B', B) \right| \leq \underbrace{c_1 \log(T) \sqrt{(s_2 - s_1) 2^d \vee 4^d}}_{\text{Concentration}} + \underbrace{4(s_2 - s_1)/2^d}_{\text{Bias}}.$$

*Proof of Corollary 2.* This follows immediately by combining the concentration event (Proposition 2) with the bias control result from Proposition 3. $\qquad \square$

### D.3 Properties of the eviction scheme of `MDBE`

We show that, under the concentration event $\mathcal{E}_1$, if a bin is *evicted* by `MDBE`, then all arms within that bin must have incurred significant regret.

**Proposition 4.** *On event $\mathcal{E}_1$, if a bin $B \in \mathcal{T}_d$ is evicted at some round $t \geq \tau_{l,m}$, then for all arm $x \in B$, there exists an interval $[s_1, s_2]$ with $s_1 \geq \tau_{l,m}$ and $s_2 = t$ such that $x$ has incurred significant regret on this interval.*

*Proof of Proposition 4.* Assume $B \in \mathcal{T}_d$ has been evicted at a round $t$. By design of MDBE, there exists an interval $[s_1, s_2]$ with $s_1 \geq \tau_{l,m}$ and $s_2 = t$, such that

$$\exists d' \in \bigcap_{t=s_1}^{s_2} \mathcal{D}_t, \, \exists B_{\text{parent}} \in \mathcal{B}_{[s_1,s_2]}(d') \text{ s.t. } B \subseteq B_{\text{parent}} \text{ and } B_{\text{parent}} \text{ is evicted at round } t \,,$$

which means that there exists a bin $B' \in \mathcal{B}_{[s_1,s_2]}(d')$ such that

$$\sum_{t=s_1}^{s_2} \hat{\delta}_t(B', B_{\text{parent}}) > c_0 \log(T)\sqrt{(s_2 - s_1)2^{d'} \vee 4^{d'}} + \frac{4(s_2 - s_1)}{2^{d'}} \,.$$

On the concentration event $\mathcal{E}_1$, we have by Corollary 2, on interval $[s_1, s_2]$,

$$\sum_{t=s_1}^{s_2} \delta_t(B', B_{\text{parent}}) \geq \sum_{t=s_1}^{s_2} \hat{\delta}_t(B', B_{\text{parent}}) - c_1 \log(T) \left( \sqrt{(s_2 - s_1)2^{d'} \vee 4^{d'}} \right) - \frac{4(s_2 - s_1)}{2^{d'}} \,.$$

and therefore by eviction criteria $(\star)$ we have

$$\sum_{t=s_1}^{s_2} \delta_t(B', B_{\text{parent}}) \geq c_0 \log(T) \left( \sqrt{(s_2 - s_1)2^{d'} \vee 4^{d'}} \right) + \frac{4(s_2 - s_1)}{2^{d'}}$$

$$- c_1 \log(T) \left( \sqrt{(s_2 - s_1)2^{d'} \vee 4^{d'}} \right) - \frac{4(s_2 - s_1)}{2^{d'}}$$

$$\geq 3 \log(T)\sqrt{(s_2 - s_1)2^{d'} \vee 4^{d'}} \,,$$

for $c_0$ satisfying $c_0 \geq 3 + c_1$. Since $\sum_{t=s_1}^{s_2} \delta_t(B', B_{\text{parent}}) \leq \sum_{t=s_1}^{s_2} \delta_t(B_{\text{parent}})$, this implies that $B_{\text{parent}}$ incurs significant regret on $[s_1, s_2]$ (Definition 4). Since $x \in B_{\text{parent}}$, it also implies that $x$ incurs significant regret (Definition 1) by Proposition 1. $\qquad\square$

We now present a complementary result showing that bins which are not evicted must exhibit low relative cumulative regret.

**Proposition 5 (Safe bins have low relative regret).** *On concentration event $\mathcal{E}_1$, $\forall [s_1, s_2]$, $\forall d \in \bigcap_{t=s_1}^{s_2} \mathcal{D}_t$, $\forall B, B' \in \mathcal{B}_{[s_1,s_2]}(d)$, if both $B$ and $B'$ are not evicted during this interval, then*

$$\sum_{t=s_1}^{s_2} \delta_t(B, B') \leq (c_0 + c_1) \log(T)\sqrt{(s_2 - s_1)2^d \vee 4^d} + \frac{8(s_2 - s_1)}{2^d} \,,$$

*where $c_0$ and $c_1$ are the positive numerical constants defined respectively in $(\star)$ and Corollary 2.*

*Proof of Proposition 5.* If $B$ and $B'$ are not evicted on $[s_1, s_2]$, then for all $B'' \in \mathcal{B}_{[s_1,s_2]}(d)$, the eviction condition does not hold, *i.e.* we have

$$\sum_{t=s_1}^{s_2} \hat{\delta}_t(B'', B') \leq c_0 \log(T)\sqrt{(s_2 - s_1)2^d \vee 4^d} + \frac{4(s_2 - s_1)}{2^d} \,.$$

In particular, for $B'' = B$, we get

$$\sum_{t=s_1}^{s_2} \hat{\delta}_t(B, B') \leq c_0 \log(T)\sqrt{(s_2 - s_1)2^d \vee 4^d} + \frac{4(s_2 - s_1)}{2^d} \,.$$

By Corollary 2, the true cumulative gap satisfies

$$\sum_{t=s_1}^{s_2} \delta_t(B, B') \leq \sum_{t=s_1}^{s_2} \hat{\delta}_t(B, B') + c_1 \log(T)\sqrt{(s_2 - s_1)2^d \vee 4^d} + \frac{4(s_2 - s_1)}{2^d}$$

$$\leq (c_0 + c_1) \log(T)\sqrt{(s_2 - s_1)2^d \vee 4^d} + \frac{8(s_2 - s_1)}{2^d} \,.$$

$\qquad\square$

## D.4 Relating episode and significant shifts

We show that, with high probability, a new episode begins only if a significant shift occurs within it. This ensures a correspondence between *episodes* $[t_l, t_{l+1}[$ and *significant phases* $[\tau_i, \tau_{i+1}[$. In particular, each significant phase overlaps with at most two episodes.

**Proposition 6.** *On concentration event $\mathcal{E}_1$, for each episode $[t_l, t_{l+1}[$, there exists at least one significant shift $\tau_i \in [t_l, t_{l+1}[$.*

*Proof of Proposition 6.* Let $[\tau_{l,m}, \tau_{l,m+1}[$ be the block during which episode $[t_l, t_{l+1}[$ terminates. By the design of MDBE, the episode ends whenever $\mathcal{B}_{\text{MASTER}}(m) = \emptyset$ at the end of this block. Thus, by the eviction condition $(\star)$, we have:

$$\forall x \in \mathcal{X}, \ \exists [s_1, s_2] \subseteq [\tau_{l,m}, \tau_{l,m+1}[, \ \exists d \in \bigcap_{s=s_1}^{s_2} \mathcal{D}_s, \ \exists B \in \mathcal{B}_{[s_1,s_2]}(d) \text{ with } x \in B, \ \exists B' \in \mathcal{B}_{[s_1,s_2]}(d) \text{ s.t.}$$

$$\sum_{t=s_1}^{s_2} \hat{\delta}_t(B', B) > c_0 \log(T) \sqrt{(s_2 - s_1)2^d \vee 4^d} + \frac{4(s_2 - s_1)}{2^d}.$$

Since we are on event $\mathcal{E}_1$, by Corollary 2,

$$\sum_{t=s_1}^{s_2} \hat{\delta}_t(B', B) \le \sum_{t=s_1}^{s_2} \delta_t(B', B) + c_1 \log(T) \sqrt{(s_2 - s_1)2^d \vee 4^d} + \frac{4(s_2 - s_1)}{2^d}$$

$$\implies \sum_{t=s_1}^{s_2} \delta_t(B', B) \ge \sum_{t=s_1}^{s_2} \hat{\delta}_t(B', B) - c_1 \log(T) \sqrt{(s_2 - s_1)2^d \vee 4^d} - \frac{4(s_2 - s_1)}{2^d}$$

$$\implies \sum_{t=s_1}^{s_2} \delta_t(B', B(x)) \ge (c_0 - c_1) \log(T) \sqrt{(s_2 - s_1)2^d \vee 4^d} \ge 3 \log(T) \sqrt{(s_2 - s_1)2^d},$$

where we used $B_d(x)$ to denote the bin at depth $d$ containing arm $x$.

This implies that bin $B_d(x)$ incurs significant regret on interval $[s_1, s_2]$ (by Definition 4). Then, by Proposition 1, the arm $x \in B$ also incurs significant regret on $[s_1, s_2]$ (Definition 1).

Thus, every episode contains at least one significant shift. $\qquad\square$

## D.5 Regret decomposition and discretion within one block

We first introduce important definitions used throughout the regret analysis.

**Definition 7 (Last safe arm $x_i^\sharp$).** *For each significant phase $[\tau_i, \tau_{i+1}[$, we define $x_i^\sharp$ as the last safe arm (Definition 1), with ties broken arbitrarily. For all $t \in [\tau_i, \tau_{i+1}[$, we define $x_t^\sharp = x_i^\sharp$.*

**Definition 8 (Last safe bin $B_{i,d}^\sharp$).** *We denote by $B_{i,d}^\sharp \in \mathcal{T}_d$ the unique bin at depth $d$ containing $x_i^\sharp$. For all $t \in [\tau_i, \tau_{i+1}[$, we define $B_{t,d}^\sharp = B_{i,d}^\sharp$.*

**Definition 9 (Bin $B_{t,d}$).** *For each round $t$, let $B_{t,d} \in \mathcal{T}_d$ denote the unique bin at depth $d$ containing the arm $x_t$ played at time $t$: $x_t \in B_{t,d}$.*

Note that $B_{t,d}$ may not belong to the active set $\mathcal{B}_t(d)$.

**Decomposing dynamic regret into relative regrets.** We begin from the definition of the expected dynamic regret:

$$\mathbb{E}\left[R(\pi_{\text{MDBE}}, T)\right] = \sum_{t=1}^{T} \sup_{x \in \mathcal{X}} \mu_t(x) - \mathbb{E}\left[\sum_{t=1}^{T} \mu_t(x_t)\right] = \mathbb{E}\left[\sum_{t=1}^{T} \delta_t(x_t)\right],$$

which decomposes as

$$\mathbb{E}\left[\sum_{t=1}^{T} \delta_t(x_t)\right] = \sum_{t=1}^{T} \delta_t(x_t^\sharp) + \mathbb{E}\left[\sum_{t=1}^{T} \delta_t(x_t^\sharp, x_t)\right],$$

where $x_t^\sharp$ is deterministic. The first term can be bounded using the definition of $x_t^\sharp$:

$$\sum_{t=1}^{T} \delta_t(x_t^\sharp) = \sum_{i=0}^{\tilde{L}_T} \sum_{t=\tau_i}^{\tau_{i+1}} \delta_t(x_i^\sharp) \leq \sum_{i=0}^{\tilde{L}_T} \log(T) \, (\tau_{i+1} - \tau_i)^{2/3} \ .$$

The difficulty lies in upper bounding the second sum $\mathbb{E}\left[\sum_t \delta_t(x_t^\sharp, x_t)\right]$. Without loss of generality, there are $T$ total episodes, and by convention we set $t_l = T + 1$ if only $l - 1$ episodes occur by round $t$. Summing over episodes gives

$$\mathbb{E}\left[\sum_{t=1}^{T} \delta_t(x_t^\sharp, x_t)\right] \leq \sum_{l=1}^{T} \mathbb{E}\left[\sum_{t=t_l}^{t_{l+1}} \delta_t(x_t^\sharp, x_t)\right] \ .$$

We further decompose each episode $[t_l, t_{l+1}[$ into blocks. Let $M_l = \lceil \log_8(7(t_{l+1} - t_l) + 1) - 1 \rceil$ be the maximum number of block within this episode. For notational convenience, we can extend $\tau_{l,m} = t_{l+1} - 1$ for $m > M_l$. Then,

$$\sum_{l=1}^{T} \mathbb{E}\left[\sum_{t=t_l}^{t_{l+1}} \delta_t(x_t^\sharp, x_t)\right] \leq \sum_{l=1}^{T} \mathbb{E}\left[\sum_{m=0}^{M_l} \sum_{t=\tau_{l,m}}^{\tau_{l,m+1}-1} \delta_t(x_t^\sharp, x_t)\right] \ .$$

**Discretization trick within each block.** Within each block $[\tau_{l,m}, \tau_{l,m+1}[$, we relate the regret $\delta_t(x_t^\sharp, x_t)$ to the regret of their corresponding parent bins at depth $m$, introducing a discretization bias of order $1/2^m$:

$$\sum_{t=\tau_{l,m}}^{\tau_{l,m+1}-1} \delta_t(x_t^\sharp, x_t) = \sum_{t=\tau_{l,m}}^{\tau_{l,m+1}-1} \mu_t(x_t^\sharp) - \mu_t(x_t)$$

$$= \sum_{t=\tau_{l,m}}^{\tau_{l,m+1}-1} \mu_t(x_t^\sharp) - \mu_t(B_{t,m}^\sharp) + \sum_{t=\tau_{l,m}}^{\tau_{l,m+1}-1} \mu_t(B_{t,m}^\sharp) - \mu_t(x_t)$$

The first term captures the *cumulative discretization error* over one block $[\tau_{l,m}, \tau_{l,m+1}[$ at depth $m$, and by Assumption 1, it is upper bounded as

$$\sum_{t=\tau_{l,m}}^{\tau_{l,m+1}-1} \mu_t(x_t^\sharp) - \mu_t(B_{t,m}^\sharp) \leq \sum_{t=\tau_{l,m}}^{\tau_{l,m+1}-1} |\mu_t(x_t^\sharp) - \mu_t(B_{t,m}^\sharp)| \leq \frac{1}{2^m}(\tau_{l,m+1} - \tau_{l,m}) \leq 4^m \ .$$

For the second term, by design of our sampling scheme (Algorithm 2), if $B_{t,m}$ is active, then conditionally on $B_{t,m} = B$ we have $x_t \sim \mathcal{U}(B)$. Otherwise, $x_t$ is sampled uniformly from one of its active parent at higher depth (that has no active child), and conditionally on $B_{t,m} = B$ we also have $x_t \sim \mathcal{U}(B)$, and therefore in both cases, $\forall t, \ \mathbb{E}\left[\mu_t(x_t) \mid B_{t,m} = B\right] = \mu_t(B)$. This yields

$$\mathbb{E}\left[\sum_{t=\tau_{l,m}}^{\tau_{l,m+1}-1} \mu_t(B_{t,m}^\sharp) - \mu_t(x_t)\right] = \mathbb{E}\left[\sum_{t=\tau_{l,m}}^{\tau_{l,m+1}-1} \mu_t(B_{t,m}^\sharp) - \mu_t(B_{t,m})\right]$$

$$= \mathbb{E}\left[\sum_{t=\tau_{l,m}}^{\tau_{l,m+1}-1} \delta_t(B_{t,m}^\sharp, B_{t,m})\right] \ .$$

Putting this together,

$$\mathbb{E}\left[\sum_{m=0}^{M_l} \sum_{t=\tau_{l,m}}^{\tau_{l,m+1}-1} \delta_t(x_t^\sharp, x_t)\right] \leq \mathbb{E}\left[\sum_{m=0}^{M_l} 4^m\right] + \mathbb{E}\left[\sum_{m=0}^{M_l} \sum_{t=\tau_{l,m}}^{\tau_{l,m+1}-1} \delta_t(B_{t,m}^\sharp, B_{t,m})\right] \ . \qquad (9)$$

**Upper bounding the bias.** Using the definition of $M_l$, we upper bound the bias over one episode:

$$\mathbb{E}\left[\sum_{m=0}^{M_l} 4^m\right] = \frac{4^{M_l+1}-1}{3} \leq 6\mathbb{E}\left[(t_{l+1}-t_l)^{2/3}\right].$$

Summing over all episodes yields:

$$\sum_{l=1}^{T} \mathbb{E}\left[\sum_{m=0}^{M_l} 4^m\right] \leq 6\sum_{l=1}^{T} \mathbb{E}\left[(t_{l+1}-t_l)^{2/3}\right]. \tag{10}$$

**Summary of regret decomposition.** So far, we have shown that expected dynamic regret is upper bounded as

$$\mathbb{E}\left[R(\pi_{\text{MDBE}},T)\right] \leq \log(T)\sum_{i=0}^{\tilde{L}_T}(\tau_{i+1}-\tau_i)^{2/3} + 6\sum_{l=1}^{T}\mathbb{E}\left[(t_{l+1}-t_l)^{2/3}\right]$$

$$+ \sum_{l=1}^{T}\mathbb{E}\left[\sum_{m=0}^{M_l}\sum_{t=\tau_{l,m}}^{\tau_{l,m+1}-1}\delta_t(B_{t,m}^{\sharp}, B_{t,m})\right]$$

Using the fact that event $\mathcal{E}_1$ holds with probability at least $1-1/T^3$ (Proposition 2), and the fact that rewards are bounded in $[0,1]$, we get

$$\mathbb{E}\left[R(\pi_{\text{MDBE}},T)\right] \leq \log(T)\sum_{i=0}^{\tilde{L}_T}(\tau_{i+1}-\tau_i)^{2/3} + 6\sum_{l=1}^{T}\mathbb{E}\left[\mathbb{1}\{\mathcal{E}_1\}(t_{l+1}-t_l)^{2/3}\right] + 6\frac{T^2}{T^3}$$

$$+ \sum_{l=1}^{T}\mathbb{E}\left[\mathbb{1}\{\mathcal{E}_1\}\sum_{m=0}^{M_l}\sum_{t=\tau_{l,m}}^{\tau_{l,m+1}-1}\delta_t(B_{t,m}^{\sharp}, B_{t,m})\right] + \frac{T^2}{T^3}$$

$$= \log(T)\sum_{i=0}^{\tilde{L}_T}(\tau_{i+1}-\tau_i)^{2/3} + 6\sum_{l=1}^{T}\mathbb{E}\left[\mathbb{1}\{\mathcal{E}_1\}(t_{l+1}-t_l)^{2/3}\right]$$

$$+ \sum_{l=1}^{T}\mathbb{E}\left[\mathbb{1}\{\mathcal{E}_1\}\sum_{m=0}^{M_l}\sum_{t=\tau_{l,m}}^{\tau_{l,m+1}-1}\delta_t(B_{t,m}^{\sharp}, B_{t,m})\right] + \frac{7}{T}. \tag{11}$$

It remains to bound the terms

$$\sum_{l=1}^{T}\mathbb{E}\left[\mathbb{1}\{\mathcal{E}_1\}(t_{l+1}-t_l)^{2/3}\right]$$

and

$$\sum_{l=1}^{T}\mathbb{E}\left[\mathbb{1}\{\mathcal{E}_1\}\sum_{m=0}^{M_l}\sum_{t=\tau_{l,m}}^{\tau_{l,m+1}-1}\delta_t(B_{t,m}^{\sharp}, B_{t,m})\right].$$

We first focus on the latter term. We introduce the following useful definition.

**Definition 10 (Last bin in $\mathcal{B}_{\textbf{MASTER}}(m)$: $B_{l,m}^{\textbf{last}}$).** *We denote by $B_{l,m}^{\text{last}} \in \mathcal{T}_m$ the last bin (with ties broken arbitrarily) at depth $m$ within block $[\tau_{l,m}, \tau_{l,m+1}[$ that belongs to $\mathcal{B}_{\text{MASTER}}(m)$.*

Then we decompose the last sum of (11), and condition on $\mathcal{F}_{\tau_{l,m}}$ using a tower rule,

$$\mathbb{E}\left[\sum_{m=0}^{M_l}\sum_{t=\tau_{l,m}}^{\tau_{l,m+1}-1}\delta_t(B_{t,m}^{\sharp}, B_{t,m})\right]$$

$$= \mathbb{E}\left[\sum_{m=0}^{M_l}\sum_{t=\tau_{l,m}}^{\tau_{l,m+1}-1}\delta_t(B_{l,m}^{\text{last}}, B_{t,m})\right] + \mathbb{E}\left[\sum_{m=0}^{M_l}\sum_{t=\tau_{l,m}}^{\tau_{l,m+1}-1}\delta_t(B_{t,m}^{\sharp}, B_{l,m}^{\text{last}})\right]$$

$$= \mathbb{E}\left[\sum_{m=0}^{M_l}\mathbb{E}\left[\sum_{t=\tau_{l,m}}^{\tau_{l,m+1}-1}\delta_t(B_{l,m}^{\text{last}}, B_{t,m})\,\Big|\,\mathcal{F}_{\tau_{l,m}}\right]\right] + \mathbb{E}\left[\sum_{m=0}^{M_l}\mathbb{E}\left[\sum_{t=\tau_{l,m}}^{\tau_{l,m+1}-1}\delta_t(B_{t,m}^{\sharp}, B_{l,m}^{\text{last}})\,\Big|\,\mathcal{F}_{\tau_{l,m}}\right]\right], \tag{12}$$

where we recall we defined the filtration $(\mathcal{F}_t)_{t\geq 1}$ as $\mathcal{F}_t = \sigma(\{x_s, Y_s(x_s)\}_{s=1}^{t-1})$, with by convention $\mathcal{F}_1$ being the trivial sigma algebra. For the next two subsections, we focus on the conditional expectations

$$(A) = \mathbb{E}\left[\mathbb{1}\{\mathcal{E}_1\} \sum_{t=\tau_{l,m}}^{\tau_{l,m+1}-1} \delta_t(B_{l,m}^{\text{last}}, B_{t,m}) \,\Big|\, \mathcal{F}_{\tau_{l,m}}\right]$$

and

$$(B) = \mathbb{E}\left[\mathbb{1}\{\mathcal{E}_1\} \sum_{t=\tau_{l,m}}^{\tau_{l,m+1}-1} \delta_t(B_{t,m}^{\sharp}, B_{l,m}^{\text{last}}) \,\Big|\, \mathcal{F}_{\tau_{l,m}}\right],$$

and our goal is to show that these terms are upper-bounded almost surely by a term of order of $4^m$.

We conclude this subsection by noting two important observations:

- At the start of each block $\tau_{l,m}$, all previous observations are discarded.
- Replays within block $[\tau_{l,m}, \tau_{l,m+1}[$ are scheduled at $\tau_{l,m}$, *independently of observations prior to that round*. Thus, for $t \in [\tau_{l,m}, \tau_{l,m+1}[$ and $d < m$, $R_{t,d}$ is $\mathcal{F}_{\tau_{l,m}}$-measurable.

### D.6 Upper bounding (A)

The term $\sum_{t=\tau_{l,m}}^{\tau_{l,m+1}-1} \delta_t(B_{l,m}^{\text{last}}, B_{t,m})$ captures the cumulative regret between any bin that is deemed safe by the *algorithm* and the bin $B_{t,m}$ selected by the algorithm at each round $t$. Notably, $B_{t,m}$ is either

- A bin at depth $m$ deemed safe and *retained* in the active set $\mathcal{B}_{\text{MASTER}}(m)$, or
- A bin that has been *evicted* (or whose parent was evicted at a shallower depth $d < m$) and is being re-explored as part of a *replay* initiated at any active depth $d < m$.

Importantly, evicted bins can reappear during replays, but only through replays initiated at depths strictly less than $m$. We aim to show that (A) is upper-bounded almost surely by a quantity of order $4^m$.

**Proposition 7** (**Upper bound of (A)**). *There exists a positive numerical constant $c_A > 0$ such that*

$$(A) \leq c_A \log^2(T) 4^m .$$

Our approach begins by upper bounding (A) through a decomposition at depth $d_0(t)$ and leverage the properties of our sampling scheme (Algorithm 2) . In particular, it leverages the fact that at each round $t$, we choose *uniformly* an active bin at depth $d_0(t)$.

**Lemma 2.** *We have that (A) is upper bounded as*

$$(A) \leq \mathbb{E}\left[\mathbb{1}\{\mathcal{E}_1\} \sum_{t=\tau_{l,m}}^{\tau_{l,m+1}-1} \left(\sum_{B\in\mathcal{B}_t(d_0(t))} \frac{\delta_t\left(\text{Parent}(B_{l,m}^{\text{last}}, d_0(t)), B\right)}{|\mathcal{B}_t(d_0(t))|} + \frac{4}{2^{d_0(t)}}\right) \,\Big|\, \mathcal{F}_{\tau_{l,m}}\right] .$$

*Proof of Lemma 2.* For all $t \in [\tau_{l,m}, \tau_{l,m+1}[$, we first relate the relative regret of $B_{t,m}$ to $B_{l,m}^{\text{last}}$ to the relative regret of their parent at depth $d_0(t)$,

$$\begin{aligned}
\delta_t(B_{l,m}^{\text{last}}, B_{t,m}) &= \mu_t(B_{l,m}^{\text{last}}) - \mu_t(B_{t,m}) \\
&= \mu_t(B_{l,m}^{\text{last}}) - \mu_t\left(\text{Parent}(B_{l,m}^{\text{last}}, d_0(t))\right) \\
&+ \mu_t\left(\text{Parent}(B_{l,m}^{\text{last}}, d_0(t))\right) - \mu_t(\text{Parent}(B_{t,m}, d_0(t))) \\
&+ \mu_t(\text{Parent}(B_{t,m}, d_0(t))) - \mu_t(B_{t,m}) \\
&\leq \left|\mu_t(B_{l,m}^{\text{last}}) - \mu_t\left(\text{Parent}(B_{l,m}^{\text{last}}, d_0(t))\right)\right| \\
&+ \mu_t\left(\text{Parent}(B_{l,m}^{\text{last}}, d_0(t))\right) - \mu_t(\text{Parent}(B_{t,m}, d_0(t))) \\
&+ \left|\mu_t(\text{Parent}(B_{t,m}, d_0(t))) - \mu_t(B_{t,m})\right| \\
&\leq \delta_t\left(\text{Parent}(B_{l,m}^{\text{last}}, d_0(t)), \text{Parent}(B_{t,m}, d_0(t))\right) + \frac{4}{2^{d_0(t)}} ,
\end{aligned}$$

where in the last inequality we used the fact that $B_{t,m} \subseteq \text{Parent}(B_{t,m}, d_0(t))$, $B_{l,m}^{\text{last}} \subseteq \text{Parent}(B_{l,m}^{\text{last}}, d_0(t))$ and Assumption 1. Note that $\text{Parent}(B_{t,m}, d_0(t))$ is necessarily active at depth $d_0(t)$. Summing this inequality over the block $[\tau_{l,m}, \tau_{l,m+1}[$,

$$\sum_{t=\tau_{l,m}}^{\tau_{l,m+1}-1} \delta_t \left(B_{l,m}^{\text{last}}, B_{t,m}\right) \leq \underbrace{\sum_{t=\tau_{l,m}}^{\tau_{l,m+1}-1} \delta_t \left(\text{Parent}(B_{l,m}^{\text{last}}, d_0(t)), \text{Parent}(B_{t,m}, d_0(t))\right)}_{\text{Regret of parents at minimum active depth}}$$

$$+ \underbrace{\sum_{t=\tau_{l,m}}^{\tau_{l,m+1}-1} \frac{4}{2^{d_0(t)}}}_{\text{cumulative bias}} \, .$$

By tower rule, since $\mathcal{F}_{\tau_{l,m}} \subseteq \mathcal{F}_{t-1}$ we have for the first sum, for any bin $B \in \mathcal{T}_m$

$$\mathbb{E}\left[\mathbb{1}\{\mathcal{E}_1\} \sum_{t=\tau_{l,m}}^{\tau_{l,m+1}-1} \delta_t \left(\text{Parent}(B, d_0(t)), \text{Parent}(B_{t,m}, d_0(t))\right) \,\Big|\, \mathcal{F}_{\tau_{l,m}}\right]$$

$$= \mathbb{E}\left[\mathbb{1}\{\mathcal{E}_1\} \sum_{t=\tau_{l,m}}^{\tau_{l,m+1}} \mathbb{E}\left[\delta_t \left(\text{Parent}(B, d_0(t)), \text{Parent}(B_{t,m}, d_0(t))\right) \,\Big|\, \mathcal{F}_{t-1}\right] \,\Big|\, \mathcal{F}_{\tau_{l,m}}\right] \, .$$

Since $\text{Parent}(B_{t,m}, d_0(t))$ is an active bin at depth $d_0(t)$ and since we choose *uniformly* an active bin at depth $d_0(t)$, we can write for any bin $B \in \mathcal{T}_m$,

$$\mathbb{E}\left[\delta_t \left(\text{Parent}(B, d_0(t)), \text{Parent}(B_{t,m}, d_0(t))\right) \,\Big|\, \mathcal{F}_{t-1}\right]$$

$$= \sum_{B \in \mathcal{B}_t(d_0(t))} \delta_t \left(\text{Parent}(B, d_0(t)), B\right) \mathbb{P}\left(\text{Parent}(B_{t,m}, d_0(t)) = B \,\big|\, \mathcal{F}_{t-1}\right)$$

$$= \sum_{B \in \mathcal{B}_t(d_0(t))} \delta_t \left(\text{Parent}(B, d_0(t)), B\right) \frac{1}{|\mathcal{B}_t(d_0(t))|} \, ,$$

and therefore for all $B \in \mathcal{T}_m$,

$$\mathbb{E}\left[\mathbb{1}\{\mathcal{E}_1\} \sum_{t=\tau_{l,m}}^{\tau_{l,m+1}-1} \delta_t \left(\text{Parent}(B, d_0(t)), \text{Parent}(B_{t,m}, d_0(t))\right) \,\Big|\, \mathcal{F}_{\tau_{l,m}}\right]$$

$$= \mathbb{E}\left[\mathbb{1}\{\mathcal{E}_1\} \sum_{t=\tau_{l,m}}^{\tau_{l,m+1}-1} \sum_{B \in \mathcal{B}_t(d_0(t))} \delta_t \left(\text{Parent}(B, d_0(t)), B\right) \frac{1}{|\mathcal{B}_t(d_0(t))|} \,\Big|\, \mathcal{F}_{\tau_{l,m}}\right]$$

Choosing $B = B_{l,m}^{\text{last}} \in \mathcal{T}_m$ in particular, the result above implies

$$\mathbb{E}\left[\mathbb{1}\{\mathcal{E}_1\} \sum_{t=\tau_{l,m}}^{\tau_{l,m+1}-1} \delta_t \left(\text{Parent}(B_{l,m}^{\text{last}}, d_0(t)), \text{Parent}(B_{t,m}, d_0(t))\right) \,\Big|\, \mathcal{F}_{\tau_{l,m}}\right]$$

$$= \mathbb{E}\left[\mathbb{1}\{\mathcal{E}_1\} \sum_{t=\tau_{l,m}}^{\tau_{l,m+1}-1} \sum_{B \in \mathcal{B}_t(d_0(t))} \delta_t \left(\text{Parent}(B_{l,m}^{\text{last}}, d_0(t)), B\right) \frac{1}{|\mathcal{B}_t(d_0(t))|} \,\Big|\, \mathcal{F}_{\tau_{l,m}}\right] \, ,$$

and therefore

$$\mathbb{E}\left[\mathbb{1}\{\mathcal{E}_1\} \sum_{t=\tau_{l,m}}^{\tau_{l,m+1}-1} \delta_t \left(B_{l,m}^{\text{last}}, B_{t,m}\right) \,\Big|\, \mathcal{F}_{\tau_{l,m}}\right]$$

$$\leq \mathbb{E}\left[\mathbb{1}\{\mathcal{E}_1\} \sum_{t=\tau_{l,m}}^{\tau_{l,m+1}-1} \left(\sum_{B \in \mathcal{B}_t(d_0(t))} \delta_t \left(\text{Parent}(B_{l,m}^{\text{last}}, d_0(t)), B\right) \frac{1}{|\mathcal{B}_t(d_0(t))|} + \frac{4}{2^{d_0(t)}}\right) \,\Big|\, \mathcal{F}_{\tau_{l,m}}\right]$$

$$\square$$

We are now ready to prove Proposition 7. The rest of this subsection is dedicated to this purpose.

Thanks to Lemma 2, it now suffices to bound both the parent's regret contribution at depth $d_0(t)$

$$(A.1) = \mathbb{E}\left[\mathbb{1}\{\mathcal{E}_1\} \sum_{t=\tau_{l,m}}^{\tau_{l,m+1}-1} \sum_{B \in \mathcal{B}_t(d_0(t))} \frac{\delta_t\left(\text{Parent}(B_{l,m}^{\text{last}}, d_0(t)), B\right)}{|\mathcal{B}_t(d_0(t))|} \,\Bigg|\, \mathcal{F}_{\tau_{l,m}}\right]$$

and the cumulative bias term

$$(A.2) = \mathbb{E}\left[\mathbb{1}\{\mathcal{E}_1\} \sum_{t=\tau_{l,m}}^{\tau_{l,m+1}-1} \frac{4}{2^{d_0(t)}} \,\Bigg|\, \mathcal{F}_{\tau_{l,m}}\right].$$

**Upper bounding (A.1.).** Considering all possible value of $d_0(t)$, (A.1) can be rewritten using our ordering introduced in Definition 5: denoting $B_d^{(1)}, B_d^{(1)} \ldots, B_d^{(2^d)}$ the bins at depth $d$,

$(A.1)$

$$= \mathbb{E}\left[\mathbb{1}\{\mathcal{E}_1\} \sum_{t=\tau_{l,m}}^{\tau_{l,m+1}-1} \sum_{d=0}^{m} \sum_{B \in \mathcal{B}_t(d)} \mathbb{1}\{d_0(t) = d\} \frac{\delta_t\left(\text{Parent}(B_{l,m}^{\text{last}}, d), B\right)}{|\mathcal{B}_t(d)|} \,\Bigg|\, \mathcal{F}_{\tau_{l,m}}\right]$$

$$= \mathbb{E}\left[\mathbb{1}\{\mathcal{E}_1\} \sum_{t=\tau_{l,m}}^{\tau_{l,m+1}-1} \sum_{d=0}^{m} \sum_{k=1}^{2^d} \mathbb{1}\{d_0(t) = d\} \frac{\delta_t\left(\text{Parent}(B_{l,m}^{\text{last}}, d), B_d^{(k)}\right)}{|\mathcal{B}_t(d)|} \mathbb{1}\left\{B_d^{(k)} \in \mathcal{B}_t(d)\right\} \,\Bigg|\, \mathcal{F}_{\tau_{l,m}}\right]$$

$$= \underbrace{\mathbb{E}\left[\mathbb{1}\{\mathcal{E}_1\} \sum_{t=\tau_{l,m}}^{\tau_{l,m+1}-1} \sum_{k=1}^{2^m} \mathbb{1}\{d_0(t) = m\} \frac{\delta_t\left(B_{l,m}^{\text{last}}, B_m^{(k)}\right)}{|\mathcal{B}_t(m)|} \mathbb{1}\left\{B_m^{(k)} \in \mathcal{B}_t(m)\right\} \,\Bigg|\, \mathcal{F}_{\tau_{l,m}}\right]}_{\text{Regret contribution at depth } m}$$

$$+ \underbrace{\mathbb{E}\left[\mathbb{1}\{\mathcal{E}_1\} \sum_{t=\tau_{l,m}}^{\tau_{l,m+1}-1} \sum_{d=0}^{m-1} \sum_{k=1}^{2^d} \mathbb{1}\{d_0(t) = d\} \frac{\delta_t\left(\text{Parent}(B_{l,m}^{\text{last}}, d), B_d^{(k)}\right)}{|\mathcal{B}_t(d)|} \mathbb{1}\left\{B_d^{(k)} \in \mathcal{B}_t(d)\right\} \,\Bigg|\, \mathcal{F}_{\tau_{l,m}}\right]}_{\text{Regret contribution of replays of depth } d < m}.$$

$$(13)$$

We introduce the *eviction time* of a bin $B$, which will be convenient to quantify the regret contribution of each bin.

**Definition 11** (**Eviction time within a replay $M(s, d, B)$**). *For all bin $B$, $M(s, d, B)$ denotes the last round where bin $B$ is active within a replay at depth $d$ starting at round $s$, $M(s, d, B) \in [s, s+8^d]$. If bin $B$ is not evicted during this replay, we define $M(s, d, B) = s + 8^d$. By abuse of notations, we define $M(\tau_{l,m}, m, B)$ as the last round where bin $B$ is retained in $\mathcal{B}_{\text{MASTER}}(m)$, and define $M(\tau_{l,m}, m, B) = \tau_{l,m} + 8^m - 1 = \tau_{l,m+1} - 1$ if it is not evicted during the block $[\tau_{l,m}, \tau_{l,m+1}[$.*

**Regret contribution at depth $m$.** Since $\mathcal{B}_{\text{MASTER}}(m)$ is initialized at round $t = \tau_{l,m}$, we have for all bins $B_m^{(k)}$, on event $\mathcal{E}_1$,

$$\sum_{t=\tau_{l,m}}^{\tau_{l,m+1}-1} \sum_{k=1}^{2^m} \mathbb{1}\{d_0(t) = m\} \frac{1}{|\mathcal{B}_t(m)|} \delta_t\left(B_{l,m}^{\text{last}}, B_m^{(k)}\right) \mathbb{1}\left\{B_m^{(k)} \in \mathcal{B}_t(m)\right\}$$

$$= \sum_{k=1}^{2^m} \sum_{t=\tau_{l,m}}^{M(\tau_{l,m}, m, B_m^{(k)})} \mathbb{1}\{d_0(t) = m\} \frac{1}{|\mathcal{B}_t(m)|} \delta_t\left(B_{l,m}^{\text{last}}, B_m^{(k)}\right).$$

Without any loss of generality, we assume that $\{B_m^{(1)}, \cdots, B_m^{(2^m)}\}$ is the ordering according to the *eviction time* of these bins in the block $[\tau_{l,m}, \tau_{l,m+1}[$ (otherwise we can always re-index the bins), that is,

$$\tau_{l,m} \leq M\left(\tau_{l,m}, d, B_m^{(1)}\right) \leq M\left(\tau_{l,m}, m, B_m^{(2)}\right) \leq \cdots \leq \underbrace{M\left(\tau_{l,m}, m, B_d^{(2^m)}\right)}_{=B_{l,m}^{\text{last}}} \leq \tau_{l,m+1} - 1.$$

Thanks to this ordering, we can lower bound the active bin set at depth $d$ as

$$\min_{t \in [\tau_{l,m}, M(\tau_{l,m}, m, B_m^{(k)})]} |\mathcal{B}_t(m)| \geq 2^m + 1 - k,$$

and therefore

$$\sum_{k=1}^{2^m} \sum_{t=\tau_{l,m}}^{M(\tau_{l,m}, m, B_m^{(k)})} \mathbb{1}\{d_0(t) = m\} \frac{1}{|\mathcal{B}_t(m)|} \delta_t \left( B_{l,m}^{\text{last}}, B_m^{(k)} \right)$$

$$\leq \sum_{k=1}^{2^m} \frac{1}{2^m + 1 - k} \sum_{t=\tau_{l,m}}^{M(\tau_{l,m}, m, B_m^{(k)})} \mathbb{1}\{d_0(t) = m\} \delta_t \left( B_{l,m}^{\text{last}}, B_m^{(k)} \right).$$

Moreover, since $B_{l,m}^{\text{last}}$ and $B_m^{(k)}$ are by definition not evicted on interval $[\tau_{l,m}, M(\tau_{l,m}, m, B_m^{(k)})]$, we have by Proposition 5 on event $\mathcal{E}_1$

$$\sum_{t=\tau_{l,m}}^{M(\tau_{l,m}, m, B_m^{(k)})} \mathbb{1}\{d_0(t) = m\} \delta_t \left( B_{l,m}^{\text{last}}, B_m^{(k)} \right)$$

$$\leq (c_0 + c_1) \log(T) \sqrt{\left( M(\tau_{l,m}, m, B_m^{(k)}) - \tau_{l,m} \right) 2^m \vee 4^m} + \frac{8 \left( M(\tau_{l,m}, m, B_m^{(k)}) - \tau_{l,m} \right)}{2^m}$$

$$\leq (c_0 + c_1) \log(T) 4^m + 8 \times 4^m$$
$$\leq (c_0 + c_1 + 8) \log(T) 4^m$$
$$\leq c_2 \log(T) 4^m,$$

assuming again horizon $T$ satisfies $\log(T) \geq 1$ and setting $c_2 = c_0 + c_1 + 8$. Therefore, the regret contribution of replay at depth $m$ is upper bounded as

$$\sum_{k=1}^{2^m} \sum_{t=\tau_{l,m}}^{M(\tau_{l,m}, m, B_m^{(k)})} \mathbb{1}\{d_0(t) = m\} \frac{\delta_t \left( B_{l,m}^{\text{last}}, B_m^{(k)} \right)}{|\mathcal{B}_t(m)|} \leq c_2 \log(T) 4^m \sum_{k=1}^{2^m} \frac{1}{2^m + 1 - k}$$

$$\leq (c_0 + c_1 + 8) \log(T) 4^m (\log(2^m) + 1)$$

$$\leq (c_2 + 1) \log^2(T) 4^m,$$

where we used the fact that $m \leq \log(T)$.

Therefore, the regret contribution of the replay at depth $m$ is upper bounded as

$$\mathbb{E}\left[ \mathbb{1}\{\mathcal{E}_1\} \sum_{t=\tau_{l,m}}^{\tau_{l,m+1}-1} \sum_{k=1}^{2^m} \mathbb{1}\{d_0(t) = m\} \frac{\delta_t \left( B_{l,m}^{\text{last}}, B_m^{(k)} \right)}{|\mathcal{B}_t(m)|} \mathbb{1}\left\{ B_m^{(k)} \in \mathcal{B}_t(m) \right\} \,\Big|\, \mathcal{F}_{\tau_{l,m}} \right]$$

$$\leq (c_2 + 1) \log^2(T) 4^m. \tag{14}$$

**Regret contribution of replays at depth $d < m$.** Fix a depth $d \in [\![ 0, m-1 ]\!]$ and a bin $B_d^{(k)}$ at this depth. We consider the set of rounds $t > \tau_{l,m}$ for which $d_0(t) = d$ and $B_d^{(k)} \in \mathcal{B}_t(d)$, and partition this set into contiguous intervals (represented as blue intervals in Figure 2).

Due to the structure of our replay schedule, each such interval must correspond to a single replay at depth $d$, which begins at some round $s$ satisfying $R_{s,d} = 1$. That is, each interval is initiated at the start of a replay and includes only the rounds during which $B_d^{(k)}$ remains active in that replay.

Each interval concludes either when $B_d^{(k)}$ is evicted from the active set $\mathcal{B}_t(d)$, or when the replay itself ends at time $s + 8^d$. Therefore, each interval spans the range $[s, M(s, d, B_d^{(k)})]$, where $M(s, d, B_d^{(k)})$ is the final round during which $B_d^{(k)}$ is active within that specific replay (see Definition 11).

The key idea of our analysis is to treat these replays at depth $d$ *independently*, and quantify the regret incurred due to playing the bin $B_d^{(k)}$ during each such replay window. This allows us to control the regret by bounding its contribution separately within each replay interval.

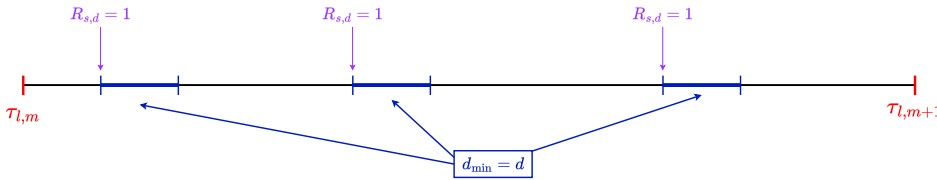

Figure 2: For a given depth $d$, partition of the rounds of a block where $d_0(t) = d$. Any of these blue intervals should be initialized by the start of a replay at this depth, *i.e.* $R_{s,d} = 1$.

Following these remarks, we can upper bound the second sum of Equation (13) on event $\mathcal{E}_1$ as

$$\sum_{t=\tau_{l,m}}^{\tau_{l,m+1}-1} \sum_{d=0}^{m-1} \sum_{k=1}^{2^d} \mathbb{1}\{d_0(t)=d\} \frac{\delta_t\left(\text{Parent}(B_{l,m}^{\text{last}}, d), B_d^{(k)}\right)}{|\mathcal{B}_t(d)|} \mathbb{1}\left\{B_d^{(k)} \in \mathcal{B}_t(d)\right\}$$

$$\leq \sum_{s=\tau_{l,m}}^{\tau_{l,m+1}-1} \sum_{d=0}^{m-1} \sum_{k=1}^{2^d} R_{s,d} \left(\sum_{t=s}^{M(s,d,B_d^{(k)})} \frac{\delta_t\left(\text{Parent}(B_{l,m}^{\text{last}}, d), B_d^{(k)}\right)}{|\mathcal{B}_t(d)|}\right)_+$$

$$\leq \sum_{s=\tau_{l,m}}^{\tau_{l,m+1}-1} \sum_{d=0}^{m-1} \sum_{k=1}^{2^d} R_{s,d} \frac{1}{\min_{t \in [s, M(s,d,B_d^{(k)})]} |\mathcal{B}_t(d)|} \left(\sum_{t=s}^{M(s,d,B_d^{(k)})} \delta_t\left(\text{Parent}(B_{l,m}^{\text{last}}, d), B_d^{(k)}\right)\right)_+.$$

Again, without any loss of generality, we assume that $\{B_d^{(1)}, \cdots, B_d^{(2^d)}\}$ is the ordering corresponding to eviction time of the replay at depth $d$ starting at round $s$,

$$s \leq M\left(s,d,B_d^{(1)}\right) \leq M\left(s,d,B_d^{(2)}\right) \leq \cdots \leq M\left(s,d,B_d^{(2^d)}\right) \leq s + 8^d.$$

Thanks to this ordering, we can lower bound the size of the active bin set at depth $d$ as

$$\min_{t \in [s, M(s,d,B_d^{(k)})]} |\mathcal{B}_t(d)| \geq 2^d + 1 - k,$$

and therefore,

$$\sum_{s=\tau_{l,m}}^{\tau_{l,m+1}-1} \sum_{d=0}^{m-1} \sum_{k=1}^{2^d} R_{s,d} \frac{1}{\min_{t \in [s, M(s,d,B_d^{(k)})]} |\mathcal{B}_t(d)|} \left(\sum_{t=s}^{M(s,d,B_d^{(k)})} \delta_t\left(\text{Parent}(B_{l,m}^{\text{last}}, d), B_d^{(k)}\right)\right)_+$$

$$\leq \sum_{d=0}^{m-1} \sum_{k=1}^{2^d} \frac{1}{2^d - k + 1} \sum_{s=\tau_{l,m}}^{\tau_{l,m+1}-1} R_{s,d} \left(\sum_{t=s}^{M(s,d,B_d^{(k)})} \delta_t\left(\text{Parent}(B_{l,m}^{\text{last}}, d), B_d^{(k)}\right)\right)_+.$$

We now proceed to bound the cumulative relative regret incurred by a bin $B_d^{(k)}$ with respect to its parent $\text{Parent}(B_{l,m}^{\text{last}}, d)$. By design of $B_{l,m}^{\text{last}}$ (it is a safe bin over the entire block), it follows that its parents $\{\text{Parent}(B_{l,m}^{\text{last}}, d)\}_{d \in \mathcal{D}_t}$ must also be safe. Indeed, if a parent $\text{Parent}(B_{l,m}^{\text{last}}, d)$ were deemed unsafe at some active depth $d \in \mathcal{D}_t$, it would necessarily be evicted. Due to the eviction mechanism, this would result in the eviction of all of its descendants, including $B_{l,m}^{\text{last}}$ itself, contradicting the assumption that $B_{l,m}^{\text{last}}$ remains safe.

Now, since the bin $B_d^{(k)}$ is active over the interval $[s, M(s,d,B_d^{(k)})]$, we can apply Proposition 5 to obtain the desired bound on the event $\mathcal{E}_1$.

$$\sum_{t=s}^{M(s,d,B_d^{(k)})} \delta_t\left(\mathrm{Parent}(B_{l,m}^{\mathrm{last}},d), B_d^{(k)}\right) \le (c_0+c_1)\log(T)\sqrt{2^d\left(M(s,d,B_d^{(k)})-s\right)\vee 4^d}$$

$$+\frac{8\left(M(s,d,B_d^{(k)})-s\right)}{2^d}$$

$$\le (c_0+c_1)\log(T)\sqrt{2^d 8^d \vee 4^d}+\frac{8\times 8^d}{2^d}$$

$$\le (c_0+c_1)\log(T)4^d + 8\times 4^d$$

$$\le (c_0+c_1+8)\log(T)4^d$$

$$\le c_2\log(T)4^d,$$

assuming horizon $T$ satisfies $\log(T)\ge 1$, and where the second inequality follows from the fact that the length of a replay at depth $d$ stating at round $s$ is at most $8^d$. We used the same definition of $c_2$ as above. Therefore, we have

$$\mathbb{E}\left[\mathbb{1}\{\mathcal{E}_1\}\sum_{d=0}^{m-1}\sum_{k=1}^{2^d}\frac{1}{2^d-k+1}\sum_{s=\tau_{l,m}}^{\tau_{l,m+1}-1}R_{s,d}\left(\sum_{t=s}^{M(s,d,B_d^{(k)})}\delta_t\left(\mathrm{Parent}(B_{l,m}^{\mathrm{last}},d),B_d^{(k)}\right)\right)_+ \,\Bigg|\, \mathcal{F}_{\tau_{l,m}}\right]$$

$$\le c_2\log(T)\mathbb{E}\left[\mathbb{1}\{\mathcal{E}_1\}\sum_{d=0}^{m-1}\sum_{k=1}^{2^d}\frac{1}{2^d-k+1}\sum_{s=\tau_{l,m}}^{\tau_{l,m+1}-1}R_{s,d}4^d\,\Bigg|\,\mathcal{F}_{\tau_{l,m}}\right]$$

$$\le c_2\log(T)\mathbb{E}\left[\mathbb{1}\{\mathcal{E}_1\}\sum_{d=0}^{m-1}\sum_{s=\tau_{l,m}}^{\tau_{l,m+1}-1}R_{s,d}(\log(2^d)+1)4^d\,\Bigg|\,\mathcal{F}_{\tau_{l,m}}\right]$$

$$\le c_2\log^2(T)\mathbb{E}\left[\mathbb{1}\{\mathcal{E}_1\}\sum_{d=0}^{m-1}\sum_{s=\tau_{l,m}}^{\tau_{l,m+1}-1}R_{s,d}4^d\,\Bigg|\,\mathcal{F}_{\tau_{l,m}}\right]$$

$$\le c_2\log^2(T)\mathbb{E}\left[\sum_{d=0}^{m-1}\sum_{s=\tau_{l,m}}^{\tau_{l,m+1}-1}R_{s,d}4^d\,\Bigg|\,\mathcal{F}_{\tau_{l,m}}\right],$$

where we used the fact that $m\le\log(T)$.

We now leverage the fact that *relatively few replays occur in expectation*, and the fact that the Bernoulli random variables $R_{s,d}$ are sampled *independently of the observations collected during the episode*. More precisely, all Bernoulli variables are drawn *i.i.d.* at the beginning of the block, *i.e.* at round $t=\tau_{l,m}$. This allows us to treat the scheduling decisions as fixed ahead of time. Furthermore, by the design of the replay scheduling mechanism in Algorithm 2,

$$\mathbb{E}\left[\sum_{d=0}^{m-1}\sum_{s=\tau_{l,m}}^{\tau_{l,m+1}-1}R_{s,d}4^d\,\Bigg|\,\mathcal{F}_{\tau_{l,m}}\right]\le\mathbb{E}\left[\sum_{d=0}^{m-1}\sum_{s=\tau_{l,m}+1}^{\tau_{l,m+1}-1}\frac{\mathbb{1}\{s-\tau_{l,m}\equiv 0[8^d]\}}{\sqrt{s-\tau_{l,m}}}\sqrt{8^d}4^d\,\Bigg|\,\mathcal{F}_{\tau_{l,m}}\right]$$

$$\le\sum_{d=0}^{m-1}\sum_{i=1}^{8^{m-d}+1}\frac{1}{\sqrt{i8^d}}\sqrt{8^d}4^d$$

$$\le 2\sqrt{8}\sum_{d=0}^{m-1}4^d\sqrt{8^{m-d}}$$

$$\le 2\sqrt{8}\sqrt{8^m}\frac{\sqrt{2^m}}{\sqrt{2}-1}\le 6\sqrt{8}\times 4^m. \tag{15}$$

Therefore, the regret contribution of replays at depth $d \in [\![0, m-1]\!]$ is upper bounded as

$$\mathbb{E}\left[\mathbb{1}\{\mathcal{E}_1\} \sum_{t=\tau_{l,m}}^{\tau_{l,m+1}-1} \sum_{d=0}^{m-1} \sum_{k=1}^{2^d} \mathbb{1}\{d_0(t) = d\} \frac{\delta_t\left(\text{Parent}(B_{l,m}^{\text{last}}, d), B_d^{(k)}\right)}{|\mathcal{B}_t(d)|} \mathbb{1}\left\{B_d^{(k)} \in \mathcal{B}_t(d)\right\} \Bigg| \mathcal{F}_{\tau_{l,m}}\right]$$

$$\leq 6\sqrt{8}c_2 \log^2(T) 4^m \,. \tag{16}$$

From Equation (13), combining Equations (14) and (16) gives the desired rate for (A.1),

$$(A.1) \leq 18(c_2 + 1) \log^2(T) 4^m \,. \tag{17}$$

**Upper bounding (A.2).** To bound this term, we apply the same reasoning previously used to control (A.1), and in particular the partition argument of Figure 2. By observing that each replay at depth $d_0(t) = d$ is initiated by a round $s$ such that $R_{s,d} = 1$, and by using the fact that each replay at depth $d$ lasts at most $8^d$ rounds, we distinguish the cumulative discretization bias incurred at depth $m$ and the one incurred at other depths $d < m$,

$$(A.2) = \mathbb{E}\left[\mathbb{1}\{\mathcal{E}_1\} \sum_{t=\tau_{l,m}}^{\tau_{l,m+1}-1} \frac{4}{2^{d_0(t)}} \Bigg| \mathcal{F}_{\tau_{l,m}}\right]$$

$$\leq \mathbb{E}\left[\sum_{t=\tau_{l,m}}^{\tau_{l,m+1}-1} \frac{4}{2^m} \mathbb{1}\{d_0(t) = m\} \Bigg| \mathcal{F}_{\tau_{l,m}}\right] + \mathbb{E}\left[\sum_{d=0}^{m-1} \sum_{t=\tau_{l,m}}^{\tau_{l,m+1}-1} \frac{4}{2^d} \mathbb{1}\{d_0(t) = d\} \Bigg| \mathcal{F}_{\tau_{l,m}}\right]$$

$$\leq 4^{m+1} + 4\mathbb{E}\left[\sum_{d=0}^{m-1} \sum_{t=\tau_{l,m}}^{\tau_{l,m+1}-1} \frac{1}{2^d} \mathbb{1}\{d_0(t) = d\} \Bigg| \mathcal{F}_{\tau_{l,m}}\right]$$

$$\leq 4^{m+1} + 4\mathbb{E}\left[\sum_{d=0}^{m-1} \sum_{s=\tau_{l,m}}^{\tau_{l,m+1}-1} R_{s,d} \frac{1}{2^d} 8^d \Bigg| \mathcal{F}_{\tau_{l,m}}\right]$$

$$= 4^{m+1} + 4\mathbb{E}\left[\sum_{d=0}^{m-1} \sum_{s=\tau_{l,m}}^{\tau_{l,m+1}-1} R_{s,d} 4^d \Bigg| \mathcal{F}_{\tau_{l,m}}\right] \,,$$

which is exactly the same term we bounded in the proof of (A.1). By (15) we thus have the upper bound

$$(A.2) \leq 4^{m+1} + 24\sqrt{8} \times 4^m = 28\sqrt{8} \times 4^m \,. \tag{18}$$

**Conclusion.** Combining (17) and (18) gives the desired rate of $4^m$ for term (A), as

$$(A) = \mathbb{E}\left[\mathbb{1}\{\mathcal{E}_1\} \sum_{t=\tau_{l,m}}^{\tau_{l,m+1}-1} \delta_t(B_{l,m}^{\text{last}}, B_{t,m}) \Bigg| \mathcal{F}_{\tau_{l,m}}\right]$$

$$\leq \left(18(c_2 + 1) \log^2(T) + 28\sqrt{8}\right) 4^m$$

Choosing $c_A = 18(c_2 + 2)$ concludes the proof.

### D.7 Upper bounding (B)

We recall that term (B) writes

$$(B) = \mathbb{E}\left[\mathbb{1}\{\mathcal{E}_1\} \sum_{t=\tau_{l,m}}^{\tau_{l,m+1}-1} \delta_t(B_{t,m}^\sharp, B_{l,m}^{\text{last}}) \Bigg| \mathcal{F}_{\tau_{l,m}}\right] \,.$$

As discussed in Section 5.3, the main challenge arises in scenarios where a significant shift occurs (as defined in Definition 1), causing the identity of the last safe arm $x_i^\sharp$ (and consequently the bin $B_{t,m}^\sharp$) to change. In order to avoid incurring large cumulative regret before this shift is detected, the algorithm must initiate a replay at the *appropriate depth* to identify the change in environment. We claim that (B) is also of order $4^m$ almost surely.

**Proposition 8** (**Upper bound of (B)**). *There exists a positive numerical constant $c_B$ such that*

$$(B) \leq c_B \log^2(T) 4^m .$$

To prove Proposition 8, we analyse the cumulative regret incurred by $B_{l,m}^{\text{last}}$ with respect to $B_{l,m}^{\sharp}$ by decomposing the rounds of the blocks into *bad segments*, that is, time intervals during which $B_{l,m}^{\text{last}}$ incurs significant regret relative to the safe bin $B_{t,m}^{\sharp}$. Importantly, by construction, $B_{l,m}^{\text{last}}$ is never evicted during the block $[\tau_{l,m}, \tau_{l,m+1} - 1]$, and thus remains active at depth $m$ throughout. However, since $B_{l,m}^{\text{last}}$ is only revealed at the *end* of the block, we must define bad segments relative to all possible bins at depth $m$.

**Definition 12** (**Bad segment and midpoint of a bad segment**). *Let $[\tau_{l,m}, \tau_{l,m+1}[$ a block, and let $[\tau_i, \tau_{i+1}[$ be any phase intersecting this block. For all bin $B \in \mathcal{T}_m$, define rounds $(s_{i,j}^m(B))_j$ with $s_{i,j}^m(B) \in [\tau_{l,m} \vee \tau_i, \tau_{i+1} \wedge \tau_{l,m} + 8^m[$ recursively as follows:*

- $s_{i,0}^m(B) = \tau_{l,m} \vee \tau_i.$

- $s_{i,j}^m(B)$ *is the smallest round in* $]s_{i,j-1}^m(B), \tau_{i+1} \wedge \tau_{l,m+1}[$ *such that*

$$\sum_{t=s_{i,j-1}^m(B)}^{s_{i,j}^m(B)} \delta_t(B_{t,m}^{\sharp}, B) \geq c_3 \log(T) \sqrt{2^m \left(s_{i,j}^m(B) - s_{i,j-1}^m(B)\right)} ,$$

*where $c_3$ is a fixed positive numerical constant.*

*For all bin $B \in \mathcal{T}_m$, we define the **midpoint of a bad segment** $[s_{i,j}^m(B), s_{i,j+1}^m(B)[$ as the round*

$$\tilde{s}_{i,j}^m(B) = \left\lfloor \frac{s_{i,j}^m(B) + s_{i,j+1}^m(B)}{2} \right\rfloor .$$

Now, our goal is to analyse the relative regret of any bin $B \in \mathcal{T}_m$ with respect to $B_{t,m}^{\sharp}$ on these *bad segments*. To build intuition, our objective is to prevent bad segments from accumulating for any bin $B \in \mathcal{T}_m$. Fortunately, the design of our replay mechanism ensures that if a *well-timed* replay at a *suitable depth* is triggered at the start of a bad segment, it can promptly detect the shift in rewards and evict $B$, thereby limiting the incurred regret.

**Proposition 9** (**Properties on bad segments**). *Let $B \in \mathcal{T}_m$ be a bin and let $[s_{i,j}^m(B), s_{i,j+1}^m(B)[$ be a bad segment. Then, on $\mathcal{E}_1$, any replay starting at round $t_{\text{start}} \in [s_{i,j}^m(B), s_{i,j+1}^m(B)[$ never evicts $\text{Parent}(B_{t,m}^{\sharp}, d)$ for $d \in \bigcap_{t=t_{\text{start}}}^{s_{i,j+1}^m(B)} \mathcal{D}_t$. Moreover, if a replays at depth $d_{i,j}$ satisfying*

$$8^{d_{i,j}+1} \leq s_{i,j+1}^m(B) - s_{i,j}^m(B) \leq 8^{d_{i,j}+2}$$

*starts at $t_{\text{start}} \in [s_{i,j}^m(B), \tilde{s}_{i,j}^m(B)[$, then $B$ is evicted by round $s_{i,j+1}^m(B)$.*

*Proof of Proposition 9.* Assume a replay evicts $\text{Parent}(B_{t,m}^{\sharp}, d)$ with $d \in \bigcap_{t=t_{\text{start}}}^{s_{i,j+1}^m(B)} \mathcal{D}_t$ and $t_{\text{start}} \geq s_{i,j}^m(B)$ before round $s_{i,j+1}^m(B)$. Then, by Proposition 1, it means that, on event $\mathcal{E}_1$, every arm $x \in B_{t,m}^{\sharp}$ incurs significant regret (Definition 1). This contradicts the definition of $B_{l,m}^{\sharp}$. Indeed, by definition, $x_t^{\sharp}$ does not change on $[t_{\text{start}}, s_{i,j+1}^m] \subset [\tau_i, \tau_{i+1}]$, so neither does $B_{t,m}^{\sharp}$. Therefore, such replay never evicts $\text{Parent}(B_{t,m}^{\sharp}, d)$.

Now, for all $B \in \mathcal{T}_m$, we have

$$\sum_{t=\tilde{s}_{i,j}^m(B)}^{s_{i,j+1}^m(B)} \delta_t(B_{t,m}^{\sharp}, B) = \sum_{s_{i,j}^m(B)}^{s_{i,j+1}^m(B)} \delta_t(B_{t,m}^{\sharp}, B) - \sum_{s_{i,j}^m(B)}^{\tilde{s}_{i,j}^m(B)-1} \delta_t(B_{t,m}^{\sharp}, B) .$$

By definition of a bad segment (Definition 12),

$$\sum_{s_{i,j}^m(B)}^{s_{i,j+1}^m(B)} \delta_t(B_{t,m}^{\sharp}, B) \geq c_3 \log(T) \sqrt{2^m \left(s_{i,j+1}^m(B) - s_{i,j}^m(B)\right)} .$$

and since $s_{i,j+1}^m(B)$ is defined as the *first* round that satisfies the above inequality,

$$\sum_{s_{i,j}^m(B)}^{\tilde{s}_{i,j}^m(B)} \delta_t(B_{t,m}^\sharp, B) \le c_3 \log(T)\sqrt{2^m\left(\tilde{s}_{i,j}^m(B) - s_{i,j}^m(B)\right)}\,.$$

By combining these two inequalities above,

$$\sum_{t=\tilde{s}_{i,j}^m(B)}^{s_{i,j+1}^m(B)} \delta_t(B_{t,m}^\sharp, B) \ge c_3 \log(T)\sqrt{2^m}\left(\sqrt{s_{i,j+1}^m(B) - s_{i,j}^m(B)} - \sqrt{\tilde{s}_{i,j}^m(B) - s_{i,j}^m(B)}\right)$$

$$\ge \frac{c_3}{4}\log(T)\sqrt{2^m\left(s_{i,j+1}^m(B) - \tilde{s}_{i,j}^m(B)\right)}\,,$$

where we used the inequality $\sqrt{a+b} - \sqrt{a} \ge \frac{\sqrt{b}}{4}$.

Since $B_{l,m}^\sharp$ is active during the whole of the bad segment, on $\mathcal{E}_1$, we have by Corollary 2, on event $\mathcal{E}_1$,

$$\sum_{t=\tilde{s}_{i,j}^m(B)}^{s_{i,j+1}^m(B)} \hat{\delta}_t(B_{t,m}^\sharp, B)$$

$$\ge \sum_{t=\tilde{s}_{i,j}^m(B)}^{s_{i,j+1}^m(B)} \delta_t(B_{t,m}^\sharp, B) - c_1\log(T)\sqrt{(s_{i,j+1}^m(B) - \tilde{s}_{i,j}^m(B))2^m} - \frac{4(s_{i,j+1}^m(B) - \tilde{s}_{i,j}^m(B))}{2^m}$$

$$\ge \left(\frac{c_3}{4} - c_1\right)\log(T)\sqrt{\left(s_{i,j+1}^m(B) - \tilde{s}_{i,j}^m(B)\right)2^m} - \frac{4(s_{i,j+1}^m(B) - \tilde{s}_{i,j}^m(B))}{2^m}\,.$$

Using the fact that $s_{i,j+1}^m(B) - \tilde{s}_{i,j}^m(B) \le 8^{d_{i,j}+2} \le 8^{m+1}$, we have

$$\frac{s_{i,j+1}^m(B) - \tilde{s}_{i,j}^m(B)}{2^m} \le 8\sqrt{s_{i,j+1}^m(B) - \tilde{s}_{i,j}^m(B)}\frac{(2\sqrt{2})^m}{2^m}$$

$$\le 8\sqrt{2^m(s_{i,j+1}^m(B) - \tilde{s}_{i,j}^m(B))}\,,$$

and therefore

$$\sum_{t=\tilde{s}_{i,j}^m(B)}^{s_{i,j+1}^m(B)} \hat{\delta}_t(B_{t,m}^\sharp, B) \ge \left(\frac{c_3}{4} - c_1 - \frac{32}{\log(T)}\right)\log(T)\sqrt{\left(s_{i,j+1}^m(B) - \tilde{s}_{i,j}^m(B)\right)2^m}\,.$$

Assuming horizon large enough, *e.g.* $\log(T) \ge 1$, setting $c_3 = 140 + 4c_1$ implies

$$\sum_{t=\tilde{s}_{i,j}^m(B)}^{s_{i,j+1}^m(B)} \hat{\delta}_t(B_{t,m}^\sharp, B) \ge 3\log(T)\sqrt{\left(s_{i,j+1}^m(B) - \tilde{s}_{i,j}^m(B)\right)2^m}\,,$$

and hence by ($\star$), on event $\mathcal{E}_1$, $B$ is evicted. $\qquad\square$

Having established that a well-timed replay at the appropriate depth can evict a bin $B$, it remains to demonstrate that such a replay is indeed *triggered* before the cumulative regret accrued over the bad segments is too large. To formally define the *earliest* round at which this cumulative regret becomes significant, we introduce the notion of a *bad round*.

**Definition 13** (**Bad round with respect to bin $B$**)**.** *For all block $[\tau_{l,m}, \tau_{l,m+1}[$, for all bin $B \in \mathcal{T}_m$, we define the bad round with respect to $B$ as*

$$s_{l,m}(B) = \inf\left\{s > \tau_{l,m} \;:\; \sum_{(i,j)\in\mathcal{P}(B,s)}\sqrt{2^m(s_{i,j+1}^m(B) - s_{i,j}^m(B))} > c_4\log(T)\sqrt{2^m(s - \tau_{l,m})}\right\}$$

$$\wedge\left(\tau_{l,m} + 8^m\right),$$

*where $c_4$ is a fixed positive numerical constant, and we define the set $\mathcal{P}(B,s)$ as*

$$\mathcal{P}(B,s) = \left\{(i,j) \;:\; i,j \in \mathbb{N}, \text{ such that } [s_{i,j}^m(B), s_{i,j+1}^m(B)] \text{ is a bad segment and } s_{i,j+1}^m(B) < s\right\}.$$

Now, let $[\tau_{l,m}, \tau_{l,m+1})$, be a block, and let $B \in \mathcal{T}_m$. We distinguish two cases for this bin.

**Case 1: the bad round happens after the end of the block** $(s_{l,m}(B) = \tau_{l,m} + 8^m)$. By applying the definition of $s_{l,m}(B)$ (Definition 13) directly, we have

$$\sum_{(i,j) \in \mathcal{P}(B, s_{l,m}(B))} \sqrt{2^m(s_{i,j+1}^m(B) - s_{i,j}^m(B))} \leq c_4 \log(T)\sqrt{8^m 2^m} = c_4 \log(T) 4^m .$$

**Case 2: the bad round happens before the end of the block** $(s_{l,m}(B) < \tau_{l,m} + 8^m)$. In this case, we claim the following result.

**Proposition 10** (**A well-timed replay at a suitable depth is triggered**). *Let* $[\tau_{l,m}, \tau_{l,m+1}[$ *be a block. We define* $\mathcal{E}_2(\tau_{l,m})$ *as the following event:*

$\forall B \in \mathcal{T}_m$ *such that* $s_{l,m}(B) < \tau_{l,m+1}$, *there exists a bad segment* $[s_{i,j}^m(B), s_{i,j+1}^m(B)]$ *such that* $s_{i,j+1}^m(B) < s_{l,m}(B)$ *and a* $\underbrace{\texttt{Replay}(t_{\text{start}}, d_{i,j})}_{\text{(Definition 6)}}$ *such that* $t_{\text{start}}$ *and* $d_{i,j}$ *satisfy*

$t_{\text{start}} \in [s_{i,j}^m(B), s_{i,j+1}^m(B)]$ *and* $8^{d_{i,j}+1} \leq s_{i,j+1}^m(B) - s_{i,j}^m(B) \leq 8^{d_{i,j}+2}$ .

*Then,* $\mathcal{E}_2(\tau_{l,m})$ *holds with probability at least* $1 - 1/T^2$.

*Proof of Proposition 10.* Let $[\tau_{l,m}, \tau_{l,m+1}[$ be a block, and let $B \in \mathcal{T}_m$ such that $s_{l,m}(B) < \tau_{l,m} + 8^m$. Let $d_{i,j}$ the integer satisfying

$$8^{d_{i,j}+1} \leq s_{i,j+1}^m(B) - s_{i,j}^m(B) \leq 8^{d_{i,j}+2} .$$

First, we remark that $R_{t,d_{i,j}}$, $s_{i,j}^m(B)$, $\tilde{s}_{i,j}^m(B)$ and $s_{l,m}(B)$ only depend on the fixed bin $B$, the starting round of the block $\tau_{l,m}$ $B_{t,m}^\sharp$ (which is deterministic), and the randomness of scheduling of the replays. Using Chernoff's bound over randomness of the algorithm conditionally on $\mathcal{F}_{\tau_{l,m}}$ gives

$$\mathbb{P}\left(\sum_{(i,j) \in \mathcal{P}(B, s_{l,m}(B))} \sum_{t=s_{i,j}^m(B)}^{\tilde{s}_{i,j}^m(B)} R_{t,d_{ij}} \leq \frac{1}{2}\mathbb{E}\left[\sum_{(i,j) \in \mathcal{P}(B, s_{l,m}(B))} \sum_{t=s_{i,j}^m(B)}^{\tilde{s}_{i,j}^m(B)} R_{t,d_{ij}} \,\Big|\, \mathcal{F}_{\tau_{l,m}}\right] \,\Big|\, \mathcal{F}_{\tau_{l,m}}\right)$$

$$\leq \exp\left(-\frac{1}{8}\mathbb{E}\left[\sum_{(i,j) \in \mathcal{P}(B, s_{l,m}(B))} \sum_{t=s_{i,j}^m(B)}^{\tilde{s}_{i,j}^m(B)} R_{t,d_{ij}} \,\Big|\, \mathcal{F}_{\tau_{l,m}}\right]\right) . \tag{19}$$

Then, we lower bound the expectation of (19),

$$\mathbb{E}\left[\sum_{(i,j) \in \mathcal{P}(B, s_{l,m}(B))} \sum_{t=s_{i,j}^m(B)}^{\tilde{s}_{i,j}^m(B)} R_{t,d_{ij}} \,\Big|\, \mathcal{F}_{\tau_{l,m}}\right]$$

$$= \mathbb{E}\left[\sum_{(i,j) \in \mathcal{P}(B, s_{l,m}(B))} \sum_{t=s_{i,j}^m(B)}^{\tilde{s}_{i,j}^m(B)} \frac{\mathbb{1}\{t - \tau_{l,m} \equiv 0[8^{d_{i,j}}]\}}{\sqrt{t - \tau_{l,m}}} \sqrt{8^{d_{i,j}}} \,\Big|\, \mathcal{F}_{\tau_{l,m}}\right]$$

$$\geq \frac{1}{8}\mathbb{E}\left[\sum_{(i,j) \in \mathcal{P}(B)} \frac{1}{\sqrt{\tilde{s}_{i,j}^m(B) - \tau_{l,m}}} \sqrt{8^{d_{i,j}+2}} \,\Big|\, \mathcal{F}_{\tau_{l,m}}\right]$$

$$\geq \frac{1}{8}\mathbb{E}\left[\sum_{(i,j) \in \mathcal{P}(B)} \sqrt{\frac{s_{i,j+1}^m(B) - s_{i,j}^m(B)}{s_{l,m}(B) - \tau_{l,m}}} \,\Big|\, \mathcal{F}_{\tau_{l,m}}\right]$$

$$\geq \frac{c_4}{8}\log(T) ,$$

where we use the fact that there is at least one round $t \in [s_{i,j}^m(B), \tilde{s}_{i,j}^m(B)]$ such that $t - \tau_{l,m} \equiv 0[8^{d_{ij}}]$, *i.e.* that at least a replay at depth $d_{i,j}$ that can be scheduled in $[s_{i,j}^m(B), s_{i,j+1}^m(B)]$, and where we

used Definition 13 in the last inequality. Setting $c_4 = 192$, we have

$$\mathbb{P}\left(\sum_{(i,j)\in\mathcal{P}(B,s_{l,m}(B))}\sum_{t=s_{i,j}^m(B)}^{\tilde{s}_{i,j}^m(B)} R_{t,d_{ij}} \leq \frac{1}{2}\mathbb{E}\left[\sum_{(i,j)\in\mathcal{P}(B,s_{l,m}(B))}\sum_{t=s_{i,j}^m(B)}^{\tilde{s}_{i,j}^m(B)} R_{t,d_{ij}}\,\middle|\,\mathcal{F}_{\tau_{l,m}}\right]\,\middle|\,\mathcal{F}_{\tau_{l,m}}\right)$$
$$\leq 1/T^3\,.$$

Taking a union bound with respect to all possible bins $B \in \mathcal{T}_m$ (there are $2^m \leq T$ choices) concludes the proof. $\qquad\square$

Therefore, using Proposition 10, for any bin $B \in \mathcal{T}_m$, on event $\mathcal{E}_1 \cap \mathcal{E}_2(\tau_{l,m})$, we have that there exists a bad segment $[s_{i,j_0}^m(B), s_{i,j_0+1}^m(B)]$ such that $s_{i,j_0+1}^m(B) < s_{l,m}(B) < \tau_{l,m+1}$, and there exists a $\mathtt{Replay}(t_{\text{start}}, d_{i,j_0})$ starting within this segment, $t_{\text{start}} \in [s_{i,j_0}^m(B), s_{i,j_0+1}^m(B)]$ and $d_{i,j_0}$ satisfying $8^{d_{i,j_0}} \leq s_{i,j_0+1}^m(B) - s_{i,j_0}^m(B) \leq 8^{d_{i,j_0}+2}$. By Proposition 9, this implies that bin $B$ is evicted by round $s_{i,j_0+1}^m(B)$. Therefore, on $\mathcal{E}_1 \cap \mathcal{E}_2(\tau_{l,m})$, using the definition of the bad round $s_{l,m}(B) < \tau_{l,m+1}$,

$$\sum_{(i,j)\in\mathcal{P}(B,s_{l,m}(B))} \sqrt{2^m(s_{i,j+1}^m(B) - s_{i,j}^m(B))} \leq c_4 \log(T)\sqrt{2^m(s_{i,j_0} - \tau_{l,m})}$$
$$\leq c_4 \log(T)4^m\,.$$

Since Proposition 10 holds uniformly over all bins $B \in \mathcal{T}_m$, it also holds for the particular choice $B = B_{l,m}^{\text{last}}$, and hence in both cases $s_{l,m}(B) = \tau_{l,m} + 8^m$ and $s_{l,m}(B) < \tau_{l,m} + 8^m$, we have

$$\sum_{(i,j)\in\mathcal{P}(B_{l,m}^{\text{last}},s_{l,m}(B_{l,m}^{\text{last}}))} \sqrt{2^m(s_{i,j+1}^m(B_{l,m}^{\text{last}}) - s_{i,j}^m(B_{l,m}^{\text{last}}))} \leq c_4 \log(T)4^m\,.$$

Now recall by Definition 12 that $s_{i,j_0+1}^m(B_{l,m}^{\text{last}})$ is the *earliest* round satisfying the lower bound on the cumulative regret

$$\sum_{t=s_{i,j_0}^m(B_{l,m}^{\text{last}})}^{s_{i,j_0+1}^m(B_{l,m}^{\text{last}})} \delta_t(B_{t,m}^{\sharp}, B_{l,m}^{\text{last}}) \geq c_3 \log(T)\sqrt{2^m\left(s_{i,j_0}^m(B_{l,m}^{\text{last}}) - s_{i,j_0-1}^m(B_{l,m}^{\text{last}})\right)},$$

Therefore, up to round $s_{i,j_0+1}^m(B_{l,m}^{\text{last}}) - 1$ included, the cumulative regret can be *upper bounded* as

$$\sum_{t=s_{i,j_0}^m(B_{l,m}^{\text{last}})}^{s_{i,j_0+1}^m(B_{l,m}^{\text{last}})-1} \delta_t(B_{t,m}^{\sharp}, B_{l,m}^{\text{last}}) \leq c_3 \log(T)\sqrt{2^m\left(s_{i,j_0}^m(B_{l,m}^{\text{last}}) - s_{i,j_0-1}^m(B_{l,m}^{\text{last}}) - 1\right)}\,.$$

Since on event $\mathcal{E}_1 \cap \mathcal{E}_2(\tau_{l,m})$, $B_{l,m}^{\text{last}}$ is evicted by round $s_{i,j_0+1}(B_{l,m}^{\text{last}})$, we can upper bound the cumulative regret contribution of $B_{l,m}^{\text{last}}$ with respect to $B_{t,m}^{\sharp}$ over all bad segments of $B_{l,m}^{\text{last}}$,

$$\sum_{(i,j)\in\mathcal{P}(B_{l,m}^{\text{last}},s_{l,m}(B_{l,m}^{\text{last}}))}\sum_{t=s_{i,j}^m(B_{l,m}^{\text{last}})}^{s_{i,j+1}^m(B_{l,m}^{\text{last}})} \delta_t(B_{t,m}^{\sharp}, B_{l,m}^{\text{last}}) \leq c_3 c_4 \log^2(T)4^m\,. \tag{20}$$

It remains to bound the regret incurred over the *non-bad segments*. To this end, we observe from Definition 12 that each phase contains *at most one such segment*. Therefore, there exists a positive numerical constant $c_5$ such that

$$\sum_{I\subseteq[\tau_{l,m},\tau_{l,m+1}[\text{ s.t. } I \text{ is non-bad segment}}\sum_{t\in I} \delta_t(B_{t,m}^{\sharp}, B_{l,m}^{\text{last}}) \leq c_5 \log(T)4^m\,. \tag{21}$$

Summing Equations (20) and (21) gives the total relative regret of $B_{l,m}^{\text{last}}$ with respect to $B_{t,m}^{\sharp}$ over the block on event $\mathcal{E}_1 \cap \mathcal{E}_2(\tau_{l,m})$,

$$\sum_{t=\tau_{l,m}}^{\tau_{l,m+1}-1} \delta_t(B_{t,m}^{\sharp}, B_{l,m}^{\text{last}}) \leq \overbrace{(c_3 c_4 + c_5)}^{=c_6} \log^2(T) 4^m .$$

Finally, we use the fact that event $\mathcal{E}_2(\tau_{l,m})$ holds with probability at least $1 - 1/T^2$ (Proposition 10),

$$(B) = \mathbb{E}\left[ \mathbb{1}\{\mathcal{E}_1\} \sum_{t=\tau_{l,m}}^{\tau_{l,m+1}-1} \delta_t(B_{t,m}^{\sharp}, B_{l,m}^{\text{last}}) \,\middle|\, \mathcal{F}_{\tau_{l,m}} \right]$$

$$= \mathbb{E}\left[ \mathbb{1}\{\mathcal{E}_1 \cap \mathcal{E}_2(\tau_{l,m})\} \sum_{t=\tau_{l,m}}^{\tau_{l,m+1}-1} \delta_t(B_{t,m}^{\sharp}, B_{l,m}^{\text{last}}) \,\middle|\, \mathcal{F}_{\tau_{l,m}} \right]$$

$$+ \mathbb{E}\left[ \mathbb{1}\{\mathcal{E}_1 \cap \mathcal{E}_2^c(\tau_{l,m})\} \sum_{t=\tau_{l,m}}^{\tau_{l,m+1}-1} \delta_t(B_{t,m}^{\sharp}, B_{l,m}^{\text{last}}) \,\middle|\, \mathcal{F}_{\tau_{l,m}} \right]$$

$$\leq c_6 \log^2(T) 4^m + \frac{8^m}{T^2}$$

$$\leq c_6 \log^2(T) 4^m + \frac{1}{T} ,$$

where we used the fact that $\mathbb{1}\{\mathcal{E}_1 \cap \mathcal{E}_2^c(\tau_{l,m})\} \leq \mathbb{1}\{\mathcal{E}_2^c\}$.

Choosing $c_B = c_6 + 1$ concludes the proof.

### D.8 Conclusion of the proof

Recall from Equation (11) that we have

$$\mathbb{E}\left[ R(\pi_{\text{MDBE}}, T) \right] \leq \log(T) \sum_{i=0}^{\tilde{L}_T} (\tau_{i+1} - \tau_i)^{2/3} + \frac{7}{T}$$

$$+ 6 \sum_{l=1}^{T} \mathbb{E}\left[ \mathbb{1}\{\mathcal{E}_1\} (t_{l+1} - t_l)^{2/3} \right] + \sum_{l=1}^{T} \mathbb{E}\left[ \mathbb{1}\{\mathcal{E}_1\} \sum_{m=0}^{M_l} \sum_{t=\tau_{l,m}}^{\tau_{l,m+1}-1} \delta_t(B_{t,m}^{\sharp}, B_{t,m}) \right] .$$

Propositions 7 and 8 prove that the last sum is upper bounded as

$$\mathbb{E}\left[ \mathbb{1}\{\mathcal{E}_1\} \sum_{m=0}^{M_l} \sum_{t=\tau_{l,m}}^{\tau_{l,m+1}-1} \delta_t(B_{t,m}^{\sharp}, B_{t,m}) \right] \leq (c_A + c_B) \log^2(T) \mathbb{E}\left[ \sum_{m=0}^{M_l} 4^m \right]$$

$$\leq 6(c_A + c_B) \log^2(T) \mathbb{E}\left[ (t_{l+1} - t_l)^{2/3} \right]$$

$$\leq 6(c_A + c_B) \log^2(T) \mathbb{E}\left[ \mathbb{1}\{\mathcal{E}_1\} (t_{l+1} - t_l)^{2/3} \right] + \frac{c_A + c_B}{T^2} .$$

where we used Equation (10) for the second inequality, and where we reconditioned on event $\mathcal{E}_1$ in the last line. Therefore,

$$\mathbb{E}\left[ R(\pi_{\text{MDBE}}, T) \right] \leq \log(T) \sum_{i=0}^{\tilde{L}_T} (\tau_{i+1} - \tau_i)^{2/3} + \frac{7 + c_A + c_B}{T}$$

$$+ 6 \mathbb{E}\left[ \mathbb{1}\{\mathcal{E}_1\} \sum_{l=1}^{T} (t_{l+1} - t_l)^{2/3} \right] + 6(c_A + c_B) \log^2(T) \mathbb{E}\left[ \mathbb{1}\{\mathcal{E}_1\} \sum_{l=1}^{T} (t_{l+1} - t_l)^{2/3} \right] .$$

$$(22)$$

We conclude the proof by relating episodes and phases, which will permit to bound the final term

$$\mathbb{E}\left[ \mathbb{1}\{\mathcal{E}_1\} \sum_{l=1}^{T} (t_{l+1} - t_l)^{2/3} \right] .$$

**Definition 14** (**Phases intersecting episode** $l$). *We define the phases intersecting episode* $[t_l, t_{l+1}[$ *as*

$$\text{Phases}(t_l, t_{l+1}) = \left\{ i \in [\tilde{L}_T] \ : \ [\tau_i, \tau_{i+1}[ \cap [t_l, t_{l+1}[ \neq \emptyset \right\}.$$

With this definition, we can rewrite the upper bound on cumulative regret over each episode as

$$\mathbb{E}\left[\mathbb{1}\{\mathcal{E}_1\} \sum_{l=1}^{T} (t_{l+1} - t_l)^{2/3}\right] \leq \mathbb{E}\left[\mathbb{1}\{\mathcal{E}_1\} \sum_{l=1}^{T} \sum_{i \in \text{Phases}(t_l, t_{l+1})} (\tau_{i+1} - \tau_i)^{2/3}\right].$$

Recall that Proposition 6 shows that on $\mathcal{E}_1$, each phase $[\tau_i, \tau_{i+1}[$ intersects at most two episodes. Therefore,

$$\mathbb{E}\left[\mathbb{1}\{\mathcal{E}_1\} \sum_{l=1}^{T} \sum_{i \in \text{Phases}(t_l, t_{l+1})} (\tau_{i+1} - \tau_i)^{2/3}\right] \leq 2 \sum_{i=1}^{\tilde{L}_T} (\tau_{i+1} - \tau_i)^{2/3},$$

and finally we can conclude the proof by plugging the bound above into (22),

$$\mathbb{E}\left[R(\pi_{\text{MDBE}}, T)\right] \leq \log(T) \sum_{i=0}^{\tilde{L}_T} (\tau_{i+1} - \tau_i)^{2/3} + \frac{7 + c_A + c_B}{T}$$

$$+ 12 \sum_{i=1}^{\tilde{L}_T} (\tau_{i+1} - \tau_i)^{2/3} + 12(c_A + c_B) \log^2(T) \sum_{i=1}^{\tilde{L}_T} (\tau_{i+1} - \tau_i)^{2/3}$$

$$\leq \left(\log(T) + 12 + 12(c_A + c_B) \log^2(T)\right) \sum_{i=1}^{\tilde{L}_T} (\tau_{i+1} - \tau_i)^{2/3} + \frac{7 + c_A + c_B}{T}$$

$$\leq \bar{c}_1 \log^2(T) \sum_{i=1}^{\tilde{L}_T} (\tau_{i+1} - \tau_i)^{2/3},$$

where $\bar{c}_1 = 12(c_A + c_B) + 2$. This completes the proof. We summarize in Table 2 the exact values of the numerical constants used in the proof.

| Constant | Numerical value |
|:---:|:---:|
| $c_0$ | $3 + 7(e-1)\sqrt{2}$ |
| $c_1$ | $7(e-1)\sqrt{2}$ |
| $c_2$ | $c_0 + c_1 + 8$ |
| $c_3$ | $140 + 4c_1$ |
| $c_4$ | $192$ |
| $c_5$ | Equation (21) |
| $c_6$ | $c_3 c_4 + c_5$ |
| $c_A$ | $18(c_0 + c_1 + 10)$ |
| $c_B$ | $c_6 + 1$ |
| $\bar{c}_1$ | $12(c_A + c_B) + 2$ |

Table 2: Summary of numerical constants used in the proof

# E Proof of Corollary 1

Recall the definition of the total variation over the horizon $T$:

$$V_T = \sum_{t=1}^{T-1} \max_{x \in [0,1]} |\mu_{t+1}(x) - \mu_t(x)|.$$

Fix a phase $[\tau_i, \tau_{i+1}[$, where $\tau_{i+1} \leq T + 1$. Define the total variation within this phase as

$$V_{[\tau_i, \tau_{i+1}[} = \sum_{t=\tau_i}^{\tau_{i+1}-1} \max_{x \in [0,1]} |\mu_{t+1}(x) - \mu_t(x)|.$$

Let $x_i = \mathrm{argmax}_{x \in [0,1]} \mu_{\tau_{i+1}}(x)$ denote the best arm at the end of phase $i$. By definition, there exists an interval $[s_1, s_2] \subset [\tau_i, \tau_{i+1}[$ over which $x_i$ suffers significant regret:

$$\sum_{t=s_1}^{s_2} \delta_t(x_i) \geq \log(T)\,(s_2 - s_1)^{2/3}.$$

Now observe that

$$(s_2 - s_1)^{2/3} = \sum_{t=s_1}^{s_2} \frac{1}{(s_2 - s_1)^{1/3}} \geq \sum_{t=s_1}^{s_2} \frac{1}{(\tau_{i+1} - \tau_i)^{1/3}}\,,$$

which implies

$$\sum_{t=s_1}^{s_2} \delta_t(x_i) \geq \sum_{t=s_1}^{s_2} \frac{\log(T)}{(\tau_{i+1} - \tau_i)^{1/3}}\,.$$

Hence, there exists at least one round $t_i \in [s_1, s_2]$ such that:

$$\delta_{t_i}(x_i) \geq \frac{\log(T)}{(\tau_{i+1} - \tau_i)^{1/3}}\,.$$

Define $\tilde{x}_i = \mathrm{argmax}_{x \in [0,1]} \mu_{t_i}(x)$. Then:

$$\delta_{t_i}(x_i) \leq \mu_{t_i}(\tilde{x}_i) - \mu_{t_i}(x_i) + \underbrace{\mu_{\tau_{i+1}}(x_i) - \mu_{\tau_{i+1}}(\tilde{x}_i)}_{\geq 0}$$

$$\leq |\mu_{\tau_{i+1}}(x_i) - \mu_{t_i}(x_i)| + |\mu_{t_i}(\tilde{x}_i) - \mu_{\tau_{i+1}}(\tilde{x}_i)|$$

$$\leq 2V_{[\tau_i, \tau_{i+1})}\,.$$

Therefore, assuming $\log(T) \geq 1$, we conclude that

$$\frac{1}{(\tau_{i+1} - \tau_i)^{1/3}} \leq 2V_{[\tau_i, \tau_{i+1})}\,. \tag{23}$$

We now recall the upper bound from Theorem 2:

$$\mathbb{E}\left[R(\pi_{\mathrm{MDBE}}, T)\right] \leq \bar{c}\log^2(T) \sum_{i=0}^{\tilde{L}_T} (\tau_{i+1} - \tau_i)^{2/3}\,.$$

We decompose the sum as:

$$\sum_{i=0}^{\tilde{L}_T} (\tau_{i+1} - \tau_i)^{2/3} = (T + 1 - \tau_{\tilde{L}_T})^{2/3} + \sum_{i=0}^{\tilde{L}_T - 1} (\tau_{i+1} - \tau_i)^{2/3} \leq T^{2/3} + \sum_{i=0}^{\tilde{L}_T - 1} (\tau_{i+1} - \tau_i)^{2/3}\,.$$

Applying Hölder's inequality gives

$$\sum_{i=0}^{\tilde{L}_T - 1} (\tau_{i+1} - \tau_i)^{2/3} = \sum_{i=0}^{\tilde{L}_T - 1} \frac{1}{(\tau_{i+1} - \tau_i)^{1/12}}\,(\tau_{i+1} - \tau_i)^{3/4}$$

$$\leq \left(\sum_{i=0}^{\tilde{L}_T - 1} \frac{1}{(\tau_{i+1} - \tau_i)^{1/3}}\right)^{1/4} \left(\sum_{i=0}^{\tilde{L}_T - 1} (\tau_{i+1} - \tau_i)\right)^{3/4}\,.$$

Finally, by inequality (23), we have

$$\sum_{i=0}^{\tilde{L}_T - 1} \frac{1}{(\tau_{i+1} - \tau_i)^{1/3}} \leq 2 \sum_{i=0}^{\tilde{L}_T - 1} V_{[\tau_i, \tau_{i+1})} \leq 2V_T\,.$$

Therefore,

$$\sum_{i=0}^{\tilde{L}_T - 1} (\tau_{i+1} - \tau_i)^{2/3} \leq (2V_T)^{1/4} \times T^{3/4} = 2^{1/4} V_T^{1/4} T^{3/4}\,.$$

# F    Proof of the lower bound (Theorem 1)

**Proof sketch.** The proof of Theorem 1 relies on the following ideas. First, note that the lower bound $T^{2/3}$ follows from classical results of Lipschitz bandits [Kleinberg, 2004]. To prove the lower bound either with respect to total variation $V_T$, or with respect to the total number of changes $L_T$, we construct $T/\tau$ stationary sub-problems with horizon $\tau$ to be chosen later. For each problem, we design $K = \tau^{1/3}$ possible Lipschitz-continuous rewards function. Each of these functions is almost flat, with the exception of one bump of size $\epsilon \approx \tau^{-1/3}$ hidden among $K$ possible bins. Classical arguments show that any bandit algorithm will misidentify the optimal bin with constant probability in some problem instance, thus suffering a regret $\tau^{2/3}$. Then, we create a dynamic problem instance by concatenating $T/\tau$ such problems, where the reward function corresponding to each problem of horizon $\tau$ is chosen independently. The total regret of any algorithm must then be

$$\frac{T}{\tau}\tau^{2/3} = T\tau^{-1/3}.$$

To obtain a lower bound depending on the number of reward changes $L_T$, we set $\tau = T/L$, so that there are exactly $T/\tau$ stationary sub-problems. This yields a regret of the order $L_T^{1/3}T^{2/3}$. To obtain a lower bound depending on the total variation $V_T$, we set $\tau = (T/V_T)^{3/4}$. Then, we see that there are $T/\tau$ changes of magnitude $\tau^{-1/3}$ each, so that the total variation is indeed $T/\tau \times \tau^{-1/3} = V_T$; moreover the cumulative regret over all problems is $T\tau^{-1/3} = T^{3/4}V^{1/4}$.

In the following, we assume that, conditionally on $x_t$, the reward is a Bernoulli random variable with mean $\mu_t(x_t)$.

**Notations.** For some integer $K \geq 2$ to be specified later, we set $\tau = K^3$ and $\epsilon = 1/(2K)$ (note that $\epsilon \leq 1/4$). We define $L = \lfloor T/\tau \rfloor$, and for $l \in [\![L]\!]$, we define the stationary phase $\mathcal{P}^l = \{(l-1)\tau + 1, \ldots, l\tau - 1\}$. We also define the last stationary phase $\mathcal{P} = \{\tau L, \ldots, T\}$. For $k < K$, define the intervals $I_k = [\frac{k-1}{K}, \frac{k}{K}[$, and we define $I_K = [\frac{K-1}{K}, 1]$. Finally, for a given sequence of rewards $\mu$, we denote by $\mathbb{E}_\mu$ and $\mathbb{P}_\mu$ the expectation and probability when the reward sequence is $\mu$. Note that at each round $t$, the algorithm $\pi$ only depends on an internal randomisation and on $\mathcal{F}_{t-1}$-measurable events. Thus, for all $l \leq L$ and all $\mathcal{F}_{\tau l}$-measurable random variable $X$ and event $E$, $\mathbb{E}_\mu[X]$ and $\mathbb{P}_\mu(E)$ only depend on the sequence of rewards up to phase $l$, *i.e.* on $(\mu_t)_{t \leq \tau l}$. In the following, we therefore abuse notation, and define for reward sequences $\mu$ of length $\tau l$, the expectation $\mathbb{E}_\mu$ and probability $\mathbb{P}_\mu$ pertaining to the first $l$ phases, *i.e.* to $\mathcal{F}_{\tau l}$-measurable random variables and events.

**Reward function for a stationary phase.** We define the possible choice of reward function during one stationary phase $\mathcal{P}^l$. For each $k \in \{2, ..., K\}$, define the function $m^k : [0, 1] \mapsto [0, 1]$ as

$$m^k(x) = \frac{1-\epsilon}{2} + \frac{\epsilon - \left|x - \frac{1}{2}\right|}{2}\mathbb{1}\left\{x \in I_1\right\} + \left(\epsilon - \left|x - \frac{2k-1}{2K}\right|\right)\mathbb{1}\left\{x \in I_k\right\}.$$

We also define

$$m^1(x) = \frac{1-\epsilon}{2} + \frac{\epsilon - \left|x - \frac{1}{2}\right|}{2}\mathbb{1}\left\{x \in I_1\right\}.$$

Note that for $k \in [\![K]\!]$, when the mean reward function is $m^k$ the optimal action is $\frac{2k-1}{K}$; moreover any action chosen outside of interval $I_k$ will have an instantaneous regret at least $\frac{\epsilon}{2}$. For $l \in [\![L]\!]$ and $k \in [\![K]\!]$, we denote $N_k^l = \sum_{t \in \mathcal{P}^l} \mathbb{1}\left\{x_t \in I_k\right\}$ the number of actions chosen in interval $I_k$ during the stationary phase $\mathcal{P}^l$. Examples of possible reward functions for $K = 5$ are given in Figure 3.

**Sequential construction of the non-stationary reward function.** For a given algorithm $\pi$, we build a difficult non-stationary reward function sequentially. We show by induction that for all $l \leq L$, there exists a non-stationary sequence of reward functions $(\mu_t)_{t \leq l\tau}$ such that the expected cumulative regret of algorithm $\pi$ over the stationary phases $\mathcal{P}_1, ..., \mathcal{P}_l$ is at least $0.01K^2l$ for all $l \in [\![L]\!]$. We abuse notation and define $\mathcal{P}_0 = \emptyset$, so that the base case of the recursion (corresponding to $l = 0$) becomes trivial.

We then proceed to prove this statement by induction. Assume that the statement holds for some $l$ such that $0 \leq l < L$ and some sequence of reward functions $(\mu_t)_{t \leq l\tau}$. For $k \leq K$, we define $\mu^k$ as the

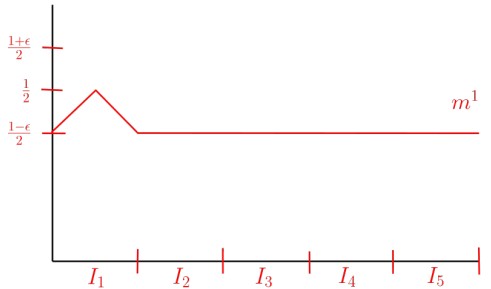 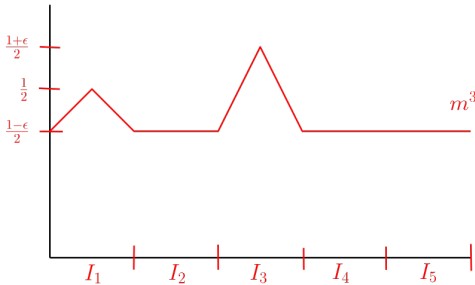

Figure 3: Reward functions $m^1$ (left) and $m^3$ (right) when $K = 5$.

extended sequence of rewards such that $\mu_t^k = \mu_t \mathbb{1}_{t \leq l\tau} + m^k \mathbb{1}_{l\tau < t \leq (l+1)\tau}$. Then, let $k^*$ be defined as

$$k^* = \underset{k}{\operatorname{argmin}} \, \mathbb{E}_{\mu^1}[N_k^{l+1}].$$

It holds that $\mathbb{E}_{\mu^1}[N_{k^*}^{l+1}] \leq \frac{\tau}{K}$.

Now, if $k^* = 1$, it means that under $\mu^1$, the algorithm $\pi$ chooses in expectation at least $\frac{(K-1)\tau}{K}$ actions outside of interval $I_1$ in phase $l + 1$. Then, the regret in this phase must be at least $\frac{(K-1)\tau}{K} \times \frac{\epsilon}{4} \geq \frac{\tau\epsilon}{8} = K^2/16$. In this case, using the assumption that the expected cumulated regret of algorithm $\pi$ over the stationary phases $\mathcal{P}_1, ..., \mathcal{P}_l$ is at least $0.01K^2l$, we find that the cumulative rewards up to phase $\mathcal{P}_{l+1}$ is at least $0.01K^2(l + 1)$, thus completing the induction step. We therefore assume henceforth that $k^* \neq 1$. Now, denoting $\mathcal{K}(\mathbb{P}, \mathbb{Q})$ the Kullback-Leibler divergence between probabilities $\mathbb{P}$ and $\mathbb{Q}$, and by $kl(p, q)$ the Kullback-Leibler divergence between Bernoulli distributions with parameters $p$ and $q$, classical arguments (see, *e.g.*, [Lattimore and Szepesvári, 2020, Chapter 15.1]) show that

$$\mathcal{K}\left(\mathbb{P}_{\mu^1}, \mathbb{P}_{\mu^{k^*}}\right) = \sum_{\tau l < t \leq \tau(l+1)} \mathbb{E}_{\mu^1}\left[kl\left(m^1\left(x_t\right), m^{k^*}\left(x_t\right)\right)\right].$$

Now, for $x_t \notin I_{k^*}$, $kl\left(m^1\left(x_t\right), m^{k^*}\left(x_t\right)\right) = 0$. Moreover, for all $x_t \in [0, 1]$, $kl\left(m^1\left(x_t\right), m^{k^*}\left(x_t\right)\right) \leq kl\left(m^1\left(\frac{2k^*-1}{K}\right), m^{k^*}\left(\frac{2k^*-1}{K}\right)\right)$. Thus,

$$\mathcal{K}\left(\mathbb{P}_{\mu^1}, \mathbb{P}_{\mu^{k^*}}\right) \leq \sum_{\tau l < t \leq \tau(l+1)} \mathbb{E}_{\mu^1}\left[\mathbb{1}\{x_t \in I_{k^*}\} kl\left(m^1\left(\frac{2k^*-1}{K}\right), m^{k^*}\left(\frac{2k^*-1}{K}\right)\right)\right]$$

$$\leq \mathbb{E}_{\mu^1}\left[N_{k^*}^{l+1}\right] kl\left(\frac{1-\epsilon}{2}, \frac{1+\epsilon}{2}\right)$$

$$\leq \frac{\tau}{K} 4\epsilon^2$$

$$\leq \frac{4K^3}{K \times 4K^2} = 1$$

where we have used the definition of $k^*$ and of $\tau$ and $\epsilon$. Now, we apply Bretagnolle-Huber Inequality (see, *e.g.*, [Lattimore and Szepesvári, 2020, Chapter 14]) to the $\mathcal{F}_{\tau(l+1)}$-measurable event $A = \{N_1^{l+1} \leq \frac{\tau}{2}\}$. This yields

$$\mathbb{P}_{\mu_1}(A) + \mathbb{P}_{\mu^{k^*}}(A^c) \geq \frac{1}{2} \exp\left(-\mathcal{K}\left(\mathbb{P}_{\mu^1}, \mathbb{P}_{\mu^{k^*}}\right)\right)$$

$$\geq \frac{1}{2} \exp(-1).$$

Using $\frac{1}{2} \exp(-1) \geq 0.18$ and $2 \max(a, b) \geq a + b$, we find that

$$\max\{\mathbb{P}_{\mu_1}(A), \mathbb{P}_{\mu^{k^*}}(A^c)\} \geq 0.09.$$

Assume that $\mathbb{P}_{\mu_1}(A) \geq 0.09$. On the event $A$, the regret on phase $l$ under reward sequence $\mu^1$ is at least $\tau/2 \times \epsilon/2 = K^2/8$. Thus, the expected regret on phase $l$ under reward sequence $\mu^1$ is at least

$0.01K^2$. Similar arguments show that if $\mathbb{P}_{\mu_{k^*}}(A^c) \geq 0.09$, the expected regret on phase $l$ under reward sequence $\mu^{k^*}$ is at least $0.01K^2$. Using the induction assumption, we find that the cumulative rewards up to phase $\mathcal{P}_{l+1}$ is at least $0.01K^2(l+1)$, thus completing the induction step.

**Conclusion.** The induction above shows that for all $K \geq 2$, there exists a sequence of reward functions $\mu_t$ with a most $\lfloor T/K^3 \rfloor$ shifts such that the total cumulative regret of algorithm $\pi$ is at least $0.01\lfloor T/K^3 \rfloor K^2$. Assuming moreover that $K < (T/2)^{1/3}$, we see that $\lfloor T/K^3 \rfloor \geq T/(2K^3)$, so that the regret is at least $0.005T/K$. We now consider the two lower bounds separately.

**Lower bound depending on $L_T$.** For $L_T \in \{0, 1, 2\}$, the bound follows from classical results on Lipschitz-continuous bandits (see, *e.g.*, Kleinberg [2004]), by noticing that since there exists a stationary problem and a constant $c > 0$ on which any bandit algorithm must have regret at least $cT^{2/3}$, there exists a problem with 0 shifts (*i.e.*, at most $L_T$) such that the regret is at least $cL^{1/3}T^{2/3}/2$. Therefore, we assume that $L_T > 2$. Then, setting $K = \lfloor (T/L_T)^{1/3} \rfloor$, we see that $K^3 \leq T/2$. Thus, the non-stationary reward function designed above has regret at least $0.005T/K \geq 0.005T^{2/3}L_T^{1/3}$. Moreover, this reward function has at most $L_T$ changes.

**Lower bound depending on $V_T$.** For $V_T \leq 2T^{-1/3}$, we have $V_T^{1/4}T^{3/4} \leq 2T^{2/3}$, the result follows from classical results on Lipschitz-continuous bandits (see, *e.g.*, Kleinberg [2004]) by noticing that there exists a stationary environment and a constant $c > 0$ on which any bandit algorithm must have regret at least $cT^{2/3}$. Therefore, we assume henceforth that $V_T > T^{-1/3}$. Then, setting $K = \lfloor (T/V_T)^{1/4} \rfloor$, we see that $K^3 \leq T/2$. Thus, the non-stationary reward function designed above has regret at least $0.005T/K \geq 0.005T^{3/4}V_T^{1/4}$. Moreover, the cumulative variation of this reward function is at most $V_T$.

# G Extensions

## G.1 Extension to multi-dimensional action space

In this subsection, we discuss how to extend our algorithm and results to multi-dimensional spaces.

**Problem setting in dimension $p$.** We assume that at each round, the player chooses an action in the set $[0, 1]^p$, where $p \in \mathbb{N}_*$ is the dimension. We assume that each mean reward function $\mu_t : [0, 1]^p \to [0, 1]$ satisfies the following Lipschitz condition:

$$\forall x, x' \in [0, 1]^p, \ |\mu_t(x') - \mu_t(x)| \leq \frac{1}{\sqrt{p}}\|x - x'\|_p,$$

where $\|\cdot\|_p$ denotes the Euclidean norm in dimension $p$.

In dimension $p$, classical results show that when the horizon is $T$ and the problem is stationary, in the worst case a minimax-optimal algorithm can learn the reward function up to an error of order $\log(T)T^{-\frac{1}{p+2}}$, and incurs a regret of order $\log(T)T^{\frac{p+1}{p+2}}$. This justifies extending the notion of significant regret to $p$-dimensional action space as follows. We say that an arm $x \in [0, 1]^p$ incurs significant regret on interval $[s_1, s_2]$ if its cumulative regret on this interval is lower bounded as

$$\sum_{t=s_1}^{s_2} \delta_t(x) \geq \log(T)(s_2 - s_1)^{\frac{p+1}{p+2}}. \tag{24}$$

Then, based on this definition, we can define the *significant shifts* and *significant phases* the same as in Definition 2.

**Adapting the algorithm.** To estimate the mean reward function in dimension $p$, we rely on a recursive $p$-adic partitioning of the action space. More precisely, we can define ***p*-adic tree** $\mathcal{T} = \{\mathcal{T}_d\}_{d \in \mathbb{N}}$ as the hierarchy of nested partitions of $[0, 1]^p$ at all possible depth $d \in \mathbb{N}$, where $\mathcal{T}_d$ denotes the partition of $[0, 1]^p$ into bins (*i.e.* hypercubes) with side length $1/2^d$.

We also adapt our doubling trick, considering **blocks** of duration $\tau_{l,m+1} - \tau_{l,m} = 2^{m(p+2)}$. Then, on the $m$-th block of the $l$-th episode, we consider as the MASTER set the bins at depths $m$. If the rewards are stable enough during this round, we expect the regret of the block to be of order $2^{m(p+1)}$.

We can use the same **sampling strategy** as presented in Section 3: given a set of active depth at time $t$, we sample a bin uniformly at the shallowest active depth, and then proceed by sampling uniformly bins among its active children at the next depth. An analysis similar to that of Proposition 2 reveals that the probability of sampling a bin at depth $d$ (given that this bin is active) is lower bounded by $2^{-pd}$. Then, the error for estimating the cumulative gaps between two bins at depth $d$ active between rounds $s_1$ and $s_2$ must be of order $\sqrt{(s_2 - s_1)2^{pd} \vee 4^{pd}} + \frac{s_2 - s_1}{2^d}$.

This motivates our considering an new **eviction criteria** of the form

$$\max_{B' \in \mathcal{B}_{[s_1, s_2]}(d)} \sum_{t=s_1}^{s_2} \hat{\delta}_t(B', B) > c_0' \left( \log(T)\sqrt{(s_2 - s_1)2^{pd} \vee 4^{pd}} + \frac{(s_2 - s_1)}{2^d} \right)$$

for some well-chosen positive numerical constant $c_0'$.

As previously, we choose as duration of a replay at depth $d$ the length of the $d$-th block of an episode. In the $p$-dimensional setting, this implies that we should schedule **replays of duration $2^{d(p+2)}$ at depth** $d$. Note that such a replay allows to estimate the cumulative gaps of two bins up to a precision $2^{d(p+1)}$.

It remains to choose the **probability of scheduling a replay** at depth $d$. At all times $s \in [\tau_{l,m}, \tau_{l,+1}[$ such that $t - \tau_{l,m} \equiv 0 \, [2^{d(p+2)}]$, we schedule a replay of duration $d <$ with probability $p_{s,d}$ given by

$$p_{s,d} = \sqrt{\frac{2^{d(p+2)}}{s - \tau_{l,m}}}.$$

**Regret analysis.** Simple computations show that there are on average $\sqrt{2^{m(p+2)}/2^{d(p+2)}}$ replays of length $2^{d(p+2)}$ during a block of length $2^{m(p+2)}$, each one with regret of order $2^{d(p+1)}$. Thus, the regret due to replays at depth $d$ over this block is of order $\sqrt{2^{m(p+2)}/2^{d(p+2)}}2^{d(p+1)} = \sqrt{2^{m(p+2)}2^{dp}}$. Summing over the depths $d < m$, we find that the total contribution of the replays over a block of length $2^{m(p+2)}$ is of order $2^{m(p+1)}$, *i.e.* of the same order as the minimax-optimal regret over this phase in the stationary setting.

To conclude, we argue that this choice of replay probability allows to detect significant shifts fast enough. Assume that the mean reward of an optimal arm undergoes a shift of magnitude $2^{-dp}$, so that it becomes sub-optimal. The algorithm should identify it as unsafe in less than $D$ rounds, where $D$ is such that $D2^{-dp} \leq 2^{m(p+1)}$, *i.e.*, $D \leq 2^{m(p+1)+dp}$. Now, our choice of $p_{s,d}$ ensures that there are approximately $\sqrt{2^{m(p+2)}/2^{d(p+2)}}$ replays at depth $d$ scheduled during the block, so that on average, a replay at depth $d$ is scheduled every $\sqrt{2^{m(p+2)}2^{d(p+2)}}$ rounds. Noticing that $\sqrt{2^{m(p+2)}2^{d(p+2)}} \leq 2^{m(p+1)+dp}$, we see that enough replays of the adequate length are schedule, so to ensure that a significant shift does not go undetected for too long.

Thus, the algorithm has almost *minimax-optimal* regret over stable phases, and detects quickly significant shifts. Conducting the same analysis as in the 1-dimensional case, we see that in dimension $p$ the regret of our algorithm should be bounded as

$$\mathbb{E}[R(\pi_{\texttt{MDBE}}, T)] \leq c \log(T)^2 \sum_{i=1}^{\tilde{L}_T} (\tau_{i+1}(\mu) - \tau_i(\mu))^{\frac{p+1}{p+2}} \, ,$$

which yields the *worst-case regret*

$$\mathbb{E}[R(\pi_{\texttt{MDBE}}, T)] \leq c' \log(T)^2 \tilde{L}_T^{\frac{1}{p+2}} T^{\frac{p+1}{p+2}} \, ,$$

for some positive numerical constant $c'$. We emphasize here that both the phases $\tau_i(\mu)$ and number of phases $\tilde{L}_T$ are based on the definition of a significant shift for $p$-dimensional actions, given in Equation (24).

## G.2 Extension to Hölder bandits

In this subsection, we discuss how to extend our algorithm and results to Hölder bandits.

**Assumption 2** (($\kappa, \beta$)-**Hölder mean reward functions**). *Each mean reward function $\mu_t$ satisfies* ($\beta \leq 1$)

$$\forall x, x' \in [0, 1], \quad |\mu_t(x) - \mu_t(x')| \leq \kappa |x - x'|^\beta .$$

**Problem setting for Hölder bandits.** We start by defining a significant regret in this setting. Arm $x \in [0, 1]$ incurs significant regret on interval $[s_1, s_2]$ if

$$\sum_{t=s_1}^{s_2} \delta_t(x) \geq \log(T)(s_2 - s_1)^{\frac{1+\beta}{1+2\beta}} \kappa^{\frac{1}{1+2\beta}} .$$

To give intuition, an oracle policy $\pi_{\text{oracle}}$ that knows when the significant shifts $\tau_i$'s discretizes the space into $K_i = (\tau_{i+1} - \tau_i)^{\frac{1}{1+2\beta}} \kappa^{\frac{2}{1+2\beta}}$ bins at each phase $[\tau_i, \tau_{i+1}[$, and incurs a regret of $\tilde{\mathcal{O}}\left( (\tau_{i+1} - \tau_i)^{\frac{1+\beta}{1+2\beta}} \kappa^{\frac{1}{1+2\beta}} \right)$ (by directly adapting results from Kleinberg [2004]), where $\tilde{\mathcal{O}}$ only hides logarithmic factors and numerical constants that does not depend on $\kappa$ or $\beta$. Thus, the dynamic regret of the oracle is upper bounded by (up to polylog factors)

$$\mathbb{E}\left[R(\pi_{\text{oracle}}, T)\right] \leq \sum_{i=1}^{\tilde{L}_T} (\tau_{i+1} - \tau_i)^{\frac{1+\beta}{1+2\beta}} \kappa^{\frac{1}{1+2\beta}}$$

$$\leq T^{\frac{1+\beta}{1+2\beta}} \tilde{L}_T^{\frac{\beta}{1+2\beta}} \kappa^{\frac{1}{1+2\beta}} .$$

We now show how to adapt MDBE so that it achieves this rate *adaptively*.

**Adapting the algorithm.** We consider blocks $[\tau_{l,m}, \tau_{l,m+1}[$ of size

$$\tau_{l,m+1} - \tau_{l,m} = 2^{m(1+2\beta)}/\kappa^2 .$$

Thus, on each block, we discretize the space into

$$(\tau_{l,m+1} - \tau_{l,m})^{\frac{1}{1+2\beta}} \kappa^{\frac{2}{1+2\beta}} = 2^m$$

bins, allowing us to consider the MASTER set as the bins at depth $m$ (that is, the set of bins $\mathcal{T}_m$ of the dyadic tree). If the rewards are stable enough during this block, we expect the regret of the block to be of order

$$R(m) = (\tau_{l,m+1} - \tau_{l,m})^{\frac{1+\beta}{1+2\beta}} \kappa^{\frac{1}{1+2\beta}} = 2^{m(1+\beta)} .$$

For the **replays**, we choose as duration of a replay at depth $d$ the length of the $d$-th block of an episode: this implies that we should schedule replays of duration

$$\ell(d) = 2^{d(1+2\beta)}/\kappa^2$$

at depth $d$, and discretize $\ell(d)^{\frac{1}{1+2\beta}} \kappa^{\frac{2}{1+2\beta}} = 2^d$ bins. Such replay allows to estimate the cumulative gaps of two bins up to precision $2^{d(1+\beta)}$.

Since we use exactly the same dyadic tree $(\mathcal{T}_d)_{1 \leq d \leq m}$ as in the 1-Lipschitz case, **the sampling strategy is exactly the same**, as well as the concentration error Proposition 2. Only the bias for estimating the cumulative reward of two bins at depth $d$ changes, and will be exactly equals to $2(s_2 - s_1)\kappa/2^{d\beta}$ over an interval $[s_1, s_2]$. This motivates a new eviction criteria depending on the Hölder constants $(\kappa, \beta)$, of the form

$$\max_{B' \in \mathcal{B}_{[s_1, s_2]}(d)} \sum_{t=s_1}^{s_2} \hat{\delta}_t(B', B) > c_0 \log(T) \sqrt{(s_2 - s_1)2^d \vee 4^d} + 4 \frac{(s_2 - s_1)\kappa}{2^{d\beta}} ,$$

where $c_0 = 7(e - 1)\sqrt{2}$ is a numerical constant.

It remains to choose the **probability of scheduling a replay** at depth $d$, at a given round. At a round $s \in [\tau_{l,m}, \tau_{l,m+1}[$ such that $s - \tau_{l,m} \equiv 0[\ell(d)]$, we schedule a replay of depth $d < m$ with probability $p_{s,d}$ given by

$$p_{s,d} = \frac{1}{\kappa} \frac{2^{\frac{d(1+2\beta)}{2}}}{\sqrt{s - \tau_{l,m}}} \quad \text{(we always have } p_{s,d} < 1 \text{ for any } d < m) .$$

**Regret analysis.** Simple computations show that there are on average $2^{\frac{(1+2\beta)(m-d)}{2}}$ replays of length $\ell(d)$ during one block $[\tau_{l,m}, \tau_{l,m+1}[$. Thus, the regret due to replays at depth $d$ over this block is of order (ignoring multiplicative numerical constants that does not depend on $\kappa$ or $\beta$)

$$R(d)2^{\frac{(1+2\beta)(m-d)}{2}} = 2^{\frac{(1+2\beta)(m-d)+2d(1+\beta)}{2}} .$$

Summing over the depths $d < m$, we find that the total contribution of the replays over a block $m$ is exactly of order $2^{m(1+\beta)}$ (up to numerical constants), *i.e.* of the same order as the minimax-optimal regret over this phase in the non-stationary setting.

Moreover, this choice of replay ensures that there are approximatively $2^{\frac{(1+2\beta)(m-d)}{2}}$ replays at depth $d$ scheduled during block $m$, so that on average, a replay at depth $d$ is scheduled every $2^{\frac{(1+2\beta)(m+d)}{2}}$ rounds. Noticing that $2^{\frac{(1+2\beta)(m+d)}{2}} \leq 2^{m(1+\beta)} \times 2^{d\beta}$, we see that enough replays of the adequate length are scheduled so to ensure that a shift of magnitude $2^{-d\beta}$ does not go undetected for too long.

Since in each episode $[t_{l+1} - t_l[$ there are at most $M_l = c\frac{\log_2((t_{l+1}-t_l)\kappa)}{1+2\beta}$, where $c$ is a numerical constant, the regret over one episode is upper bounded as

$$\sum_{m=1}^{M_l} 2^{m(1+\beta)} \leq (t_{l+1} - t_l)^{\frac{1+\beta}{1+2\beta}} \kappa^{\frac{1}{1+2\beta}} .$$

Applying exactly Proposition 6 and applying Hölder's inequality, we have exactly the upper bound of

$$\mathbb{E}\left[R(\pi_{\mathtt{MDBE}}, T)\right] \leq \widetilde{\mathcal{O}}\left(T^{\frac{1+\beta}{1+2\beta}} \tilde{L}_T^{\frac{\beta}{1+2\beta}} \kappa^{\frac{1}{1+2\beta}}\right) ,$$

where $\widetilde{\mathcal{O}}$ hides polylog factors and numerical constants that does not depend on $\kappa$ or $\beta$. We next prove that this bound is in fact minimax-optimal with respect to $T$, $\tilde{L}_T$, $\kappa$ and $\beta$ simultaneously.

**Lower Bound for non-stationary Hölder bandits.** We adapt the proof of Theorem 1 to this setting. Any policy interacting with a $(\kappa, \beta)$-Hölder bandit environment suffers regret at least $T^{\frac{1+\beta}{1+2\beta}} \kappa^{\frac{1}{1+2\beta}}$ in the stationary setting [Kleinberg, 2004]. We divide the horizon into $T/\tau$ blocks of length $\tau$. In each, we define an amount of $K = \tau^{\frac{1}{1+2\beta}} \kappa^{\frac{2}{1+2\beta}}$ $(\kappa, \beta)$ mean reward functions satisfying Assumption 2 with a single bump of size $\varepsilon = \sqrt{K/\tau} = \tau^{-\frac{\beta}{1+2\beta}} \kappa^{\frac{1}{1+2\beta}}$, hidden uniformly at random. Standard arguments show that any algorithm misidentifies the optimal region in some instance with constant probability, implying per-block regret

$$R(\pi, \tau) \geq \tau\varepsilon = \tau^{\frac{1+\beta}{1+2\beta}} \kappa^{\frac{1}{1+2\beta}} .$$

Concatenating $T/\tau$ such blocks (with reward functions chosen independently) gives total regret of

$$(T/\tau) \times R(\pi, \tau) \geq T\tau^{-\frac{\beta}{1+2\beta}} \kappa^{\frac{1}{1+2\beta}} .$$

To derive a lower bound that depends on $\tilde{L}_T$, we first set $\tau = T/L_T$. This yields $R(\pi, T) \geq T^{\frac{1+\beta}{1+2\beta}} L_T^{\frac{\beta}{1+2\beta}} \kappa^{\frac{1}{1+2\beta}}$, and we observe that we always have $L_T \geq \tilde{L}_T$. For the dependency on $V_T$, we set

$$\tau = \kappa^{\frac{1}{1+3\beta}} T^{\frac{1+2\beta}{1+3\beta}} V_T^{-\frac{1+2\beta}{1+3\beta}} .$$

It is easy to verify that the total variation over $T$ is exactly $V_T$. So for any algorithm $\pi$, there exist an environment such that

$$R(\pi, T) \geq T^{\frac{1+\beta}{1+2\beta}} \widetilde{L}_T^{\frac{\beta}{1+2\beta}} \kappa^{\frac{1}{1+2\beta}} \wedge T^{\frac{1+\beta}{1+2\beta}} + T^{\frac{1+2\beta}{1+3\beta}} V_T^{\frac{\beta}{1+3\beta}} \kappa^{\frac{1}{1+3\beta}} .$$

We recover our results for $\kappa = 1$ and $\beta = 1$. As the smoothness of the reward functions increases ($\beta \to 1$ and/or $\kappa \to 0$), the problem becomes easier, as the lower bound decreases.

# H Computational complexity of `MDBE`

While our algorithm is feasible on small-scale problems (as showed in our experiments in Appendix I), we acknowledge that its computational complexity may limit scalability in large-scale applications. Below, we provide a more precise characterization of the computational cost.

At each round within a block of length $\tau_{l,m+1} - \tau_{l,m} = 8^m$, the algorithm maintains estimates for all bins across all discretization depths $d = 1, \ldots, m$. The total number of bins at depth $d$ is $2^d$, and across all depths, we maintain a total of

$$\text{NumberBins}_m = \sum_{d=1}^{m} 2^d = \mathcal{O}(2^m).$$

For each bin, we compute an importance-weighted mean estimate, resulting in a per-round cost of $\mathcal{O}(2^m)$ for estimation alone. In addition, at each round $t$ of this block, the algorithm performs a statistical test ($\star$) over *all* pairs of bins $(B_1, B_2)$ at each depth $d \in [1, m]$. There are $\binom{2^d}{2} = \mathcal{O}(4^d)$ such pairs per depth, and summing over all depths yields

$$\text{TotalPairs}_m = \sum_{d=1}^{m} \mathcal{O}(4^d) = \mathcal{O}(4^m)$$

per block. For each pair $(B_1, B_2)$, we must consider all possible intervals $[s_1, t] \subseteq [\tau_{l,m}, t]$ during the block. The number of such intervals is $\mathcal{O}(8^m)$ in the worst case.

Putting it all together, the worst-case time complexity for each round is

$$\mathcal{O}(2^m) + \mathcal{O}(4^m \cdot 8^m) = \mathcal{O}(32^m),$$

and the memory complexity is $\mathcal{O}(8^m)$. Since the number of block is upper bounded as $m \leq \log(T)$, we conclude that the worst-case time computational complexity of our algorithm is $\mathcal{O}(T^6)$ and its worst-case memory complexity is $\mathcal{O}(T^4)$, which are both **polynomial in $T$.**

While the computational cost per block is manageable for small values of $m$ (e.g., $m \leq 4$), it becomes quickly intractable as $m$ increases. Developing efficient **adaptive algorithms** that enjoy minimax optimal regret is therefore an important direction for future work, even in the $K$-armed setting.

# I Numerical simulations

In this section, we illustrate some numerical experiments to show the empirical performances of `MDBE` on a synthetic dataset. The code for these implementations is available at https://github.com/nguyenicolas/NS_Lipschitz_Bandits.

**Environment.** We simulate a 1-Lipschitz, piecewise-linear reward function defined over the action space $[0, 1]$, with a single peak shifting smoothly from $x = 0.3$ to $x = 0.7$ every $10^5$ rounds. This setup induces $\tilde{L}_T = 10$ significant shifts over a time horizon $T = 10^6$. Thus, the mean reward changes every round, but only ten of these changes are significant under our framework.

**Benchmarks.** We compare our method against two baselines: `BinningUCB (naive)` and `BinningUCB (oracle)`. The first baseline assumes knowledge only of the total time horizon $T$ and naively discretizes the action space into $K(T) \propto T^{1/3}$ actions, as if operating in a stationary environment of length $T$. It then runs a standard UCB algorithm without resetting its statistical estimates. The `oracle` baseline, by contrast, has access to the *exact* times of the significant shifts $\tau_i$'s and resets its estimates at each significant phase, using the optimal per-phase discretization $K_i \propto (\tau_{i+1} - \tau_i)^{1/3}$.

**Results.** We report in Figure 4 the cumulative dynamic regret of the three methods, averaged over 100 independent runs. Our method (`MDBE`) significantly outperforms `BinningUCB (naive)` by adapting to non-stationarities through replay mechanisms.

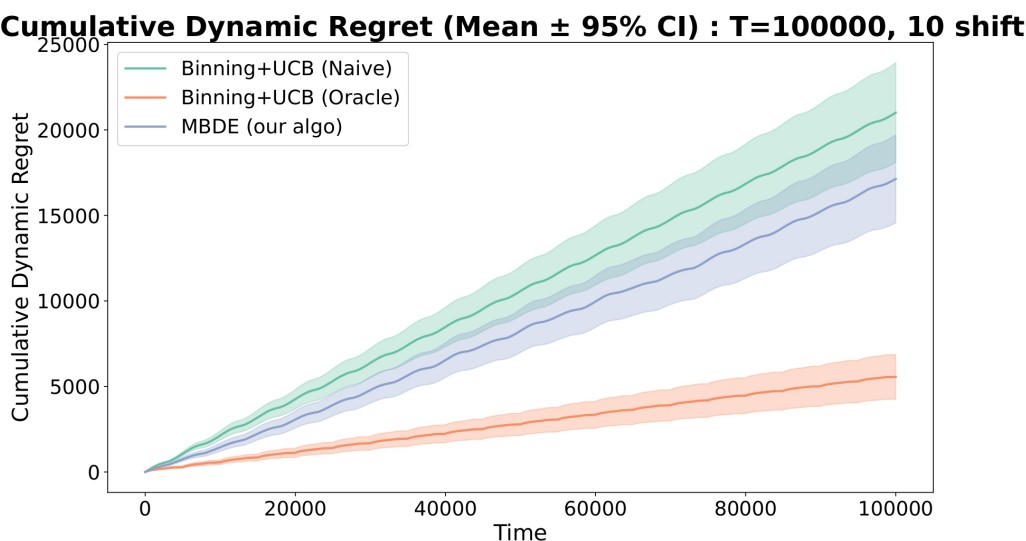

Figure 4: Cumulative dynamic regret of `MDBE`, `BinningUCB (naive)`, and `BinningUCB (oracle)` over a total horizon of $T = 10^6$ rounds with 10 significant shifts. Results are averaged over 100 independent runs, with 95% confidence intervals of the mean dynamic regret shown.

