# OpenReview forum: "Non-Stationary Lipschitz Bandits"
_NeurIPS.cc/2025/Conference — NeurIPS 2025 poster_

### Official Review · Reviewer_GmBd · 2025-06-24

**Clarity:** 3
**Significance:** 2
**Originality:** 2
**Rating:** 5
**Confidence:** 4

**Summary:**

This paper studies non-stationary Lipschitz bandits and proves the first minimax optimal dynamic regret bounds in terms of a new non-stationary metric, "significant shifts", which follows a similar line of results in the non-stationary K-armed bandits.

**Questions:**

Questions/Suggestions:
1. Why is there a log(T) factor in Definition 1? Is this because of concentration bounds or does it optimize the log dependence in the regret bound?
2. In the pseudocode of Section B, why reset the master set at each block instead of carrying over information about eliminated regions from previous block to current block?
3. Again, I believe a diagram or illustrative example would be helpful to explain the bin selection mechanism and replay scheduling.
4. The dependence on the Lipschitz constant in the regret bound and other generalities such as Holder smooth bandits are missing, some discussion on these extensions would make the paper more thorough.
5. What is the computational complexity of the algorithm and how does it compare to other algorithms for the non-stationary setting?

Typos:
1.  "almost" in Line 979
2. Line 596: "discretion"

**Ethical Concerns:**

["NO or VERY MINOR ethics concerns only"]

**Final Justification:**

Authors adequately addressed all my concerns.

**Limitations:**

Yes

**Quality:**

3

**Strengths And Weaknesses:**

Strengths:
1. The first minimax optimal regret bound for non-stationary Lipschitz bandits with both proven lower and upper bounds.
2. New techniques involving multi-depth binning strategy for Lipschitz bandits.

Weaknesses:
1. Lack of experimental validation of the MDBE algorithm.
2. I think Section 3 is quite heavy in notation and can be better served with a diagram/picture and example to illustrate the process of choosing a depth/arm.
3. Much of the analysis seems structured similarly to the previous works on learning significant shifts. In particular, the work of Suk and Kpotufe [2023] also uses a multi-depth adaptive discretization idea to address the problem of learning the right discretization level, although it is for a finite-armed problem. The authors could better highlight what are the key differences or technical challenges addressed over the Lipschitz contextual bandit setting.

---

> ### Author Rebuttal · Authors · 2025-07-30
>
> We thank the reviewer for the careful analysis.
>
> ### **Differences with *Suk and Kpotufe (2023)***
> While it is true that *Suk and Kpotufe (2023)* also operate in blocks and discretize the context space accordingly, their estimation strategy **does not involve multi-scale replays**. Once the context is discretized and observed, their analysis focuses on estimating the mean rewards of a **fixed and finite set** of $K$ actions, using replays of these same actions. Our algorithm must face three fundamental challenges: ***(i)* scheduling** (potentially simultaneous) replays at different depths, ***(ii)* leveraging the information collected across different scales**, and ***(iii)* tracking the regret contribution from all scales simultaneously**. In our regret analysis, each action $x_t \in [0,1]$ belongs to several bins across discretization levels, and we explicitly quantify the regret incurred at each scale. Moreover, to decide which bin to activate at a given depth and time step, we introduce a principled strategy for ***(a)* scheduling replays at specific scales** and ***(b)* selecting the appropriate bin at a given active depth** through a carefully designed sampling algorithm.
>
> We will incorporate this discussion at the beginning of the proof sketch in Section 5 to clarify the novelty and technical depth of our theoretical contribution.
>
> ### **$\log(T)$ in the definition of significant shift**
> While it is true that previous works do not include a logarithmic factor in the definition of a significant shift (*e.g.* *Suk and Kpotufe 2022)*, we observe that incorporating a $\log(T)$ term has no impact on the regret analysis, while leading to a **strictly sharper metric**: the number of significant shifts can only decrease with the added log factor, making the metric sharper.
>
> ### **On restarts at each block**
> Restarting estimates at the beginning of each block is primarily a **theoretical simplification** that facilitates a clean application of the doubling trick (see line 625 of our supplementary material). This strategy simplifies the regret decomposition and analysis, at the expense of only a constant numerical factor in the regret bound.
>
> ### **Presentation**
> We thank the reviewer for pointing out this presentation issue. If the paper is accepted, we will use the additional allowed page in the supplementary material to include visual illustrations of the dyadic tree structure in the main text to improve clarity.
>
> ### **Extension to smooth bandits (same answer as `scsD` and `kQLa`)**
> **Our algorithm and analysis can easily be adapted beyond $1$-Lipschitz reward function, and still enjoy minimax optimal rate**. For the sake of space, we give a sketch of proof of this generalization (similarly to *Appendix G* for the multi-dimensional setting).
>
> Our method generalizes to **$(\kappa, \beta)$-Hölder mean rewards** $(\mu_t)_t$ satisfying:
>
> $$\forall x,x'\in[0,1],\quad |\mu_t(x)-\mu_t(x')| \leq \kappa|x - x'|^\beta.$$
> In the stationary setting with horizon $\tau$, the minimax regret is $\tau^{\frac{1+\beta}{1+2\beta}} \kappa^{\frac{1}{1+2\beta}}$, achieved by discretizing into $K_\text{opt}(\tau,\kappa,\beta) = \tau^{\frac{1}{1+2\beta}}\kappa^{\frac{2}{1+2\beta}}$ bins *(Kleinberg, 2004)*. This motivates **defining significant regret** over $[s_1,s_2]$ as:
>
> $$\sum_{t=s_1}^{s_2} \delta_t(x) \geq \log(T)(s_2 - s_1)^{\frac{1+\beta}{1+2\beta}} \kappa^{\frac{1}{1+2\beta}}.$$
> We consider blocks of size $\tau_{l,m+1}-\tau_{l,m}=2^{m(1+2\beta)}/\kappa^2$, yielding $2^m$ bins and expected regret of $2^{m(1+\beta)}$ over this block if the rewards are stable enough. Replays at depth $d$ last $\ell(d) = 2^{d(1+2\beta)}/\kappa^2$ and use $2^d$ bins. The bias for estimating the cumulative reward gap is $4(s_2 - s_1)\kappa/2^{d\beta}$. This motivates the new **eviction rule**:
>
> $$\max_{B'}\sum_{t=s_1}^{s_2} \hat\delta_t(B',B) > c_0\log(T)\sqrt{(s_2-s_1)2^d} + 4\kappa(s_2 - s_1)/2^{d\beta},$$
> where the $c_0\approx 20$ is exactly the same as in the 1-Lipschitz setting.
>
> Replay at depth $d$ starting at round $s$ is scheduled with **probability**:
>
> $$p_{s,d} = \frac{1}{\kappa} \cdot \frac{2^{d(1+2\beta)/2}}{\sqrt{s - \tau_{l,m}}} \quad (d < m).$$
> Each block includes $2^{(1+2\beta)(m-d)/2}$ replays at depth $d$, each incurring regret $2^{((1+2\beta)(m-d) + 2d(1+\beta))/2}$. Summing over all dephs $d < m$ gives total replay regret $\mathcal{O}(2^{m(1+\beta)})$, matching the optimal bound per block.
>
> The expected replay interval at depth $d$ is (up to numerical constants that do not depend on $\kappa$ and $\beta$) $2^{(1+2\beta)(m+d)/2}$, while shifts of size $2^{-d\beta}$ become detectable within $2^{m(1+\beta)}$ rounds. Each episode $t_{l+1}-t_l$ has at most $M_l = \log_2((t_{l+1}-t_l)\kappa)/(1+2\beta)$ blocks, and the regret per episode is upper bounded as
>
> $$\sum_{m=1}^{M_l} 2^{m(1+\beta)} \leq (t_{l+1}-t_l)^{\frac{1+\beta}{1+2\beta}} \kappa^{\frac{1}{1+2\beta}}.$$
> Thus, the total dynamic regret of $\texttt{MBDE}$ is upper bounded as:
>
> $$\mathbb{E}[R_T] \leq \tilde{\mathcal{O}}\left(T^{\frac{1+\beta}{1+2\beta}} \tilde{L}_T^{\frac{\beta}{1+2\beta}} \kappa^{\frac{1}{1+2\beta}} \right).$$
> $ \tilde{\mathcal{O}}(\cdot)$ only hides poly-logarithmic factors and numerical constant **that does not depend on $\kappa$ or $\beta$**.
>
> With the exact same arguments as in our Lower bound proof  *(Appendix F)*, we show that **this rate is indeed minimax-optimal with respect to $T, \tilde{L}_T, \kappa$ and $\beta$**.
>
> ### **Numerical results (same answer as `VifP`, `kQLa` and `9z8n`)**
> **We implemented our algorithm and evaluated its performance**. Because of the NeurIPS policy, we cannot share code at this stage but will release it upon acceptance.
>
> We benchmark against two baselines: $\texttt{BinningUCB (no restart)}$, which uses $K(T)\propto T^{1/3}$ and never resets, and $\texttt{BinningUCB (oracle)}$, which **knows** the true change points $\tau_i$ and resets its estimate at the end phase. At each phase, it uses the optimal per-phase discretization $K_i \propto (\tau_{i+1}-\tau_i)^{1/3}$.
>
> We simulate a $1$-Lipschitz, piecewise-linear reward on $[0,1]$ with a peak shifting from $x=0.3$ to $x=0.7$ every $10^5$ rounds over horizon $T=10^6$, inducing $\tilde{L}_T = 10$ significant shifts. Regret is averaged over $100$ runs (rounded to nearest integer).
>
> | **Step (×10³)** | $\texttt{BUCB (no restart)}$ | $\texttt{BUCB (oracle)}$ | **$\texttt{MBDE}$ (ours)**    |
> |----------------:|-------------------------:|---------------------:|---------------------:|
> | 10              | 2137 ± 101               | 551 ± 30             | **1566 ± 108**        |
> | 50              | 10580 ± 393              | 2794 ± 147           | **8556 ± 349**        |
> | 100             | 21007 ± 890              | 5596 ± 396           | **17111 ± 667**       |
>
> Our method adapts to non-stationarity via replays, significantly outperforming $\texttt{BUCB (no restart)}$.
>
> ### **Computational complexity (same answer as `VifP`, `kQLa` and `9z8n`)**
> Our algorithm is feasible for small-scale problems, as shown in experiments, but we acknowledge that its time and memory complexity limits its scalability. We detail the worst-case computational cost below.
>
> In a block of length $\tau_{l,m+1}-\tau_{l,m}=8^m$, the algorithm maintains $\sum_{d=1}^m 2^d = \mathcal{O}(2^m)$ bins. Estimating bin means costs $\mathcal{O}(2^m)$ per round. Additionally, it performs the statistical test $\textcolor{red}{(\star)}$ over all bin pair of bins $(B_1, B_2)$ at each depth $d$, with $\mathcal{O}(4^d)$ pairs, totaling $\mathcal{O}(4^m)$ across depths. For each pair, it checks all intervals of the form $[s_1, t] \subseteq [\tau_{l,m}, t]$, leading to $\mathcal{O}(8^m)$ possibilities. Therefore, the total *per-round* worst-case time complexity is $\mathcal{O}(2^m + 4^m \cdot 8^m) = \mathcal{O}(32^m)$, and the *per-round* worst-case memory complexity is $\mathcal{O}(8^m)$.
>
> Since $m \leq \log(T)$, we conclude that $\texttt{MBDE}$ has an overall **worst-case time complexity $\mathcal{O}(T^6)$** and **memory $\mathcal{O}(T^4)$**: both **polynomial in $T$**. For small $m$ (e.g., $m \leq 5$), runtime is manageable for some problem instances. Designing adaptive, efficient algorithms with optimal regret remains an important open direction, even in $K$-armed settings *(Gerogiannis et al., 2024)*.
>
> We will add a paragraph discussing the computational complexity in the main text.
>
> ### **References**
> - *Kleinberg. "Nearly tight bounds for the continuum-armed bandit problem." NIPS 2004.*
> - *Suk and Kpotufe. "Tracking most significant shifts in nonparametric contextual bandits." Neurips 2023.*
> - *Gerogiannis, Huang and Veeravalli. Is Prior-Free Black-Box Non-Stationary Reinforcement Learning Feasible? AISTATS 2025.*

---

> > ### Comment · Reviewer_GmBd · 2025-08-05
> >
> > Thank you for your response. All my concerns are adequately addressed and so I raise my score to accept.

---

### Official Review · Reviewer_9z8n · 2025-07-03

**Clarity:** 3
**Significance:** 3
**Originality:** 4
**Rating:** 5
**Confidence:** 4

**Summary:**

This manuscript introduces the first algorithm with optimal regret guarantees for continuum-armed bandits
under time-varying, Lipschitz-continuous reward functions. The key idea is to focus on significant shifts,
which meaningfully affect regret. The proposed algorithm, Multi-Depth Bin Elimination (MDBE), adaptively
explores a dyadic partition of the action space and uses randomized replays to detect shifts and prune
suboptimal regions. Without knowing the number or location of changes, MDBE achieves minimax-optimal
regret. However, there are several aspects that should be addressed. My main concern lies
in the comparison with prior work and the justification for adopting the new metric. The detail comments
are listed below.

**Questions:**

1. In the related work part, when the authors are comparing [Suk and Kim 2025], it is mentioned that
[Suk and Kim 2025] assumes no regularity across arms thus it is not applicable to this manuscript.
This statement seems too vague for me. I suggest the authors make more efforts on elaborating the key
difference between this paper and prior works.
2. The notation may always be confusing. The variable x in this manuscript is denoted as the arm/action.
However, in the literature of bandit, x is denoted as the context and a is denoted as the arm/action.
I would suggest the authors make some efforts to change it, though it will not take any effect on my
scoring.
3. In definition 2, it is confusing whether [s1, s2] ⊆ [τi−1, τi
] depends with x or it exists for all x.
4. In the same time, I would like to ask the authors to compare the difference in the definition of significant
shit with [Suk and Kim 2025], which is not cited in Definition 2.
5. In the meantime, I also find
6. For the Section 2.2, though mathematically the comparison makes sense, I would also want to see a
concrete example to show using LeT metric is statistically and significantly more efficient. This would
strengthen the motivation for adopting LeT and also shows the bounds in Section 4 can be tighter than
other metrics.
7. In the proof of Proposition 3, the notation ft−1 should also indicate the condition xt ∈ B. Thus, ft−1,B
is a better representation.

**Ethical Concerns:**

["NO or VERY MINOR ethics concerns only"]

**Final Justification:**

After reading the discussion and the rebuttal, I decided to update my score.

**Limitations:**

Adequate.

**Quality:**

3

**Strengths And Weaknesses:**

Strengths
	1.	Addresses a previously unstudied intersection of settings:
The paper considers non-stationary bandits with a continuous (Lipschitz) action space, a combination that had not been analyzed in depth before. While both components—non-stationarity and Lipschitz structure—are individually well studied, their interaction introduces new technical challenges that are not addressed by existing work.
	2.	Technically non-trivial adaptation of known methods:
Although one might imagine combining known ideas—such as time-scale decomposition (as in Wang or Besbes–Zeevi) with Kleinberg-style discretization—this paper shows that a naïve hybrid is insufficient. The core difficulty lies in adaptively detecting global, cumulative changes over a continuous space without knowing when or how these changes occur. The algorithm’s hierarchical, replay-based design and multi-scale gap estimation address this challenge in a subtle and well-justified way.
	3.	Theoretical guarantees match minimax lower bounds:
The MDBE algorithm achieves a regret bound of \widetilde{O}(\tilde{L}_T^{1/3} T^{2/3}), matching the minimax lower bound up to logarithmic factors. Crucially, this is done without requiring prior knowledge of the number or locations of significant shifts. The framework is also robust to small or inconsequential fluctuations, making the bound potentially sharper than those based on classical variation metrics.

Weaknesses
Summary of the Paper: “Non-Stationary Lipschitz Bandits”

This paper investigates the problem of non-stationary bandits with a continuous action space and Lipschitz reward functions that can change arbitrarily over time. Unlike traditional non-stationary bandit approaches, which often assume a discrete action set and require prior knowledge of the non-stationarity level, this work addresses the more challenging setting of an infinite arm space under Lipschitz continuity without any such assumptions.

The authors propose a novel algorithm, MDBE (Multi-Depth Bin Elimination), which adaptively tracks a refined notion of non-stationarity called significant shifts—large aggregate changes in regret that are statistically meaningful. MDBE uses a hierarchical discretization of the action space and a replay mechanism to detect these significant shifts at different resolutions efficiently.

The main result is that MDBE achieves minimax-optimal dynamic regret of order \widetilde{O}(\tilde{L}^{1/3} T^{2/3}), where \tilde{L} is the number of significant shifts and T is the time horizon. This regret bound matches known lower bounds and does not rely on any prior knowledge of the number or timing of shifts. The paper provides theoretical analysis, a detailed algorithmic design, and high-probability guarantees.

⸻

Strengths:
	1.	Novelty and Scope:
The paper pioneers the study of non-stationary bandits in a continuous action setting under Lipschitz continuity—a previously unexplored combination—and introduces a theoretically grounded notion of significant shifts to this context.
	2.	Strong Theoretical Guarantees:
The algorithm achieves minimax-optimal regret bounds without knowing the degree of non-stationarity in advance. This is a substantial theoretical contribution that strengthens the relevance of the proposed method.
	3.	Sophisticated Algorithm Design:
MDBE incorporates a multi-resolution replay mechanism and hierarchical bin elimination, reflecting a careful treatment of exploration vs. exploitation in a non-stationary continuous setting. The sampling and bin eviction strategies are technically sound and rigorously justified.

⸻

Weaknesses:
	1.	Lack of Empirical Validation:
The paper is entirely theoretical and does not provide any simulations or experiments to demonstrate how the algorithm performs in practice. This limits its immediate impact for practitioners.
	2.	Complexity of the Algorithm:
MDBE’s design is intricate, involving nested partitions, probabilistic replays, and importance-weighted estimators. While theoretically elegant, the practicality and implementation complexity may hinder adoption.
	3.	Computational Concerns Unaddressed:
The authors acknowledge that computational efficiency is not considered. Given the algorithm’s multi-scale structure, it is unclear whether MDBE is computationally feasible for large-scale or real-time applications.

---

> ### Author Rebuttal · Authors · 2025-07-30
>
> We thank the reviewer for the feedback.
>
> ### **Numerical results (same answer as `VifP`, `kQLa` and `GmBd`)**
> **We implemented our algorithm and evaluated its performance**. Because of the NeurIPS policy, we cannot share code at this stage but will release it upon acceptance.
>
> We benchmark against two baselines: $\texttt{BinningUCB (no restart)}$, which uses $K(T)\propto T^{1/3}$ and never resets, and $\texttt{BinningUCB (oracle)}$, which **knows** the true change points $\tau_i$ and resets its estimate at the end phase. At each phase, it uses the optimal per-phase discretization $K_i \propto (\tau_{i+1}-\tau_i)^{1/3}$.
>
> We simulate a $1$-Lipschitz, piecewise-linear reward on $[0,1]$ with a peak shifting from $x=0.3$ to $x=0.7$ every $10^5$ rounds over horizon $T=10^6$, inducing $\tilde{L}_T = 10$ significant shifts. Regret is averaged over $100$ runs (rounded to nearest integer).
>
> | **Step (×10³)** | $\texttt{BUCB (no restart)}$ | $\texttt{BUCB (oracle)}$ | **$\texttt{MBDE}$ (ours)**    |
> |----------------:|-------------------------:|---------------------:|---------------------:|
> | 10              | 2137 ± 101               | 551 ± 30             | **1566 ± 108**        |
> | 50              | 10580 ± 393              | 2794 ± 147           | **8556 ± 349**        |
> | 100             | 21007 ± 890              | 5596 ± 396           | **17111 ± 667**       |
>
> Our method adapts to non-stationarity via replays, significantly outperforming $\texttt{BUCB (no restart)}$.
>
> ### **Computational complexity (same answer as `VifP`, `kQLa` and `GmBd`)**
> Our algorithm is feasible for small-scale problems, as shown in experiments, but we acknowledge that its time and memory complexity limits its scalability. We detail the worst-case computational cost below.
>
> In a block of length $\tau_{l,m+1}-\tau_{l,m}=8^m$, the algorithm maintains $\sum_{d=1}^m 2^d = \mathcal{O}(2^m)$ bins. Estimating bin means costs $\mathcal{O}(2^m)$ per round. Additionally, it performs the statistical test $\textcolor{red}{(\star)}$ over all bin pair of bins $(B_1, B_2)$ at each depth $d$, with $\mathcal{O}(4^d)$ pairs, totaling $\mathcal{O}(4^m)$ across depths. For each pair, it checks all intervals of the form $[s_1, t] \subseteq [\tau_{l,m}, t]$, leading to $\mathcal{O}(8^m)$ possibilities. Therefore, the total *per-round* worst-case time complexity is $\mathcal{O}(2^m + 4^m \cdot 8^m) = \mathcal{O}(32^m)$, and the *per-round* worst-case memory complexity is $\mathcal{O}(8^m)$.
>
> Since $m \leq \log(T)$, we conclude that $\texttt{MBDE}$ has an overall **worst-case time complexity $\mathcal{O}(T^6)$** and **memory $\mathcal{O}(T^4)$**: both **polynomial in $T$**. For small $m$ (e.g., $m \leq 5$), runtime is manageable for some problem instances. Designing adaptive, efficient algorithms with optimal regret remains an important open direction, even in $K$-armed settings *(Gerogiannis et al., 2024)*.
>
> We will add a paragraph discussing the computational complexity in the main text.
>
>
> ### **Differences with *Suk and Kim (2025)***
> *Suk and Kim (2025)* study the non-stationary version of infinite-armed bandits with **reservoir distribution**: at each round $t$, the learner selects an action $a_t \in [0,1]$ and observes a noisy reward $Y_t(a_t)$ with mean $\mu_t(a_t)$, where $\mu_t(a_t)$ itself is drawn from a time-varying distribution $\nu_t(a_t)$. Crucially, this reservoir distribution $\nu_t(a_t)$ evolves over time depending on the learner’s past actions. Their primary assumption concerns the proportion of arms with low expected rewards, but they do **not impose any smoothness or regularity** conditions on the reward function $\mu_t(\cdot)$. In other words, **similar actions do not necessarily yield similar expected rewards**, so there is no structure to exploit.
>
> This lack of smoothness means that exploration strategies cannot benefit from generalizing across nearby arms, **and thus do not employ discretization of the action space**. Instead, their algorithms rely on *subsampling techniques* tailored to handle the complexity of the reservoir distributions. Due to these differences, their setting and methods relate more closely to *rotting bandits* literature, which deals with non-stationary rewards that evolve over time in ways **independent of spatial smoothness**.
>
> We will make it clear in our related work subsection.
>
> ### **Notations for actions**
> While we agree that in $K$-armed contextual bandit settings, it is standard to denote the context by $x$ and the action by $a$, this convention does not necessarily carry over to the Lipschitz bandits literature. In fact, denoting actions by $x$ is common in this litterature, *e.g.* in *Bubeck et al. (2011)* and *Kleinberg et al. (2019)*.
>
> ### **Clarifications on the definition of significant shift**
> - The interval $[s_1, s_2[$ **depends** on $x$: each arm may incur significant regret over a different interval. A *significant shift* is said to occur when **all** arms have experienced such significant regret over their respective intervals.
> - This notion of significant regret is indeed derived from previous work by Suk and coauthors: across all settings (including ours), it refers to an interval over which the regret of an arm **exceeds the minimax rate for that setting**. In our case, this threshold is $\tilde{O}(t^{1/3})$. Consequently, the definition of a significant shift remains consistent across all settings that adopt this significant shift framework.
>
> ### **Concrete benefit of the significant shift metric (same answer as `VifP`)**
> Consider a recommendation system with a continuous pool of content (e.g., movies), indexed by $x \in [0,1]$, where nearby values of $x$ correspond to similar content types. Suppose there exists two regions of high and comparable user preference, centered around $x_1 = 0.3$ and $x_2 = 0.7$. Imagine a scenario where preferences near $x_1$ remain stable over time (*e.g*., a timeless classic), while preferences near $x_2$ undergo very frequent changes (*e.g*., a trending topic that evolves daily).
>
> In this case, an algorithm that **consistently recommends content near $x_1$** would incur **little to no regret, even though the underlying reward function changes frequently** in other regions. From a global perspective, the *number of changes* or the *total variation in mean reward* could be **as large as $L_T = V_T= \mathcal{O}(T)$**. An algorithm that relies solely on such metrics would unnecessarily restart its estimates too frequently, as it would **overestimate** the effective difficulty of the problem. This leads to both theoretical suboptimality and practical inefficiency. However, under our framework, such changes do **not** constitute a significant shift. This example shows a key strength of our metric: it captures only those changes that are **statistically consequential** to learning, rather than indiscriminately counting all shifts in the environment.
>
> We will include this example at the end of *Section 2.2* to further clarify the motivation for our definition.
>
>
> ### **Notation $f_{t-1, B}$**
> True, thank you for pointing this!
>
> ### **References**
> - *Bubeck, Munos, Stoltz, and Szepesvári (2011). X-Armed Bandits. JMLR.*
> - *Kleinberg, Slivkins, and Upfal. Bandits and experts in metric spaces (2019). Journal of the ACM.*
> - *Suk and Kim. Tracking most significant shifts in infinite-armed bandits. ICML 2025.*

---

> > ### Comment · Reviewer_9z8n · 2025-08-01
> > **Thanks.**
> >
> > Thank you for the responses. I maintain my positive view of the paper.

---

### Official Review · Reviewer_kQLa · 2025-07-03

**Clarity:** 3
**Significance:** 3
**Originality:** 3
**Rating:** 5
**Confidence:** 2

**Summary:**

This paper studies the problem of non-stationary Lipschitz bandits, which generalizes both non-stationary bandits and Lipschitz bandits by allowing the reward function to evolve over time while satisfying a Lipschitz condition. The key contribution is MDBE (Multi-Depth Bin Elimination), a novel algorithm that adaptively adjusts the discretization level. The paper establishes the first optimal dynamic regret guarantees for non-stationary bandits with continuous action spaces under Lipschitz continuity.

**Questions:**

What is the potential for broader applicability to other non-stationary settings of the adaptive discretization strategy? For example, do non-stationary smooth bandits have similar challenges in discretization? If yes, would the strategy proposed in this work be applicable to that problem?

You clearly summarize how your algorithmic contributions relate to prior work. However, the novelty of the proof techniques is less explicitly highlighted. A brief discussion of what is technically new in the analysis would help clarify the theoretical contribution.

**Ethical Concerns:**

["NO or VERY MINOR ethics concerns only"]

**Final Justification:**

The authors have addressed most of my questions during rebuttal, including numerical results for validation, so I raised the score to a clear accept.

**Limitations:**

Yes.

**Paper Formatting Concerns:**

No major formatting issues.

**Quality:**

3

**Strengths And Weaknesses:**

Strengths:

This paper is very well-written, with a clear motivation for studying non-stationary Lipschitz bandits as a natural generalization of both non-stationary and Lipschitz bandits with potential real-world applications. The proposed algorithm, MDBE, is novel and thoughtfully designed to address the key challenges of adaptivity and continuous action spaces under unknown non-stationarity. The theoretical results are strong. Moreover, the paper does a good job positioning its contribution relative to prior work, especially in how it extends the notion of significant shifts and adapts replay-based elimination techniques to the continuous setting.

Weaknesses:

The paper introduces a three-way trade-off in the introduction, but it's unclear what the three ways are.

Some technical terms, such as “dyadic tree”, may be unfamiliar to a broader audience and could benefit from a more intuitive or visual explanation.

Notations in some key places, such as d in Proposition 1 or δ in Definition 4, could be clarified within the theorem statement to improve readability.

While the paper is theoretically focused, a small simulation or experiment could help validate the regret bounds and provide intuition about the algorithm’s behavior in practice.

---

> ### Author Rebuttal · Authors · 2025-07-30
>
> We thank the reviewer for the feedback.
>
> ### **Three-way tradeoff (same answer as `scsD`)**
> The *three-way tradeoff* refers to the balance our algorithm must deal between: ***(i)* exploration**, via replays to detect changes; ***(ii)* exploitation**, by choosing bins with the highest estimated rewards; and ***(iii)* discretization level**, that is, choosing the appropriate discretization depth at each time step. This third dimension arises specifically from our setting of Lipschitz mean reward functions that change over time. We will clarify this in Section 1.1.
>
> ### **Presentation and readability**
> We thank the reviewer for pointing the flaws in our presentation. We will add figures to enhance comprehension in the supplementary allowed page if our paper is accepted.
>
> ### **Numerical results (same answer as `VifP`, `9z8n` and `GmBd`)**
> **We implemented our algorithm and evaluated its performance**. Because of the NeurIPS policy, we cannot share code at this stage but will release it upon acceptance.
>
> We benchmark against two baselines: $\texttt{BinningUCB (no restart)}$, which uses $K(T)\propto T^{1/3}$ and never resets, and $\texttt{BinningUCB (oracle)}$, which **knows** the true change points $\tau_i$ and resets its estimate at the end phase. At each phase, it uses the optimal per-phase discretization $K_i \propto (\tau_{i+1}-\tau_i)^{1/3}$.
>
> We simulate a $1$-Lipschitz, piecewise-linear reward on $[0,1]$ with a peak shifting from $x=0.3$ to $x=0.7$ every $10^5$ rounds over horizon $T=10^6$, inducing $\tilde{L}_T = 10$ significant shifts. Regret is averaged over $100$ runs (rounded to nearest integer).
>
> | **Step (×10³)** | $\texttt{BUCB (no restart)}$ | $\texttt{BUCB (oracle)}$ | **$\texttt{MBDE}$ (ours)**    |
> |----------------:|-------------------------:|---------------------:|---------------------:|
> | 10              | 2137 ± 101               | 551 ± 30             | **1566 ± 108**        |
> | 50              | 10580 ± 393              | 2794 ± 147           | **8556 ± 349**        |
> | 100             | 21007 ± 890              | 5596 ± 396           | **17111 ± 667**       |
>
> Our method adapts to non-stationarity via replays, significantly outperforming $\texttt{BUCB (no restart)}$.
>
> ### **Computational complexity (same answer as `VifP`, `9z8n` and `GmBd`)**
> Our algorithm is feasible for small-scale problems, as shown in experiments, but we acknowledge that its time and memory complexity limits its scalability. We detail the worst-case computational cost below.
>
> In a block of length $\tau_{l,m+1}-\tau_{l,m}=8^m$, the algorithm maintains $\sum_{d=1}^m 2^d = \mathcal{O}(2^m)$ bins. Estimating bin means costs $\mathcal{O}(2^m)$ per round. Additionally, it performs the statistical test $\textcolor{red}{(\star)}$ over all bin pair of bins $(B_1, B_2)$ at each depth $d$, with $\mathcal{O}(4^d)$ pairs, totaling $\mathcal{O}(4^m)$ across depths. For each pair, it checks all intervals of the form $[s_1, t] \subseteq [\tau_{l,m}, t]$, leading to $\mathcal{O}(8^m)$ possibilities. Therefore, the total *per-round* worst-case time complexity is $\mathcal{O}(2^m + 4^m \cdot 8^m) = \mathcal{O}(32^m)$, and the *per-round* worst-case memory complexity is $\mathcal{O}(8^m)$.
>
> Since $m \leq \log(T)$, we conclude that $\texttt{MBDE}$ has an overall **worst-case time complexity $\mathcal{O}(T^6)$** and **memory $\mathcal{O}(T^4)$**: both **polynomial in $T$**. For small $m$ (e.g., $m \leq 5$), runtime is manageable for some problem instances. Designing adaptive, efficient algorithms with optimal regret remains an important open direction, even in $K$-armed settings *(Gerogiannis et al., 2024)*.
>
> We will add a paragraph discussing the computational complexity in the main text.
>
> ### **Extension to smooth bandits (same answer as `scsD` and `GmBd`)**
> **Our algorithm and analysis can easily be adapted beyond $1$-Lipschitz reward function, and still enjoy minimax optimal rate**. For the sake of space, we give a sketch of proof of this generalization (similarly to *Appendix G* for the multi-dimensional setting).
>
> Our method generalizes to **$(\kappa, \beta)$-Hölder mean rewards** $(\mu_t)_t$ satisfying:
>
> $$\forall x,x'\in[0,1],\quad |\mu_t(x)-\mu_t(x')| \leq \kappa|x - x'|^\beta.$$
>
> In the stationary setting with horizon $\tau$, the minimax regret is $\tau^{\frac{1+\beta}{1+2\beta}} \kappa^{\frac{1}{1+2\beta}}$, achieved by discretizing into $K_\text{opt}(\tau,\kappa,\beta) = \tau^{\frac{1}{1+2\beta}}\kappa^{\frac{2}{1+2\beta}}$ bins *(Kleinberg, 2004)*. This motivates **defining significant regret** over $[s_1,s_2]$ as:
>
> $$\sum_{t=s_1}^{s_2} \delta_t(x) \geq \log(T)(s_2 - s_1)^{\frac{1+\beta}{1+2\beta}} \kappa^{\frac{1}{1+2\beta}}.$$
>
> We consider blocks of size $\tau_{l,m+1}-\tau_{l,m}=2^{m(1+2\beta)}/\kappa^2$, yielding $2^m$ bins and expected regret of $2^{m(1+\beta)}$ over this block if the rewards are stable enough. Replays at depth $d$ last $\ell(d) = 2^{d(1+2\beta)}/\kappa^2$ and use $2^d$ bins. The bias for estimating the cumulative reward gap is $4(s_2 - s_1)\kappa/2^{d\beta}$. This motivates the new **eviction rule**:
>
> $$\max_{B'}\sum_{t=s_1}^{s_2} \hat\delta_t(B',B) > c_0\log(T)\sqrt{(s_2-s_1)2^d} + 4\kappa(s_2 - s_1)/2^{d\beta},$$
>
> where the $c_0\approx 20$ is exactly the same as in the 1-Lipschitz setting.
>
> Replay at depth $d$ starting at round $s$ is scheduled with **probability**:
>
> $$p_{s,d} = \frac{1}{\kappa} \cdot \frac{2^{d(1+2\beta)/2}}{\sqrt{s - \tau_{l,m}}} \quad (d < m).$$
>
> Each block includes $2^{(1+2\beta)(m-d)/2}$ replays at depth $d$, each incurring regret $2^{((1+2\beta)(m-d) + 2d(1+\beta))/2}$. Summing over all dephs $d < m$ gives total replay regret $\mathcal{O}(2^{m(1+\beta)})$, matching the optimal bound per block.
>
> The expected replay interval at depth $d$ is (up to numerical constants that do not depend on $\kappa$ and $\beta$) $2^{(1+2\beta)(m+d)/2}$, while shifts of size $2^{-d\beta}$ become detectable within $2^{m(1+\beta)}$ rounds. Each episode $t_{l+1}-t_l$ has at most $M_l = \log_2((t_{l+1}-t_l)\kappa)/(1+2\beta)$ blocks, and the regret per episode is upper bounded as
>
> $$\sum_{m=1}^{M_l} 2^{m(1+\beta)} \leq (t_{l+1}-t_l)^{\frac{1+\beta}{1+2\beta}} \kappa^{\frac{1}{1+2\beta}}.$$
>
> Thus, the total dynamic regret of $\texttt{MBDE}$ is upper bounded as:
>
> $$\mathbb{E}[R_T] \leq \tilde{\mathcal{O}}\left(T^{\frac{1+\beta}{1+2\beta}} \tilde{L}_T^{\frac{\beta}{1+2\beta}} \kappa^{\frac{1}{1+2\beta}} \right).$$
> $ \tilde{\mathcal{O}}(\cdot)$ only hides poly-logarithmic factors and numerical constant **that does not depend on $\kappa$ or $\beta$**.
>
> With the exact same arguments as in our Lower bound proof  *(Appendix F)*, we show that **this rate is indeed minimax-optimal with respect to $T, \tilde{L}_T, \kappa$ and $\beta$**.
>
> ### **Technical novelty (same answer as `VifP`)**
> The main challenge in our setting lies in estimating a continuous, non-stationary mean reward function without prior knowledge of the magnitude of distributional shifts—or, equivalently, the discretization level and time scale at which such shifts become detectable. Coarse discretizations are essential to **rapidly** detect large shifts, while finer discretizations are required to capture smaller yet statistically significant changes. This motivates a **multi-scale** replay framework in which each scale contributes adaptively based on the magnitude of the underlying shift.
>
> To address this challenge, we propose a hierarchical replay mechanism spanning multiple scales, where each scale corresponds to a different discretization of the action space $[0,1]$. In contrast, prior works that track significant shifts operate in the $K$-armed setting, where the action space remains **fixed and finite** even when contexts are continuous. Their replay mechanisms and regret analyses therefore rely on a fixed set of arms, without the need to estimate or track reward changes across multiple scales.
>
> Our algorithm must face three fundamental challenges: ***(i)* scheduling** (potentially simultaneous) replays at different depths, ***(ii)* leveraging the information collected across different scales**, and ***(iii)* tracking the regret contribution from all scales simultaneously**. In our regret analysis, each action $x_t \in [0,1]$ belongs to several bins across discretization levels, and we explicitly quantify the regret incurred at each scale. Moreover, to decide which bin to activate at a given depth and time step, we introduce a principled strategy for ***(a)* scheduling replays at specific scales** and ***(b)* selecting the appropriate bin at a given active depth** through a carefully designed sampling algorithm.
>
> We will incorporate this discussion at the beginning of the proof sketch in *Section 5* to clarify the novelty and technical depth of our theoretical contribution.
>
> ### **References**
> - *Kleinberg. "Nearly tight bounds for the continuum-armed bandit problem." NIPS 2004.*

---

> > ### Comment · Reviewer_kQLa · 2025-08-04
> >
> > Thanks for the clarification and the newly added numerical results. It would be helpful to include a brief takeaway or interpretation of the simulation—what should we learn from the comparison of the numbers? Additionally, I think reporting the slope of the log-log plot (though I understand the plot cannot be shown here due to the policy) could provide further support for your regret bound.

---

> > > ### Author Response · Authors · 2025-08-04
> > >
> > > Thanks for the reply and the helpful suggestions!
> > >
> > > The reported results show the *cumulative* dynamic regret, averaged over 100 independent runs (we show mean ± 95% confidence interval). The main takeaway is that our algorithm, $\texttt{MBDE}$, can be implemented (at least on toy numerical experiments) and successfully **adapts to non-stationarity**. This is evidenced by its **significantly lower cumulative regret compared to $\texttt{BinningUCB (no restart)}$**, which does not reset its estimates and thus struggles in changing Lipschitz bandits environments.
> > >
> > > In contrast, $\texttt{BinningUCB (oracle)}$ represents an **idealized baseline with perfect oracle knowledge** of when to reset estimates. While our method does not match this oracle performance, it shows clear improvements over the non-adaptive baseline and offers a principled approach to handling non-stationarity without requiring this oracle information.
> > >
> > > We agree that reporting the *log-log* plot would provide further insight on the regret scaling; we will include this in the revised version of the paper if accepted.

---

> > > > ### Comment · Reviewer_kQLa · 2025-08-04
> > > >
> > > > Thanks for your response. I'm happy to raise the score to a clear accept.

---

### Official Review · Reviewer_scsD · 2025-07-03

**Clarity:** 3
**Significance:** 3
**Originality:** 2
**Rating:** 5
**Confidence:** 3

**Summary:**

This paper studies a fundamental variant of the multi-armed bandit problem in which the mean reward function is Lipschitz in space $[0,1]$) and non-stationary in time. The paper considers the notion of significant changes — a refined measure of temporal non-stationarity that generalizes earlier notions like total variation or number of abrupt changes. The authors develop an algorithm that adaptively discretizes the arm space. They achieve a minimax optimal dynamic regret of $\widetilde{L}^{1/3} T^{2/3}$ where $\widetilde{L}$ is the number of significant shifts.

**Questions:**

Do the result hold in higher dimensions?

**Ethical Concerns:**

["NO or VERY MINOR ethics concerns only"]

**Final Justification:**

I will maintain my initial evaluation.

**Limitations:**

see above

**Quality:**

3

**Strengths And Weaknesses:**

**Strengths**
- Fundamental problem: The setting of spatially Lipschitz and temporally non-stationary bandits is both natural and under-explored.
- Novel technical contributions: A discretization strategy based on dyadic trees to efficiently monitor changes across scales; elimination and replay mechanisms tailored to the continuous setting.
- Theoretical guarantees: The $\widetilde{L}^{1/3} T^{2/3}$ regret is minimax optimal (in T and $\tilde L$, at least)

**Weakness:** The results are stated only for Lipschitz constant L=1. The work would be more complete if they could incorporate L and compare the dependence with stationary Lipschitz bandits.

minor comments:
- The phrase "locally stationary" phases (line 52) is unclear. Please provide a more precise definition or intuition.
- The “three-way” trade-off mentioned under forced exploration (Section 3.2) appears to be more of a two-way trade-off: frequency of restarts and discretization granularity. Please clarify the third dimension.

---

> ### Author Rebuttal · Authors · 2025-07-30
>
> We thank the reviewer for the feedback.
>
> ### **Extension to other Lipschitz constants (same answer as  `kQLa` and `GmBd`)**
> **Our algorithm and analysis can easily be adapted beyond $1$-Lipschitz reward function, and still enjoy minimax optimal rate**. For the sake of space, we give a sketch of proof of this generalization (similarly to *Appendix G* for the multi-dimensional setting).
>
> Our method generalizes to **$(\kappa, \beta)$-Hölder mean rewards** $(\mu_t)_t$ satisfying:
>
> $$\forall x,x'\in[0,1],\quad |\mu_t(x)-\mu_t(x')| \leq \kappa|x - x'|^\beta.$$
>
> In the stationary setting with horizon $\tau$, the minimax regret is $\tau^{\frac{1+\beta}{1+2\beta}} \kappa^{\frac{1}{1+2\beta}}$, achieved by discretizing into $K_\text{opt}(\tau,\kappa,\beta) = \tau^{\frac{1}{1+2\beta}}\kappa^{\frac{2}{1+2\beta}}$ bins *(Kleinberg, 2004)*. This motivates **defining significant regret** over $[s_1,s_2]$ as:
>
> $$\sum_{t=s_1}^{s_2} \delta_t(x) \geq \log(T)(s_2 - s_1)^{\frac{1+\beta}{1+2\beta}} \kappa^{\frac{1}{1+2\beta}}.$$
>
> We consider blocks of size $\tau_{l,m+1}-\tau_{l,m}=2^{m(1+2\beta)}/\kappa^2$, yielding $2^m$ bins and expected regret of $2^{m(1+\beta)}$ over this block if the rewards are stable enough. Replays at depth $d$ last $\ell(d) = 2^{d(1+2\beta)}/\kappa^2$ and use $2^d$ bins. The bias for estimating the cumulative reward gap is $4(s_2 - s_1)\kappa/2^{d\beta}$. This motivates the new **eviction rule**:
>
> $$\max_{B'}\sum_{t=s_1}^{s_2} \hat\delta_t(B',B) > c_0\log(T)\sqrt{(s_2-s_1)2^d} + 4\kappa(s_2 - s_1)/2^{d\beta},$$
>
> where the $c_0\approx 20$ is exactly the same as in the 1-Lipschitz setting.
>
> Replay at depth $d$ starting at round $s$ is scheduled with **probability**:
>
> $$p_{s,d} = \frac{1}{\kappa} \cdot \frac{2^{d(1+2\beta)/2}}{\sqrt{s - \tau_{l,m}}} \quad (d < m).$$
>
> Each block includes $2^{(1+2\beta)(m-d)/2}$ replays at depth $d$, each incurring regret $2^{((1+2\beta)(m-d) + 2d(1+\beta))/2}$. Summing over all dephs $d < m$ gives total replay regret $\mathcal{O}(2^{m(1+\beta)})$, matching the optimal bound per block.
>
> The expected replay interval at depth $d$ is (up to numerical constants that do not depend on $\kappa$ and $\beta$) $2^{(1+2\beta)(m+d)/2}$, while shifts of size $2^{-d\beta}$ become detectable within $2^{m(1+\beta)}$ rounds. Each episode $t_{l+1}-t_l$ has at most $M_l = \log_2((t_{l+1}-t_l)\kappa)/(1+2\beta)$ blocks, and the regret per episode is upper bounded as
>
> $$\sum_{m=1}^{M_l} 2^{m(1+\beta)} \leq (t_{l+1}-t_l)^{\frac{1+\beta}{1+2\beta}} \kappa^{\frac{1}{1+2\beta}}.$$
>
> Thus, the total dynamic regret of $\texttt{MBDE}$ is upper bounded as:
>
> $$\mathbb{E}[R_T] \leq \tilde{\mathcal{O}}\left(T^{\frac{1+\beta}{1+2\beta}} \tilde{L}_T^{\frac{\beta}{1+2\beta}} \kappa^{\frac{1}{1+2\beta}} \right).$$
> $ \tilde{\mathcal{O}}(\cdot)$ only hides poly-logarithmic factors and numerical constant **that does not depend on $\kappa$ or $\beta$**.
>
> With the exact same arguments as in our Lower bound proof  *(Appendix F)*, we show that **this rate is indeed minimax-optimal with respect to $T, \tilde{L}_T, \kappa$ and $\beta$**.
>
> ### **Locally stationary**
> By locally stationary phases, we mean time intervals for which the reward function does not change. We thank the reviewer for pointing this flaw and we will clarify it in the final version.
>
> ### **Three-way tradeoff (same answer as `kQLa`)**
> The *three-way tradeoff* refers to the balance our algorithm must deal between: ***(i)* exploration**, via replays to detect changes; ***(ii)* exploitation**, by choosing bins with the highest estimated rewards; and ***(iii)* discretization level**, that is, choosing the appropriate discretization depth at each time step. This third dimension arises specifically from our setting of Lipschitz mean reward functions that change over time. We will clarify this in Section 1.1.
>
> ### **Extension to multi-dimension**
> **Yes**, we showed how to adapt our algorithm and analysis in $p-$dimensional action space in **Appendix G**.
>
> ### **References**
> - *Kleinberg. "Nearly tight bounds for the continuum-armed bandit problem." NIPS 2004.*

---

### Official Review · Reviewer_VifP · 2025-07-04

**Clarity:** 2
**Significance:** 3
**Originality:** 3
**Rating:** 4
**Confidence:** 3

**Summary:**

This paper considers a fundamental online learning problem of non-stationary Lipschitz bandits. In particular, the notion of non-stationarity is new, different from previously studied ones of $L_T$ and $V_T$, both of which could be very large even if the best arm is unchanged. This new notion is called significant shifts introduced recently in other bandit problems, and the authors then propose a Multi-Depth Bin Elimination algorithm to achieve optimal regret for Lipschitz bandits under significant shifts.

**Questions:**

- The difference of significant shifts is still quite abstract. Can the authors provide some more concrete examples to help understanding? And it is more ideal if such example has strong practical motivation/relevance.
- Can the authors provide a more exact characterization of $c_0$ as mentioned in the footnote? How does it affect the algorithm performance?
- Can the authors elaborate more on the unique challenges presented in the analysis of this problem, in particular, compared to previous work by Suk and coauthors?
- Despite no numerical implementation, can the authors at least analyze the computation complexity? This would give a better sense of the feasibility of proposed approach.

**Ethical Concerns:**

["NO or VERY MINOR ethics concerns only"]

**Final Justification:**

I thank the authors’ response in their technical novelty and soundness and maintain my positive recommendation.

**Limitations:**

- No numerical experiments as acknowledged by the authors that this is pure theoretical work.

**Quality:**

3

**Strengths And Weaknesses:**

### Strength
- This paper tackles a theoretically fundamental problem in bandits with infinite continuum actions and new notion of non-stationarity.
- The results are strong with optimal rates.

### Weakness
- The presentation of algorithm is not very clear: there are two algorithms for separate submodules but actually not an overall algorithm presented in block algorithm environment.
- The explained intuition of proving Theorem also seems incomplete. It would be nice to have a more high-level proof framework connecting dots together before going into details. Some schematic illustration would also be very helpful. As in current presentation, it is hard to evaluate confidently the correctness of proof.
- It appears that the main algorithm design is a combination of binning/elimination of seminal works and the replay mechanism of recent work  Suk and Kpotufe [2022] (the same authors who also proposed significant shifts). As a result, it is not very evident how much of the theoretical contribution is conditioning on these two.

### Minor
- A few typos such as [ and ] in line 127, line 182.

---

> ### Author Rebuttal · Authors · 2025-07-30
>
> We thank the reviewer for the comment.
>
> ### **Presentation**
> We appreciate the feedback regarding the presentation and will improve it by including an illustrative figure of the sampling scheme *(Algorithm 2)* in the main text, using the space provided in the additional page if our paper is accepted.
>
> ### **Concrete benefits of the significant shift metric (same answer as `9z8n`)**
> Consider a recommendation system with a continuous pool of content (e.g., movies), indexed by $x \in [0,1]$, where nearby values of $x$ correspond to similar content types. Suppose there exists two regions of high and comparable user preference, centered around $x_1 = 0.3$ and $x_2 = 0.7$. Imagine a scenario where preferences near $x_1$ remain stable over time (*e.g*., a timeless classic), while preferences near $x_2$ undergo very frequent changes (*e.g*., a trending topic that evolves daily).
>
> In this case, an algorithm that **consistently recommends content near $x_1$** would incur **little to no regret, even though the underlying reward function changes frequently** in other regions. From a global perspective, the *number of changes* or the *total variation in mean reward* could be **as large as $L_T = V_T= \mathcal{O}(T)$**. An algorithm that relies solely on such metrics would unnecessarily restart its estimates too frequently, as it would **overestimate** the effective difficulty of the problem. This leads to both theoretical suboptimality and practical inefficiency. However, under our framework, such changes do **not** constitute a significant shift. This example shows a key strength of our metric: it captures only those changes that are **statistically consequential** to learning, rather than indiscriminately counting all shifts in the environment.
>
> We will include this example at the end of *Section 2.2* to further clarify the motivation for our definition.
>
> ### **Constant $c_0$**
> The constant $c_0$ is given in our theoretical analysis *(see line 568, in the supplementary material)*:
> $$c_0 = 3 + 7(e - 1)\sqrt{2} \approx 20.$$
> This constant is derived from the confidence bounds of *Proposition 2*, and ensures that the test procedure $\textcolor{red}{(\star)}$ used to trigger bin evictions is **theoretically valid**. However, this theoretically justified choice leads to conservative behavior in practice: bins are evicted only when the statistical evidence is strong. To balance theory with empirical performance, we use a smaller constant in our numerical experiments (see the numerical results paragraph in this rebuttal, where we specifically choose $c_0 = 1$), which leads to faster elimination and better regret in practice.
>
> ### **Unique challenges in our analysis (same answer as `kQLa`)**
> The main challenge in our setting lies in estimating a continuous, non-stationary mean reward function without prior knowledge of the magnitude of distributional shifts—or, equivalently, the discretization level and time scale at which such shifts become detectable. Coarse discretizations are essential to **rapidly** detect large shifts, while finer discretizations are required to capture smaller yet statistically significant changes. This motivates a **multi-scale** replay framework in which each scale contributes adaptively based on the magnitude of the underlying shift.
>
> To address this challenge, we propose a hierarchical replay mechanism spanning multiple scales, where each scale corresponds to a different discretization of the action space $[0,1]$. In contrast, prior works that track significant shifts operate in the $K$-armed setting, where the action space remains **fixed and finite** even when contexts are continuous. Their replay mechanisms and regret analyses therefore rely on a fixed set of arms, without the need to estimate or track reward changes across multiple scales.
>
> Our algorithm must face three fundamental challenges: ***(i)* scheduling** (potentially simultaneous) replays at different depths, ***(ii)* leveraging the information collected across different scales**, and ***(iii)* tracking the regret contribution from all scales simultaneously**. In our regret analysis, each action $x_t \in [0,1]$ belongs to several bins across discretization levels, and we explicitly quantify the regret incurred at each scale. Moreover, to decide which bin to activate at a given depth and time step, we introduce a principled strategy for ***(a)* scheduling replays at specific scales** and ***(b)* selecting the appropriate bin at a given active depth** through a carefully designed sampling algorithm.
>
> We will incorporate this discussion at the beginning of the proof sketch in *Section 5* to clarify the novelty and technical depth of our theoretical contribution.
>
> ### **Numerical results (same answer as `kQLa`, `9z8n` and `GmBd`)**
> **We implemented our algorithm and evaluated its performance**. Because of the NeurIPS policy, we cannot share code at this stage but will release it upon acceptance.
>
> We benchmark against two baselines: $\texttt{BinningUCB (no restart)}$, which uses $K(T)\propto T^{1/3}$ and never resets, and $\texttt{BinningUCB (oracle)}$, which **knows** the true change points $\tau_i$ and resets its estimate at the end phase. At each phase, it uses the optimal per-phase discretization $K_i \propto (\tau_{i+1}-\tau_i)^{1/3}$.
>
> We simulate a $1$-Lipschitz, piecewise-linear reward on $[0,1]$ with a peak shifting from $x=0.3$ to $x=0.7$ every $10^5$ rounds over horizon $T=10^6$, inducing $\tilde{L}_T = 10$ significant shifts. Regret is averaged over $100$ runs (rounded to nearest integer).
>
> | **Step (×10³)** | $\texttt{BUCB (no restart)}$ | $\texttt{BUCB (oracle)}$ | **$\texttt{MBDE}$ (ours)**    |
> |----------------:|-------------------------:|---------------------:|---------------------:|
> | 10              | 2137 ± 101               | 551 ± 30             | **1566 ± 108**        |
> | 50              | 10580 ± 393              | 2794 ± 147           | **8556 ± 349**        |
> | 100             | 21007 ± 890              | 5596 ± 396           | **17111 ± 667**       |
>
> Our method adapts to non-stationarity via replays, significantly outperforming $\texttt{BUCB (no restart)}$.
>
> ### **Computational complexity (same answer as `kQLa`, `9z8n` and `GmBd`)**
> Our algorithm is feasible for small-scale problems, as shown in experiments, but we acknowledge that its time and memory complexity limits its scalability. We detail the worst-case computational cost below.
>
> In a block of length $\tau_{l,m+1}-\tau_{l,m}=8^m$, the algorithm maintains $\sum_{d=1}^m 2^d = \mathcal{O}(2^m)$ bins. Estimating bin means costs $\mathcal{O}(2^m)$ per round. Additionally, it performs the statistical test $\textcolor{red}{(\star)}$ over all bin pair of bins $(B_1, B_2)$ at each depth $d$, with $\mathcal{O}(4^d)$ pairs, totaling $\mathcal{O}(4^m)$ across depths. For each pair, it checks all intervals of the form $[s_1, t] \subseteq [\tau_{l,m}, t]$, leading to $\mathcal{O}(8^m)$ possibilities. Therefore, the total *per-round* worst-case time complexity is $\mathcal{O}(2^m + 4^m \cdot 8^m) = \mathcal{O}(32^m)$, and the *per-round* worst-case memory complexity is $\mathcal{O}(8^m)$.
>
> Since $m \leq \log(T)$, we conclude that $\texttt{MBDE}$ has an overall **worst-case time complexity $\mathcal{O}(T^6)$** and **memory $\mathcal{O}(T^4)$**: both **polynomial in $T$**. For small $m$ (e.g., $m \leq 5$), runtime is manageable for some problem instances. Designing adaptive, efficient algorithms with optimal regret remains an important open direction, even in $K$-armed settings *(Gerogiannis et al., 2024)*.
>
> We will add a paragraph discussing the computational complexity in the main text.
>
> ### **References**
> - *Wei and Luo. "Non-stationary reinforcement learning without prior knowledge: An optimal black-box approach." COLT 2021.*
> - *Suk and Kpotufe. "Tracking most significant shifts in nonparametric contextual bandits." Neurips 2023.*
> - *Gerogiannis, Huang and Veeravalli. Is Prior-Free Black-Box Non-Stationary Reinforcement Learning Feasible? AISTATS 2025.*

---

> > ### Comment · Reviewer_VifP · 2025-08-03
> > **Thanks for the response and keep my positive score**
> >
> > Thank you for the thoughtful response. The response helped to address the concerns that I have and I recommend the authors incorporating some of them in the revision. I maintain my positive score.

---

### Decision · Program_Chairs · 2025-09-17

**Decision:**

Accept (poster)

**Comment:**

I concur with the reviewers’ positive view of the submission and think the paper makes a valuable addition. The work combines the well-studied problems of continuous-armed bandits and of non-stationary (finite-action) multi-armed bandits in a nice way to form a new problem and proposes an algorithm for the problem that achieves minimax optimal regret without knowledge of the non-stationarity. The technical contributions are solid and well situated in the literature. Some clarity and presentation issues were raised, as well as the lack of broader empirical validation, but the authors have addressed these in the rebuttal with clarifications and a small experiment. I trust these will be incorporated in the camera ready.